# Theta-phase locking of single neurons during human spatial memory

Tim A. Guth [1,2] ✉, Armin Brandt[2], Peter C. Reinacher [3,4],
Andreas Schulze-Bonhage [2], Joshua Jacobs [5,6] & Lukas Kunz [1] ✉

Memory processes may rely on complex interactions between single-neuron activity and local field potentials. To better understand such spike–field relationships in humans, we examined human theta-phase locking—neuronal firing at similar theta phases over time—using single-neuron recordings in epilepsy patients performing a spatial memory task. Applying frequency-adaptive theta-phase estimation in a broad 1–10 Hz frequency range, we found that theta-phase locking was widespread in the human medial temporal lobe during memory encoding and retrieval. Time-resolved spectral parameterization and cycle-by-cycle analysis demonstrated stronger theta-phase locking during steep aperiodic slopes and prominent theta oscillations. Phase-locking strength was similar across successful and unsuccessful memory trials, with most neurons spiking at similar theta phases during encoding and retrieval. Some neurons shifted their preferred phase, supporting theories that encoding and retrieval are separated within the theta cycle. These results show how local field potential properties and memory states influence human theta-phase locking.

The activity of single neurons is embedded into waxing and waning local field potentials that represent the combined electrical activity of small neuronal populations within the neurons' vicinity[1–5]. A major component of local field potentials are fluctuations in the theta-frequency range at around 4–8 Hz[6–12]. They are prominent in electrophysiological recordings from the rodent hippocampus and occur consistently when the animal navigates through its spatial environment[7,13–17]. This theta rhythm may play a critical role in neural computation by grouping and segregating neuronal assemblies[7,18,19] and may thus be involved in cognitive functions including memory[6,7,20–25]. In recent years, studies with epilepsy patients performing virtual and real-world navigation and memory tasks have described similar fluctuations of hippocampal local field potentials in humans, both in high and low theta-frequency ranges (1–5 and 6–10 Hz, respectively)[8,16,26–33]. Human theta oscillations appear less

clear and stable than in rodents, however[8,9,16,33,34]. It thus remains unclear to what extent their putative properties and functions—including their interactions with single-neuron activity—generalize from rodents to humans.

A prominent observation on the relationship between single neurons and theta-frequency local field potentials is theta-phase locking[35–40]. This phenomenon describes that a neuron activates at similar theta phases over time, for example at theta peaks or troughs. Theta-phase locking contrasts with other theta phase-dependent phenomena such as theta-phase precession where single neurons activate at successively earlier theta phases over time[13,41–46]. In rodents, theta-phase locking has been described in a variety of neurons including spatially-modulated neurons representing locations and directions[35,41,47–50]. Due to its high prevalence both within and across brain regions, theta-phase locking has been suggested to organize the

[1]Department of Epileptology, University Hospital Bonn, Bonn, Germany. [2]Epilepsy Center, Medical Center—University of Freiburg, Faculty of Medicine, University of Freiburg, Freiburg, Germany. [3]Department of Stereotactic and Functional Neurosurgery, Medical Center—University of Freiburg, Faculty of Medicine, University of Freiburg, Freiburg, Germany. [4]Fraunhofer Institute for Laser Technology, Aachen, Germany. [5]Department of Biomedical Engineering, Columbia University, New York, NY, USA. [6]Department of Neurological Surgery, Columbia University Medical Center, New York, NY, USA.
✉ e-mail: Tim.Guth@ukbonn.de; Lukas.Kunz@ukbonn.de

activity of single neurons into neuronal assemblies[22,33,40], to enable interregional communication between neurons[36,37,40,51], and to trigger which neurons become linked during encoding to participate in the same replay events during consolidation[47]. Phase locking of neuronal assemblies at distinct theta phases may furthermore help separate different mnemonic processes such as encoding and retrieval[50,52,53].

Based on this foundational work mostly performed in rodents, previous studies using single-neuron recordings in epilepsy patients started investigating the question whether theta-phase locking is present in the human brain as well, and whether it might be involved in human memory processes[38,39]. These studies showed that, during a virtual spatial navigation task, neurons in widespread brain regions including hippocampus, amygdala, and parahippocampal regions were phase locked to oscillations in the theta-frequency range[38]. Similar to observations in rodents[36,37], theta-phase locking also occurred between neurons in extrahippocampal regions and the hippocampal theta rhythm in patients performing another set of virtual spatial navigation tasks[40]. In a recognition memory task with static images, stronger theta-phase locking was associated with successful memory formation as assessed with spike-field coherence[39]. Similarly, in patients performing a verbal free recall task, a subset of neurons whose firing rates exhibited a subsequent memory effect showed stronger theta-phase locking during successful versus unsuccessful encoding events[54]. In that study, the authors also provided evidence for theta-phase shifts between encoding and retrieval in human neurons[54]. Together, these prior observations indicate that human single-neuron activity is related to local field potentials in the theta-frequency range, and they suggest that these relationships support memory in humans.

Inspired by this previous work, we aimed at extending our basic understanding of theta-phase locking in the human medial temporal lobe in this study. Using human single-neuron recordings during a virtual navigation task with separate and self-paced periods for memory encoding and retrieval[55–58], we specifically aimed at investigating theta-phase locking as a function of different electrophysiological properties of the local field potential and across different memory states. Because human theta oscillations are relatively unstable over time[27,28,30,33] and considerably variable in frequency, we used a generalized-phase approach[59] to identify theta phases; time-resolved spectral parameterization[60,61] to distinguish periods with varying aperiodic activity; and a cycle-by-cycle approach to identify theta oscillations[62]. Our results show that theta-phase locking is a prominent phenomenon in regions of the human medial temporal lobe and that it is strongly dependent on electrophysiological properties of the local field potential. Neuronal theta-phase locking was largely stable across different memory states and many neurons showed preferred theta phases that were similar between encoding and retrieval, with a few neurons showing significant shifts between encoding and retrieval. These findings provide insights into the analysis and properties of human theta-phase locking; may help identify the relevance of theta-phase locking for human memory processes; and may trigger further investigations into the question of how theta-phase locking supports information processing in the human brain.

## Results

### Spatial memory task

To investigate various determinants of theta-phase locking in the human medial temporal lobe, we performed direct neurophysiological recordings in epilepsy patients[63,64] during a spatial memory task[55] (Table S1). In this "Treasure Hunt" task, participants actively navigated a virtual environment using a game controller and were asked to encode and remember the locations of objects within the environment (Fig. 1; Methods). Briefly, during the encoding period of each trial, participants encountered two or three objects in treasure chests positioned at random locations on a virtual beach. They were then passively moved to an elevated recall position from where they could oversee the beach. After a short distractor period, participants performed cued recall. During location-cued object recall, they were cued with the locations and asked to speak aloud the names of the objects they had found at these locations. During object-cued location recall, they were presented with the objects and indicated the objects' remembered locations on the beach. In a full session, participants aimed at encoding and remembering a total of 100 unique object–location associations.

Eighteen epilepsy patients participated in the task and contributed a total of 27 sessions. Participants performed the task for $60 \pm 2$ min (mean ± SEM), and their memory performance was similar to previous studies with this task[55–57] (Fig. 1C, E). Participants showed memory-performance improvements over time for both types of memory recall, indicating that they successfully acquired knowledge of the spatial environment (object recall: two-sided paired $t$-test between the first and second half of all trials, $t(26) = -3.402$, $P = 0.002$; location recall: two-sided paired $t$-test, $t(26) = -2.786$, $P = 0.010$; $n = 27$ sessions; Fig. 1C, E). Better performance during object recall was correlated with better performance during location recall (Spearman's $rho = 0.879$, $P < 0.001$, $n = 27$ sessions). This task thus provided us with the opportunity to investigate theta-phase locking during memory encoding and retrieval, and as a function of memory performance.

### Human single neurons phase lock to the local theta rhythm

We performed direct neural recordings with high spatiotemporal resolution using Behnke-Fried microelectrodes[63–67] that allowed us to identify the activity of 1025 neurons in various regions of the human medial temporal lobe (Fig. 2A; Fig. S1). We excluded neurons with fewer than 25 spikes (i.e., action potentials[1,39]) in total or no spikes in more than 80% of segments across any trial condition, resulting in a total number of 666 neurons for all analyses.

We also extracted the low-frequency component of the microelectrode data in a broad theta-frequency range (1–10 Hz) using a generalized phase approach[59] to characterize the timing relationship between the neurons' spiking activity and the theta rhythm. We opted for this approach with a broad filter band of 1–10 Hz as our visual inspection of the data identified many periods in which the local field potential showed fluctuations and oscillatory activity with rapidly changing frequencies, similar to previous observations of non-stationary theta oscillations in bats and humans[16,27,28,68]. Our preliminary analyses with the filter-Hilbert method including narrow-band filtering often failed to fit these fluctuations neatly, and we thus decided to treat them as a coherent entity and opted for the generalized phase approach instead. This led to a good fit between the original and the filtered signal in the theta-frequency range (e.g., Fig. 2B; Fig. S2).

In addition to this better fit of the raw signal, our choice of using a broad 1–10 Hz frequency range was motivated by several previous studies using intracranial neural recordings in humans. These studies showed that behavior-associated theta oscillations in humans occur across higher and lower theta frequencies, which extend beyond the traditional 4–8 Hz theta band and are often referred to as high theta (6–10 Hz) and low theta (1–5 Hz)[16,29,31,32,69]. For example, previous work found that increased low-theta power at 1–3 Hz is related to successful memory encoding[31,57]; that 3-Hz oscillations are often present during virtual spatial navigation[16,32]; and that high- and low-theta power increases before the onset of virtual movement[69]. Hence, to not exclude any components of high theta and low theta, we opted for the 1–10 Hz frequency range and refer to fluctuations within this range as theta throughout the manuscript. We acknowledge though that human theta has been defined in various ways (e.g., as 4–7.5 Hz[70] or as 4–8 Hz[6]) and that the 1–10 Hz band includes parts of the traditional delta and alpha bands[70].

To obtain phase estimates within this broad frequency band, we used the recently developed generalized phase approach[59]. In brief,

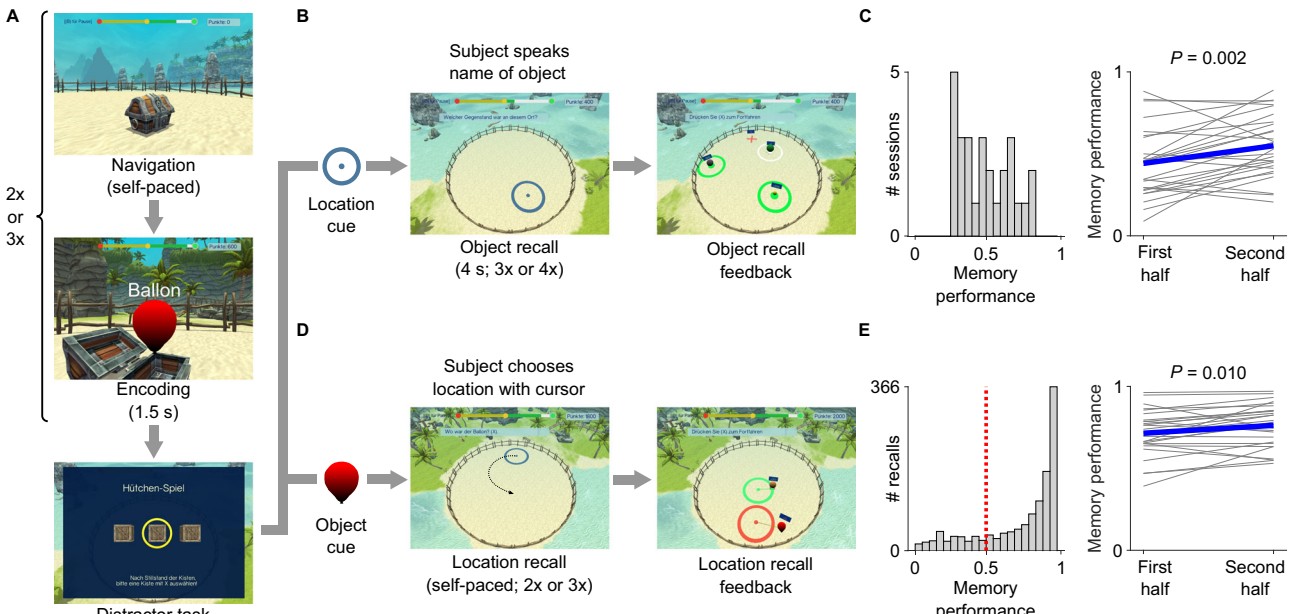

**Fig. 1 | Spatial memory task. A** During each navigation–encoding period of the task, participants freely and actively navigated a virtual beach environment using a game controller and successively encountered two or three unique objects in treasure chests at random locations on the beach. Participants aimed at encoding the objects and their associated locations to recall them during the retrieval period. After the navigation–encoding period, participants were moved to an elevated recall position where they completed a distractor task to prevent the participants from actively rehearsing the objects and their associated locations. After the distractor task, participants performed two types of memory recall. **B** During location-cued object recall, participants were presented with a location on the beach and recalled the associated object they had found at that location during encoding. **C** Histogram of session-wise memory performance during object recall (left) and improvement of object-recall memory performance over time, i.e., during the first half of all trials versus the second half of all trials (right; two-sided paired $t$-test: $t(26) = -3.402$, $P = 0.002$; $n = 27$ sessions). We computed object-recall performance in each session as the ratio of the number of successfully recalled objects divided by

the total number of object recalls. For instance, in session number one, the participant successfully recalled 9 of 33 objects, resulting in an object-recall performance of 0.273. **D** During object-cued location recall, participants viewed an object and aimed at recalling the associated location on the beach. **E** Histogram of trial-wise memory performance during location recall (left) and improvement of location-recall memory performance over time (right; two-sided paired $t$-test: $t(26) = -2.786$, $P = 0.010$; $n = 27$ sessions). We computed location-recall performance in each trial as the drop error (Euclidean distance) between the participant's response location and the correct location of the object, normalized by the possible distribution of drop errors. The closer to 1, the better memory performance. Red dotted line, chance performance. Gray lines, session-wise data; blue line, average. The virtual beach environment was created using the Unity 3D graphics engine and the Hand-painted Island pack (obtained from the Unity asset store under the Standard Unity Asset Store EULA; https://assetstore.unity.com/packages/3d/environments/fantasy/hand-painted-island-pack-36959#asset_quality). Source data are provided as a Source Data file.

the generalized phase approach involves filtering in a broad band (here, 1–10 Hz), computing the analytic signal using the Hilbert transform, estimating the phase from the analytic signal, and interpolating periods with high-frequency intrusions where phase progression reverses direction or phase progresses with a frequency <1 Hz (Fig. S3). Due to the 1/f-like shape of the power spectrum with stronger power in lower as compared to higher frequencies, generalized phase estimates are, by design, more strongly influenced by lower than higher frequencies. In comparison to the 5–40 Hz band of the original study using generalized phase[59], our study also included frequencies between 1 and 5 Hz to not miss low-theta effects. We thus performed control analyses to examine whether our frequency band might add effects related to arousal and whether it might be more prone to low-frequency intrusions (i.e., phase-distorting influences from frequencies below the band of interest). Analyzing the frequency distributions during the distractor period, we did not find evidence for the idea that lower arousal was related to a higher prevalence of 1–5 Hz activity (Fig. S4A). Furthermore, control analyses and simulations did not indicate that potential low-frequency intrusions biased the 1–10 Hz phase estimates in a relevant way (Figs. S2, S3, S5). Taken together, computing generalized phase within the 1–10 Hz frequency range appeared as a valid approach to investigate theta-phase locking in our data.

In a first step, we investigated general theta-phase locking of neuronal spikes across the entire task, without distinguishing between periods with or without clear theta oscillations, as done in prior

studies[38]. To quantify each neuron's phase-locking strength, we calculated the pairwise phase consistency (PPC) for the distribution of theta phases at which the neuron's spikes occurred[71] (if spike counts were too high to compute PPC, we used a Rayleigh test). To assess statistical significance, we ranked the empirical PPC within surrogate PPC values that we obtained by circularly shifting the theta phases relative to the action potentials and recomputing the PPC (alpha level, 0.05; Methods). We observed many neurons (single and multi-units) for which the spikes were strongly locked to a particular phase of the local field potentials filtered in our broad theta-frequency range (571 of 666 neurons; 86%; binomial test versus 5% chance, $P < 0.001$).

For example, a neuron in the left entorhinal cortex of a participant preferentially spiked at the theta troughs (one-sided surrogate analysis with circular shift of phase data, PPC = 0.496, $P < 0.001$, $n = 10,261$ spikes; Fig. 2B). Pooling all spikes from all neurons, we found that they were generally clustered around the theta trough (Rayleigh test, $z = 2.673 \times 10^4$, $P < 0.001$, $n = 7,779,085$ spikes; Fig. 2C). We obtained similar results when investigating theta-phase locking in more restricted theta-frequency bands (Figs. S6, S7, S8). When differentiating single units into putative interneurons and pyramidal cells[72,73], we found a higher percentage of theta-phase locking in putative interneurons compared to pyramidal cells (94.1% and 83.8%, respectively; two-sided surrogate analysis with unit-label shuffling, Δ = 10%, $P = 0.029$; Fig. S9). These findings replicate previous observations of strong theta-phase locking in the human medial temporal lobe[38,40] and indicate that there is a tight temporal relationship between single-

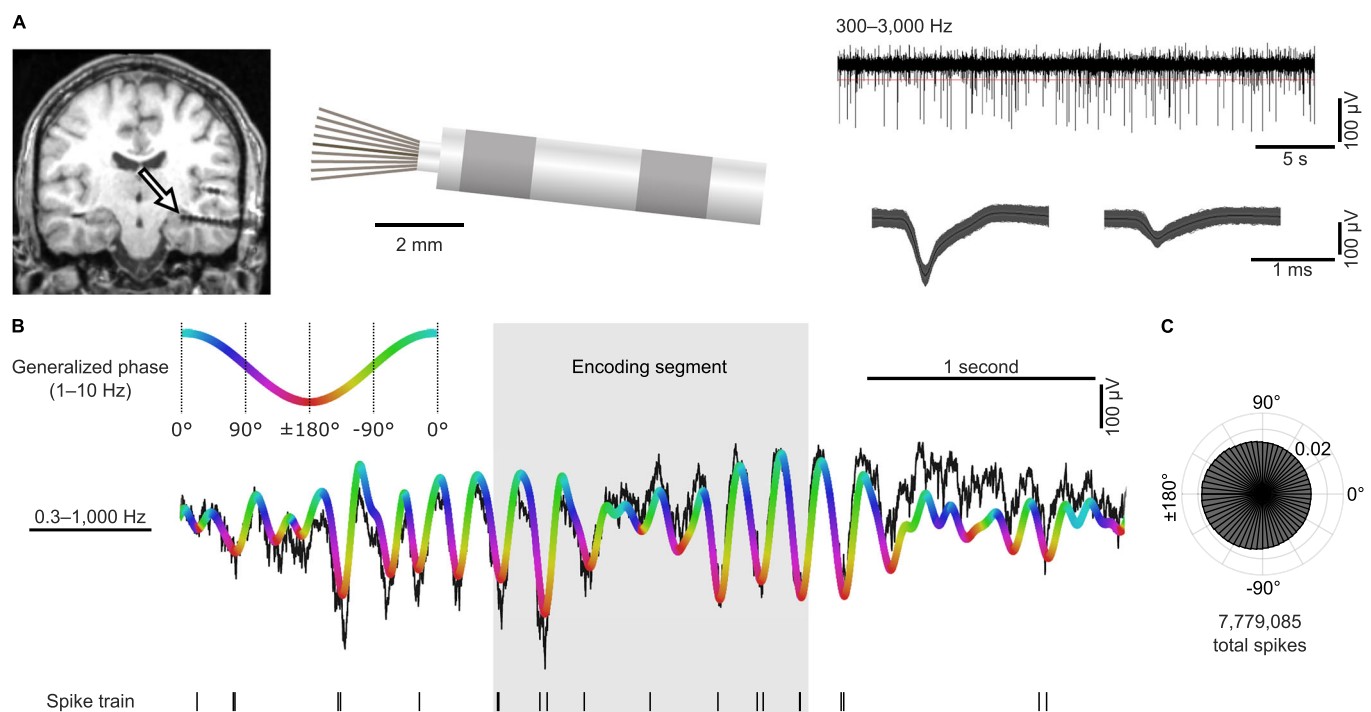

**Fig. 2 | Neural recordings and general theta-phase locking. A** Microelectrode recordings. Left, example post-implantation MRI. Arrow points at the tip of the Behnke−Fried electrode. Middle, schematic of the inner end of a Behnke−Fried electrode. Microelectrodes protrude from the tip of the macroelectrode by 3–5 mm. Top right, local field potential filtered between 300 and 3000 Hz. Horizontal red line, threshold for spike detection. Bottom right, spike waveforms of two units recorded on this microelectrode. **B** Analysis procedure. We filtered the local field potential in a broad theta-frequency range (1–10 Hz) and computed the generalized phase of the filtered trace. For each spike, we extracted its corresponding theta phase. **C** General theta-phase locking across the entire experiment. Spikes pooled across all neurons preferentially occurred at the theta trough. Panel (**A**) is adapted from Kunz, L., Staresina, B.P., Reinacher, P.C. et al. Ripple-locked coactivity of stimulus-specific neurons and human associative memory[96]. Nat Neurosci 27, 587–599 (2024). https://doi.org/10.1038/s41593-023-01550-x under a CC BY license: https://creativecommons.org/licenses/by/4.0/. Source data are provided as a Source Data file.

neuron activity and local low-frequency activity in the human medial temporal lobe.

## Theta-phase locking is increased for spikes associated with high theta power

Next, we aimed at identifying whether theta-phase locking is modulated by basic properties of the local field potential. We thus asked whether theta-phase locking was stronger during periods of elevated theta power. To this end, we computed the instantaneous theta power at each spike by squaring the magnitude of the complex signal obtained through the generalized phase approach. We found that theta-phase locking was present both during periods with high and during periods with low theta power using a median split (Rayleigh tests: high, $z = 2.851 \times 10^4$, $P_{corr.} < 0.001$, $n = 3,889,384$ spikes; low, $z = 3.918 \times 10^3$, $P_{corr.} < 0.001$, $n = 3,889,701$ spikes; Bonferroni corrected for two tests). When we directly compared theta-phase locking strengths between high versus low theta power, we found that theta-phase locking was more strongly expressed during periods with high theta power (one-sided surrogate analysis with spike-label shuffling, $\Delta z = 2.459 \times 10^4$, $P < 0.001$; Fig. 3A; Fig. S10), which is in line with previous results[38]. In designing the surrogate analysis, we ensured that the exact number of spikes in both conditions was maintained in all surrogate rounds, which is relevant as the Rayleigh $z$-value−unlike the PPC −increases with the spike count (Fig. S11).

We also tested whether this modulation of theta-phase locking by theta power varied between different trial periods including encoding, recall, and baseline (where the baseline period comprised the entire session except for encoding and recall periods). In each of these three experimental periods, we separated the spikes of each unit into groups of high and low theta power using a median split. We found that phase-

locking was consistently stronger in the high-power than in the low-power condition for baseline, encoding, and recall periods (one-sided surrogate analysis with condition-label swapping based on PPC values: baseline, $t(665) = 13.502$, $P_{corr.} < 0.001$; encoding, $t(665) = 12.393$, $P_{corr.} < 0.001$; recall, $t(665) = 13.305$, $P_{corr.} < 0.001$; Bonferroni corrected for three tests; Fig. 3B; Fig. S10). Overall, our results show that the modulation of theta-phase locking by theta power is a general phenomenon that is consistent across different memory states, at least in our spatial memory task.

## Phase locking to the local theta rhythm varies across medial temporal lobe regions

We next tested whether theta-phase locking differed between medial temporal lobe regions, including the amygdala, entorhinal cortex, hippocampus, parahippocampal cortex, and temporal pole. We found that the percentage of cells with significant phase locking varied between the different brain regions ($\chi^2$ tests: baseline, $\chi^2(4) = 15.327$, $P_{corr.} = 0.012$; encoding, $\chi^2(4) = 16.198$, $P_{corr.} = 0.008$; recall, $\chi^2(4) = 13.729$, $P_{corr.} = 0.025$; $n = 666$; Bonferroni corrected for three tests; Fig. 4A). Numerically, the percentage of phase-locking neurons was lowest in the hippocampus and highest in the parahippocampal cortex. To better understand the origin of this finding, we analyzed the units' PPC values and found that they differed between brain regions as well (ANOVA for linear mixed-effects model: fixed effect of region, $F(4) = 21.347$, $P < 0.001$; Fig. 4B). Post-hoc pairwise comparisons showed that the regional differences were driven by increased theta-phase locking in the parahippocampal cortex (Table S2). As lower theta-phase locking is equivalent to a higher variability of spiking-related theta phases that may encode additional information, we speculate that this result points at an

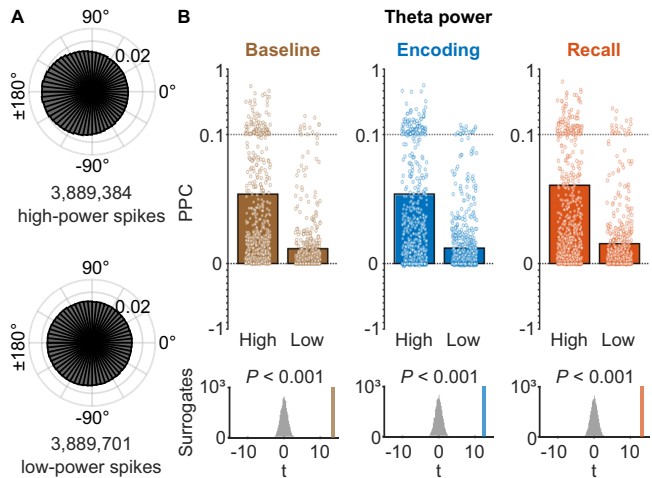

**Fig. 3 | Theta-phase locking as a function of theta power. A** Theta-phase locking across the entire experiment as a function of whether the spikes coincided with high or low theta power (median split), pooled across all units. Theta-phase locking was more strongly expressed during periods with high theta power (one-sided surrogate analysis with spike-label shuffling, $\Delta z = 2.459 \times 10^4$, $P < 0.001$). **B** Average theta-phase locking during high and low theta power across units, quantified with pairwise phase consistency (PPC). Results are shown for baseline (left; entire session except for encoding and recall periods; $n = 666$ neurons), encoding (middle; $n = 666$), and recall (right; $n = 666$). Bars show mean PPC values; dots represent PPC values of individual neurons. Y-axis is expanded between 0 and 0.1 to highlight the range containing most data points. We compared pairwise phase consistency between high and low power using $t$-tests and ranked the empirical $t$-values within distributions of surrogate $t$-values (plots at the bottom; one-sided surrogate analysis: baseline, $t(665) = 13.502$, $P_{\text{corr.}} < 0.001$; encoding, $t(665) = 12.393$, $P_{\text{corr.}} < 0.001$; recall, $t(665) = 13.305$, $P_{\text{corr.}} < 0.001$; Bonferroni corrected for three tests). Source data are provided as a Source Data file.

increased potential for theta-phase coding in the other, non-parahippocampal medial temporal lobe regions (for an illustration of the concepts of phase locking, phase coding, and how they can coexist, see Fig. S12).

We aimed at ruling out that this finding was a side-effect of other properties, such as higher spike counts, which can lower PPC variance and increase the percentage of phase-locking neurons (Fig. S11), or increased theta power, which can increase theta-phase locking (Fig. 3B). We thus performed control analyses to test whether spike counts or theta power varied between medial temporal lobe regions. This showed that spike counts did not differ between medial temporal lobe regions (Kruskal–Wallis tests: baseline, $H(4) = 8.135$, $P_{\text{corr.}} = 0.260$; encoding, $H(4) = 4.308$, $P_{\text{corr.}} = 1$; recall, $H(4) = 10.161$, $P_{\text{corr.}} = 0.113$; Bonferroni corrected for three tests; Fig. 4C). In contrast, spike-associated theta power varied between medial temporal lobe regions (ANOVA for linear mixed-effects model: fixed effect of region, $F(4) = 204.063$, $P < 0.001$; Fig. 4D). The direction of this effect could not explain the regional differences in phase-locking strength, however, as we observed highest theta power in the hippocampus where we observed the lowest number of phase-locking units (Fig. 4A; Table S3). These control analyses suggest that regional differences in theta-phase locking are not simply reducible to differences in spike counts or theta power.

### Theta-phase locking is stronger during periods with high aperiodic slopes

Local field potentials are composed of two components, aperiodic (non-oscillatory) and periodic (oscillatory) activity[60], both of which may reflect and influence different types of neural processing[74]. Both components can also lead to changes in absolute theta power[6,60] (Fig. S13) and may thus underlie our observation that higher theta-

phase locking emerges during periods of elevated theta power (Fig. 3B). We thus aimed at understanding how theta-phase locking was influenced by aperiodic, non-oscillatory and by periodic, oscillatory activity. In brief, we used Spectral Parameterization Resolved in Time (SPRiNT) to characterize aperiodic activity[61], and we used Bycycle to identify periodic theta oscillations[62]. These methods allowed us to perform two independent analyses where we grouped neuronal spikes (1) as a function of varying properties of aperiodic activity and (2) according to whether they occurred in the presence or absence of clear theta oscillations detected by Bycycle (Figs. 5; 6A).

We first asked whether theta-phase locking varied in strength as a function of non-oscillatory, aperiodic activity, which summarizes neural activity that does not need to arise from any regular, rhythmic process[60]. One can fit and parameterize aperiodic activity by means of two major parameters: the aperiodic slope and aperiodic offset of the power spectrum[60]. Here, we focused on aperiodic slopes, which are thought to reflect the balance between neural excitation and inhibition[74]. Specifically, we tested whether theta-phase locking was stronger during periods with higher (i.e., more negative) slopes as compared to those with lower slopes. This analysis was, by design, independent of the question whether theta-phase locking was stronger during periods of clear theta oscillations (see the next section). Accordingly, periods with high versus low aperiodic slopes occurred both during periods with clear theta oscillations and during periods without them (Fig. S14).

To differentiate between high and low aperiodic slopes over time, we utilized SPRiNT that enabled us to parameterize aperiodic activity in a temporally resolved manner[61]. Briefly, we estimated power spectra in short, 3-s time windows every 500 ms over the course of each session (Fig. 5). For each power spectrum, we then calculated the slope of the aperiodic activity in log-log space (i.e., the power law exponent). In preliminary analyses, we tested whether aperiodic slopes differed between the three task periods of encoding, recall, and baseline, and found that they were similar in magnitude (Fig. S15A). In general, steeper slopes were associated with lower frequencies within the 1–10 Hz theta frequency range (Fig. S4, B, C). We also analyzed whether neuronal firing rates were modulated by aperiodic slopes and found that periods with high aperiodic slopes were associated with decreased neuronal firing rates (two-sided surrogate analysis with condition-label swapping, $t(665) = -8.337$, $P < 0.001$; Fig. 6B; Fig. S10), in both putative interneurons and pyramidal cells (Fig. S9). This observation is in line with the idea that higher (i.e., more negative) aperiodic slopes are related to increased inhibition[74] and thus correlate with a suppression of neuronal activity.

We then examined the impact of aperiodic slopes on neuronal theta-phase locking. We found that theta-phase locking was stronger for neuronal spikes associated with high aperiodic slopes as compared to those associated with low aperiodic slopes (median split). This effect was consistent across baseline, encoding, and recall periods (one-sided surrogate analysis with condition-label swapping: baseline, $t(665) = 7.464$, $P_{\text{corr.}} < 0.001$; encoding, $t(664) = 2.486$, $P_{\text{corr.}} = 0.017$; recall, $t(665) = 3.697$, $P_{\text{corr.}} < 0.001$; Bonferroni corrected for three tests; Fig. 6C; Fig. S10). Different firing rates during high versus low aperiodic slopes (see above) could not explain this result as the pairwise phase consistency is independent of spike count. This finding indicates that the aperiodic slope of the local field potential is a major determinant of the strength of theta-phase locking.

### Phase locking is stronger during periods with clear theta oscillations

So far, we used all data to investigate theta-phase locking without excluding periods that did not show clear theta oscillations that were unambiguously detected by Bycycle. This approach was driven by previous studies[38] and because we wanted to approach possible

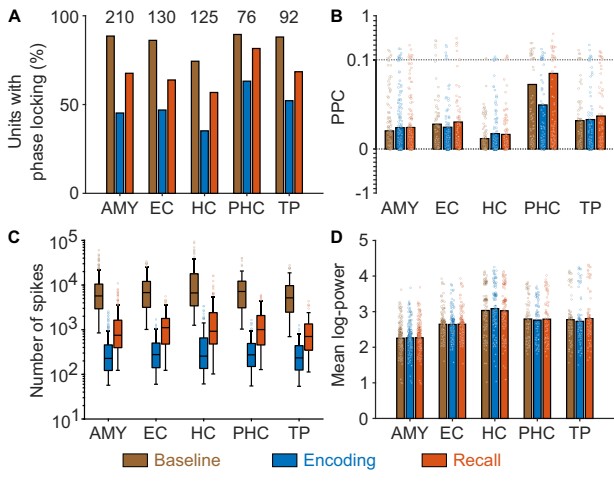

**Fig. 4 | Theta-phase locking, spike numbers, and theta power across brain regions. A–D** Theta-phase locking, number of spikes, and theta power in different brain regions during baseline (brown), encoding (blue), and recall (orange). Analyses in (**A–D**) were performed with 210 neurons from the amygdala (AMY), 130 from the entorhinal cortex (EC), 125 from the hippocampus (HC), 76 from the parahippocampal cortex (PHC), and 92 from the temporal pole (TP). **A** Percentages of units with significant phase locking. Numbers at top indicate the total number of neurons per region. The percentage of cells with significant phase locking varied between the different brain regions ($\chi^2$ tests: baseline, $\chi^2(4) = 15.327$, $P_{corr.} = 0.012$; encoding, $\chi^2(4) = 16.198$, $P_{corr.} = 0.008$; recall, $\chi^2(4) = 13.729$, $P_{corr.} = 0.025$; n = 666; Bonferroni corrected for three tests). **B** Pairwise phase consistency (PPC) across units. Bars show mean PPC values; dots represent PPC values of individual neurons. Y-axis is expanded between 0 and 0.1 to highlight the range containing most data points. PPC values differed between brain regions (ANOVA for linear mixed-effects model: fixed effect of region, $F(4) = 21.347$, $P < 0.001$). **C** Number of spikes per region, across neurons. Box plots show medians, 25th and 75th percentiles, and outliers as dots. Spike counts did not differ between medial temporal lobe regions (Kruskal–Wallis tests: baseline, $H(4) = 8.135$, $P_{corr.} = 0.260$; encoding, $H(4) = 4.308$, $P_{corr.} = 1$; recall, $H(4) = 10.161$, $P_{corr.} = 0.113$; Bonferroni corrected for three tests). **D** Log-transformed power assigned to the spikes of each neuron. Bars show mean values; dots represent values of individual neurons. Spike-associated theta power varied between medial temporal lobe regions (ANOVA for linear mixed-effects model: fixed effect of region, $F(4) = 204.063$, $P < 0.001$). Source data are provided as a Source Data file.

determinants of theta-phase locking as broadly as possible. We next asked how the presence of theta oscillations in the local field potentials modulated theta-phase locking and hypothesized that theta-phase locking would be stronger during periods with clear theta oscillations as compared to periods without them.

To identify theta oscillations, we isolated individual cycles in the 1–10 Hz filtered signal and used Bycycle to categorize each as being part of an oscillation or not (Fig. 5A). We note that this practical separation of the local field potential into periods with and without clear theta oscillations is non-trivial and not binary, as it reflects a trade-off between specificity and sensitivity determined by the parameters of the chosen oscillation detection algorithm. Here, we implemented this distinction to enable a relative comparison of theta-phase locking between periods with stronger (suprathreshold) versus weaker (subthreshold) theta oscillations. This does not exclude the presence of weaker theta oscillations during periods without clear theta oscillations, which are presumably responsible for residual theta-phase locking in those time periods.

In a preliminary step, we characterized the presence of theta oscillations in our recordings. For all 502 microwires, we computed the percentage of time windows with theta oscillations and found that theta oscillations were present about 38% of the time (Fig. 6D; Fig. S15B). To understand whether theta oscillations occurred at a preferred theta frequency, we computed the mode frequency of all

detected oscillations across wires (545,470 oscillations in total) and observed that they occurred most often at 3–5 Hz (Fig. 6E, F). Comparable frequency distributions of theta oscillations have been observed in prior studies in which participants performed similar virtual navigation tasks[16,33,40]. Performing this analysis separately for microwires from different brain regions revealed distinct region-specific distributions of theta frequencies (Fig. S16). We furthermore observed that firing rates during clear theta oscillations were higher than during time periods without them (two-sided surrogate analysis with condition-label swapping, $t(665) = 2.598$, $P = 0.007$; Fig. 6B; Fig. S10), which resembles previous observations in humans[38] and may relate to the finding of increased firing rates during periods of increased theta power in the membrane potential of CA1 pyramidal neurons in mice[75].

We then analyzed how the presence of clear theta oscillations modulated phase locking during encoding, retrieval, and baseline periods. During all three periods, we found that phase locking was stronger for neuronal spikes occurring during clear theta oscillations as compared to those outside clear theta oscillations (one-sided surrogate analysis with condition-label swapping: baseline, $t(665) = 9.163$, $P_{corr.} < 0.001$; encoding, $t(665) = 6.490$, $P_{corr.} < 0.001$; recall, $t(665) = 8.790$, $P_{corr.} < 0.001$; Bonferroni corrected for three tests; Fig. 6G; Fig. S10). We obtained similar results when using a 3–10 Hz frequency range for estimating generalized phase (Fig. S6). Residual theta-phase locking during periods without clear oscillations suggests that some oscillatory activity may have been missed by the oscillation-detection algorithm. In support of this interpretation, control simulations showed that intrinsic theta resonance of individual neurons does not lead to theta-phase locking if the local field potential only consists of pink noise and does not contain any oscillatory activity (Fig. S17). Taken together, human theta-phase locking occurs more strongly during clear theta oscillations, but it is also present during time periods where it is more difficult to detect clear oscillations.

## Human single neurons theta-phase lock during memory encoding and retrieval

We were curious whether neuronal theta-phase locking varied as a function of memory state and performance. We thus investigated theta-phase locking separately for memory encoding and retrieval. Across all 666 neurons, we found 306 neurons that exhibited theta-phase locking during encoding and 444 neurons with theta-phase locking during retrieval (binomial tests versus 5% chance, both $P_{corr.} < 0.001$, Bonferroni corrected for two tests). 273 units exhibited theta-phase locking during both encoding and retrieval (binomial test, $P < 0.001$; Fig. 7A). When only considering encoding and retrieval segments with successful memory performance, we found 163 neurons that exhibited phase locking during both encoding and retrieval, and 190 such neurons when only using segments with unsuccessful performance (binomial tests, both $P_{corr.} < 0.001$, Bonferroni corrected). These results show that theta-phase locking is prevalent during both encoding and retrieval, and that the number of phase-locking neurons is similar during periods of successful and unsuccessful memory performance.

We next tested whether the strength of theta-phase locking was higher during segments with successful versus unsuccessful memory performance, separately for encoding and retrieval. We did not find stronger theta-phase locking (i.e., higher PPC) during successful encoding, indicating that the neurons' preference toward a particular theta phase during encoding was independent of whether encoding was successful or not (one-sided surrogate analysis with condition-label swapping, $t(665) = -1.030$, $P_{corr.} = 1$, Bonferroni corrected for two tests; Fig. 7B, C; Figs. S7, S8, S10). To corroborate this observation, we calculated the spike-field coherence in a frequency-resolved way between 1 and 100 Hz[39]. We did not find any frequency with stronger

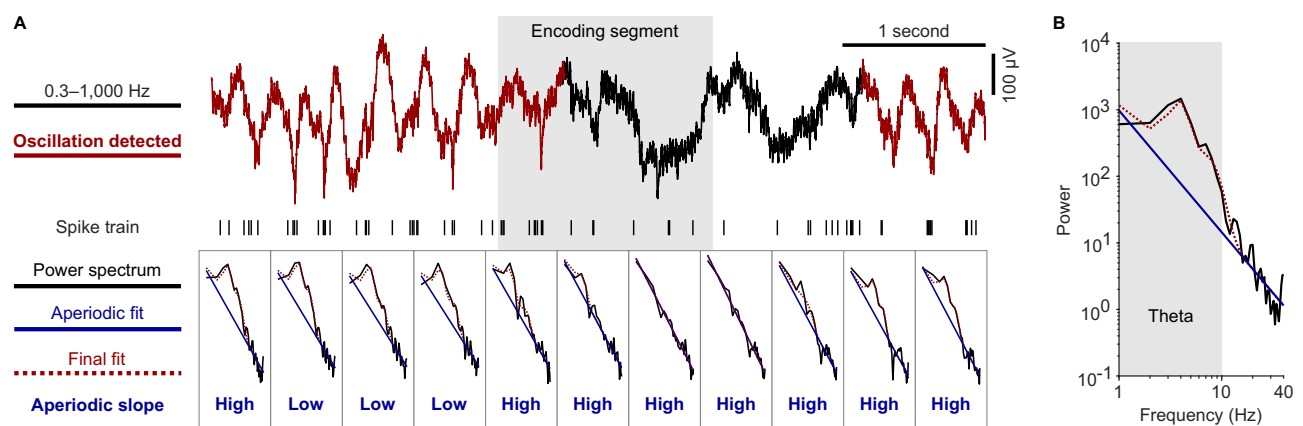

**Fig. 5 | Spectral parameterization resolved in time and detection of theta oscillations. A** We utilized Spectral Parameterization Resolved in Time (SPRiNT[61]) to analyze power spectra of the local field potentials in time steps of 500 ms. For each power spectrum (1–40 Hz; black), SPRiNT provided a fit to its aperiodic component (blue line) and a final fit to the original power spectrum (dotted red line). We grouped time windows into those with high versus low aperiodic slopes (median split). We also grouped them into time windows with versus without clear theta oscillations (red and black, respectively) based on whether Bycycle detected an oscillation using the features of individual cycles[62]. **B** Example power spectrum to illustrate the aperiodic and final fit. Black line, power spectrum; blue line, aperiodic fit; dotted red line, final fit; gray area, theta-frequency range. Source data are provided as a Source Data file.

phase locking during successful versus unsuccessful memory encoding (all $P_{corr.} = 1$, Bonferroni corrected for 100 frequencies and two tests; Fig. 7D). This effect was similar across different medial temporal lobe regions (Fig. S18).

When investigating recall periods, we similarly found that neuronal theta-phase locking did not differ between segments with successful versus unsuccessful memory performance (one-sided surrogate analysis with condition-label swapping, $t(665) = 0.576$, $P_{corr.} = 0.562$, Bonferroni corrected for two tests; Fig. 7E, F; Figs. S7, S8, S10). A more general spike-field coherence analysis, as described above, did not reveal any frequencies between 1–100 Hz with stronger phase locking during successful versus unsuccessful retrieval (all $P_{corr.} = 1$, Bonferroni corrected; Fig. 7G). In contrast to previous studies with recognition and verbal free recall memory tasks that observed stronger phase locking during successful memory encoding[39,54], these results suggest that the strength of theta-phase locking during memory encoding and retrieval did not differ as a function of memory success in our spatial memory task.

We additionally tested whether the neurons' preferred theta phases changed between successful and unsuccessful memory. This analysis showed no significant differences in preferred theta phases between segments with successful versus unsuccessful memory performance (surrogate analysis with spike-label shuffling; encoding: $F(665) = 0.652$, $P_{corr.} = 0.744$; retrieval: $F(665) = 1.352$, $P_{corr.} = 0.167$, Bonferroni corrected for two tests; Fig. 7C, F; Fig. S10).

To provide a more comprehensive assessment, we then asked whether the relationship between phase locking and memory performance might be modulated by theta power, aperiodic activity, or the presence of theta oscillations. We also distinguished between units that responded with a firing-rate increase during encoding (object-responsive units; Fig. S19) and units that did not show such object responsiveness. We separately examined units with and without significant phase locking, multi-units and single units, and unit activity during object recall or location recall. In all these cases, we found that neuronal theta-phase locking was similar between successful and unsuccessful segments, both for encoding and retrieval (one-sided surrogate analysis with condition-label swapping; all $P_{corr.} \geq 0.073$, Bonferroni corrected for performing this analysis for encoding and retrieval and for two data conditions in each case; Fig. 8A, B). Additional control analyses showed that time-frequency resolved power, which can modulate theta-phase locking effects, did not differ as a function of memory performance during encoding and retrieval

(Fig. S20). In summary, our findings suggest that theta-phase locking strength and the preferred theta phases of single neurons during encoding and recall did not depend on whether memory retrieval was successful or not.

## Some neurons shift their preferred theta phases between encoding and retrieval

We considered that neurons might exhibit theta-phase locking to different theta phases during encoding versus retrieval (Fig. 9). Such theta-phase shifts have been predicted by the Separate Phases of Encoding And Retrieval (SPEAR) model and may help avoid interference between encoding and retrieval processes[52–54,76]. We thus tested for neurons that exhibited a significant phase difference between encoding and retrieval (Fig. 9A) and found evidence for such phase shifts in some neurons of our dataset (62 of 666 neurons; 9%; binomial test versus 5% chance, $P < 0.001$; Fig. 9B, C, F; Fig. S21A). When only considering neurons that showed significant phase locking during both encoding and retrieval, we found a similar proportion of neurons with significant phase shifts between encoding and retrieval (26 of 273; 10%; binomial test, $P = 0.001$; Fig. 9C, F).

We then tested whether theta-phase shifts depended on memory performance. When only considering data from successful encoding and retrieval segments, we observed a significant number of neurons with significant phase shifts between encoding and retrieval (54 phase-shifting neurons of 666 neurons; 8%; binomial test, $P_{corr.} < 0.001$, Bonferroni corrected for performing the test for successful and unsuccessful memory performance; Fig. 9D, F; Fig. S21B). We observed a similar trend when only considering units with significant phase locking during both encoding and retrieval (14 phase-shifting neurons of 163 neurons; 9%; binomial test, $P_{corr.} = 0.070$, Bonferroni corrected; Fig. 9D, F). When only considering data from unsuccessful segments, the number of significantly shifting units was neither significant for all units (43 of 666; 6%; binomial test, $P_{corr.} = 0.110$, Bonferroni corrected; Fig. 9E, F; Fig. S21C) nor for units with significant phase locking during encoding and retrieval (12 of 190; 6%; binomial test, $P_{corr.} = 0.488$, Bonferroni corrected; Fig. 9E, F). These results suggest that phase shifts were slightly more prevalent during successful as compared to unsuccessful memory, though the direct comparisons were not significant (one-sided surrogate analysis with condition-label shuffling: all units, $\Delta = 2\%$, $P = 0.136$; phase-locking units, $\Delta = 2\%$, $P = 0.276$).

As we noticed that the shifts varied in magnitude and direction, we next quantified the phase differences of phase-shifting units

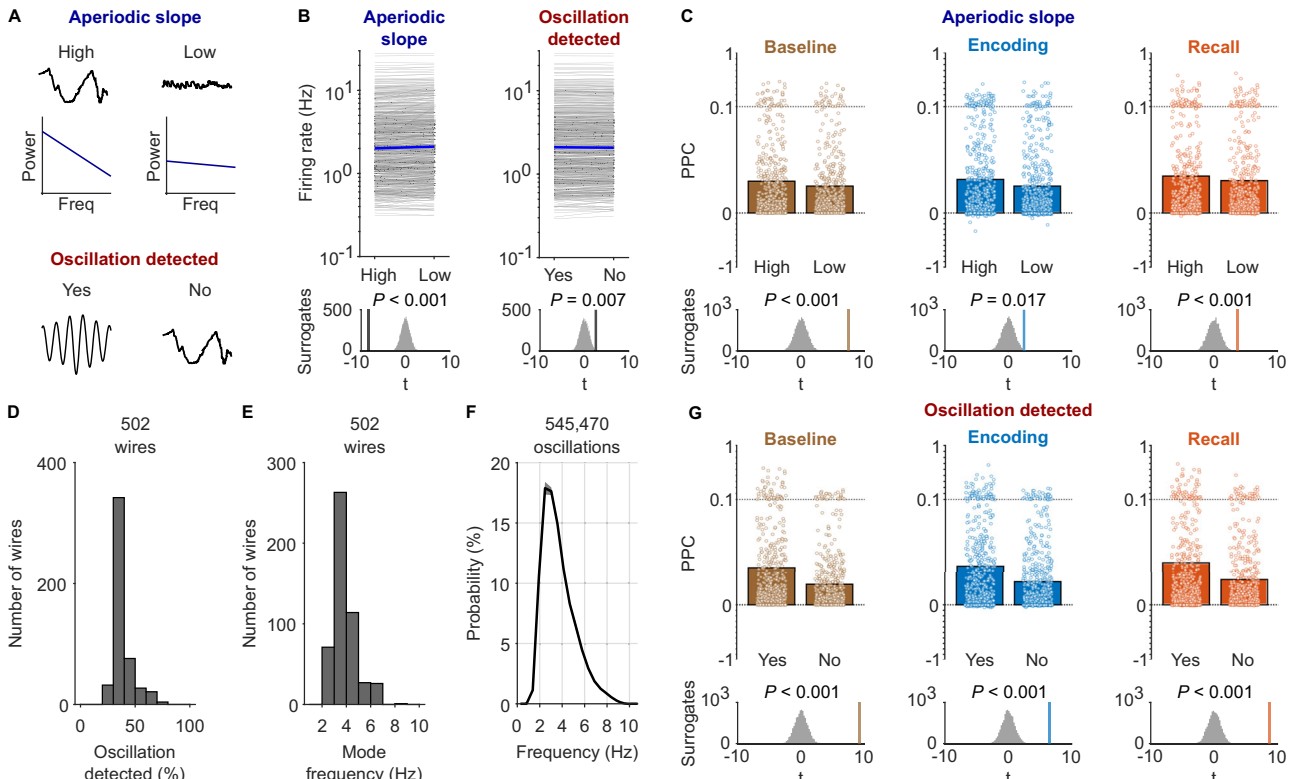

**Fig. 6 | Theta-phase locking as a function of aperiodic slope and the presence of theta oscillations. A** Schematic illustrating the independent analyses of theta-phase locking as a function of aperiodic slope and clear theta oscillations detected by Bycycle. Top: high aperiodic slopes result from increased activity in lower frequencies and decreased activity in higher frequencies. These increases and decreases need not result from rhythmic processes. Bottom: oscillations are detected if consecutive cycles have similar amplitude and frequency. **B** Firing rate as a function of aperiodic slope and the presence of clear theta oscillations. Each line, one unit (*n* = 666). Blue lines connect median firing rates. Bottom: Empirical *t*-values of paired *t*-tests comparing firing rates between conditions (vertical lines), along with distributions of surrogate *t*-values (two-sided surrogate analysis: slope, *t*(665) = −8.337, *P* < 0.001; oscillations, *t*(665) = 2.598, *P* = 0.007). **C** Theta-phase locking during high and low aperiodic slopes for baseline (*n* = 666 neurons), encoding (*n* = 665), and recall (*n* = 666). Empirical *t*-values of paired *t*-tests were ranked in surrogate distributions of *t*-values (bottom; one-sided surrogate analysis:

baseline, *t*(665) = 7.464, $P_{corr.}$ < 0.001; encoding, *t*(664) = 2.486, $P_{corr.}$ = 0.017; recall, *t*(665) = 3.697, $P_{corr.}$ < 0.001; Bonferroni corrected). Bars, mean pairwise phase consistency (PPC); dots, PPC of individual neurons. Y-axis is expanded between 0 and 0.1. **D** Histogram of percentages of time with theta oscillations. **E** Histogram of mode frequencies of oscillations in the theta-frequency range. **F** Probability distribution of frequencies of detected theta oscillations. Shaded area, standard error of the mean across microwires. **G** Theta-phase locking during the presence versus absence of clear theta oscillations, separately for baseline (*n* = 666 neurons), encoding (*n* = 666), and recall (*n* = 666). Bars, mean PPC; dots, PPC of individual neurons. Y-axis is expanded between 0 and 0.1. Empirical *t*-values were ranked in surrogate distributions to assess significance (bottom; one-sided surrogate analysis: baseline, *t*(665) = 9.163, $P_{corr.}$ < 0.001; encoding, *t*(665) = 6.490, $P_{corr.}$ < 0.001; recall, *t*(665) = 8.790, $P_{corr.}$ < 0.001; Bonferroni corrected). Source data are provided as a Source Data file.

between encoding and retrieval. We found that the absolute phase difference was 85° ± 47° on average (mean ± circular SD). Among the neurons with significant phase locking during both encoding and retrieval, the absolute phase difference between encoding and retrieval was 36° ± 22° (mean ± SD; Fig. 9C). These effects were similar for successful and unsuccessful memory performance (successful, all neurons: 101° ± 45°; successful, phase-locking neurons: 42° ± 18°; unsuccessful, all: 109° ± 45°; unsuccessful, phase-locking neurons: 47° ± 32°; Fig. 9D, E). These results show that the extent of phase shifts between encoding and retrieval in our data were lower than the originally proposed shifts between oscillatory peaks and troughs[52], though the observed shifts may still be sufficiently large to help separate encoding and retrieval processes.

We finally asked whether theta-phase shifts varied as a function of other properties of the local field potential. We thus estimated the number of units with significant phase shifts separately for spikes associated with high and low theta power, high and low aperiodic slope, during versus outside clear theta oscillations, and for those occurring at higher versus lower theta frequencies (Fig. S21D–G). We found a significant number of phase-shifting units for high theta power (64 of 666; 10%; binomial test, $P_{corr.}$ < 0.001, Bonferroni corrected for

two tests) but not for low theta power (37 of 666; 6%; binomial test, $P_{corr.}$ = 0.557, Bonferroni corrected; Fig. S21D). The number of phase-shifting neurons was also significant for spikes associated with high aperiodic slopes (52 of 666; 7.8%; binomial test, $P_{corr.}$ = 0.002, Bonferroni corrected) but not for those with low aperiodic slopes (41 of 666; 6.2%; binomial test, $P_{corr.}$ = 0.206, Bonferroni corrected; Fig. S21E). The number of phase-shifting neurons was significant for spikes in the presence of clear theta oscillations (47 of 666; 7.1%; binomial test, $P_{corr.}$ = 0.025, Bonferroni corrected) and in their absence, suggesting the presence of subthreshold theta oscillations sufficient to elicit theta-phase shifts (50 of 666; 7.5%; binomial test, $P_{corr.}$ = 0.007, Bonferroni corrected; Fig. S21F). The number of phase-shifting neurons was significant for spikes occurring at both higher and lower frequencies within the 1–10 Hz band (both 56 of 666; 8.4%; binomial tests, $P_{corr.}$ < 0.001, Bonferroni corrected; Fig. S21G). When performing the analysis of theta-phase shifts separately for low theta (2–5 Hz) and high theta (6–9 Hz) using the filter-Hilbert method, we again observed similar effects that were slightly stronger for low theta (Figs. S7, S8).

These results indicate that shifts in the preferred theta phase are—at least to some extent—modulated by electrophysiological

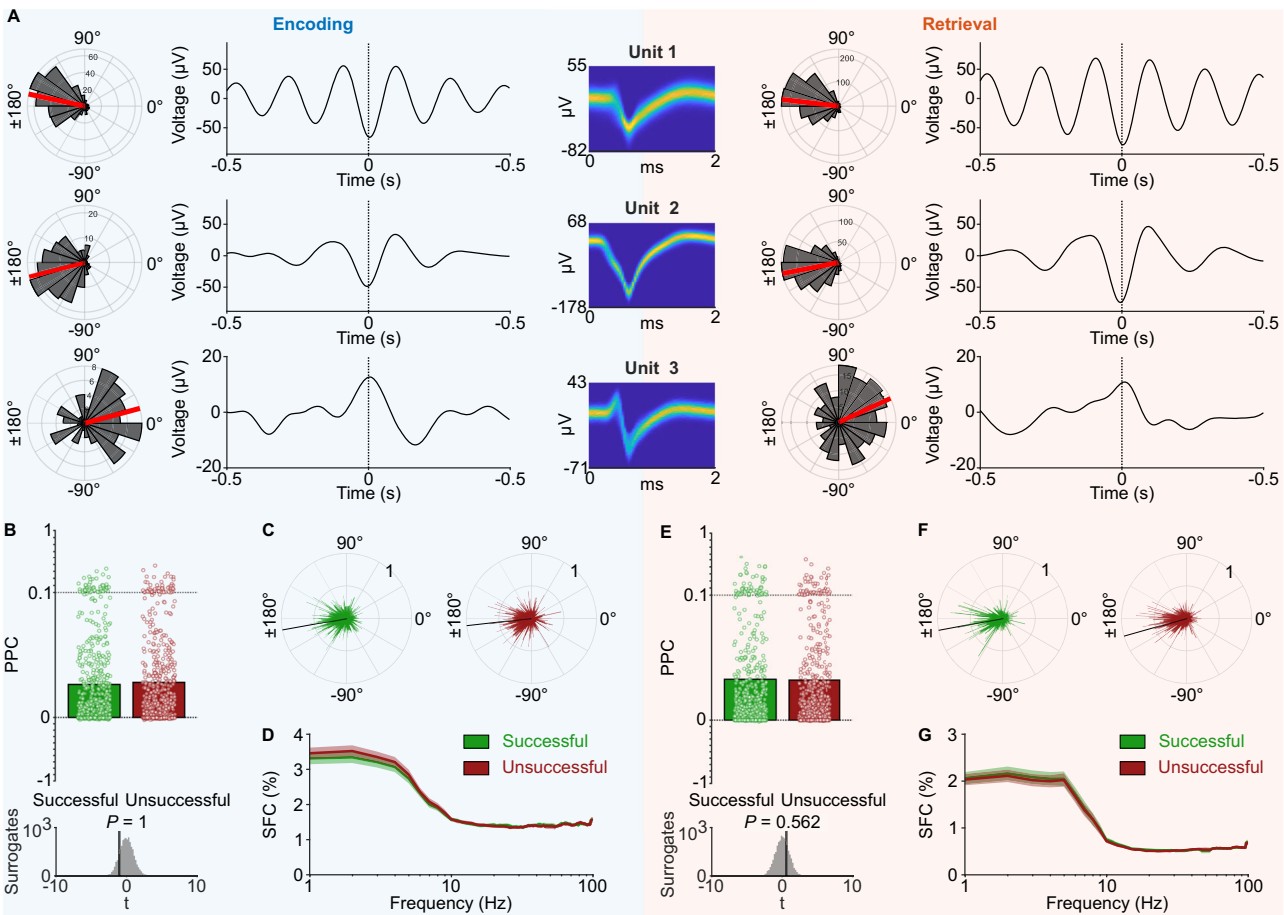

**Fig. 7 | Theta-phase locking of human single neurons during memory encoding and retrieval and as a function of memory performance. A** Examples of neuronal theta-phase locking during encoding and retrieval. Polar histograms show the theta phases of the units' spikes during encoding (left) and retrieval (right). Red lines, mean phase angles. 1–10 Hz bandpass-filtered spike-triggered local field potentials are shown for encoding (left) and retrieval (right). Spike waveforms are shown as density plots. **B** Theta-phase locking during successful (green) versus unsuccessful (red) encoding ($n = 666$ neurons). Bars show mean pairwise phase consistency (PPC) values; dots represent PPC values of individual neurons. Y-axis is expanded between 0 and 0.1 to highlight the range containing most data points. We compared PPC values between successful and unsuccessful encoding using a $t$-test and ranked the empirical $t$-value within surrogate $t$-values (bottom; one-sided surrogate analysis: $t(665) = -1.030$, $P_{corr.} = 1$, Bonferroni corrected for two tests). **C** Summary plots across all units showing their preferred theta phases and mean resultant vector lengths during successful (left) and unsuccessful (right) encoding. Black

lines, mean phase across units. **D** Spike-field coherence for successful and unsuccessful encoding. Shaded areas, mean ± standard error of the mean across units. **E** Theta-phase locking during successful (green) versus unsuccessful (red) retrieval ($n = 666$ neurons). Bars show mean PPC values; dots represent PPC values of individual neurons. Y-axis is expanded between 0 and 0.1 to highlight the range containing most data points. We compared PPC values between successful and unsuccessful retrieval using a $t$-test and ranked the empirical $t$-value within surrogate $t$-values (bottom; one-sided surrogate analysis: $t(665) = 0.576$, $P_{corr.} = 1$, Bonferroni corrected for two tests). $P$-value is Bonferroni corrected for testing both encoding and retrieval. **F** Summary plots across all units showing their preferred theta phases and mean resultant vector lengths during successful (left) and unsuccessful (right) retrieval. Black line, mean phase across units. **G** Spike-field coherence for successful and unsuccessful retrieval. Shaded areas, mean ± standard error of the mean across units. SFC spike-field coherence. Source data are provided as a Source Data file.

properties including theta power and aperiodic slope. Overall, our findings of significant phase shifts between encoding and retrieval provide additional, albeit limited, evidence for the SPEAR model[52,54,76,77]. As the majority of neurons showed similar theta phases between encoding and retrieval, however, we suggest that both theta-phase shifts and stable theta phases may contribute to encoding–retrieval processes.

## Discussion

In this study, we aimed at improving our basic understanding of neuronal theta-phase locking in humans. Our results demonstrate strong phase locking of single neurons to the local theta rhythm in different regions of the human medial temporal lobe. Several electrophysiological properties of the local field potentials including theta power, the presence of clear theta oscillations, and the slope of aperiodic activity modulated the strength of theta-phase locking. This

indicates that theta-phase locking is a dynamic phenomenon that is influenced by a variety of properties of the local field potential. Furthermore, theta-phase locking was similarly pronounced during periods with successful versus unsuccessful memory performance, suggesting that additional studies are needed to pinpoint the memory relevance of theta-phase locking. While most neurons preferred similar theta phases between different memory states, some neurons showed significant shifts in their preferred theta phases between encoding and retrieval, roughly in line with the idea that different memory processes happen at distinct parts of the theta cycle[52]. Together, these results help identify the electrophysiological determinants and functional properties of theta-phase locking in humans. This contributes to a better understanding of the relationship between single neurons and neuronal populations in the human brain and may lead to further investigations into the relevance of theta-phase locking in human memory and information processing.

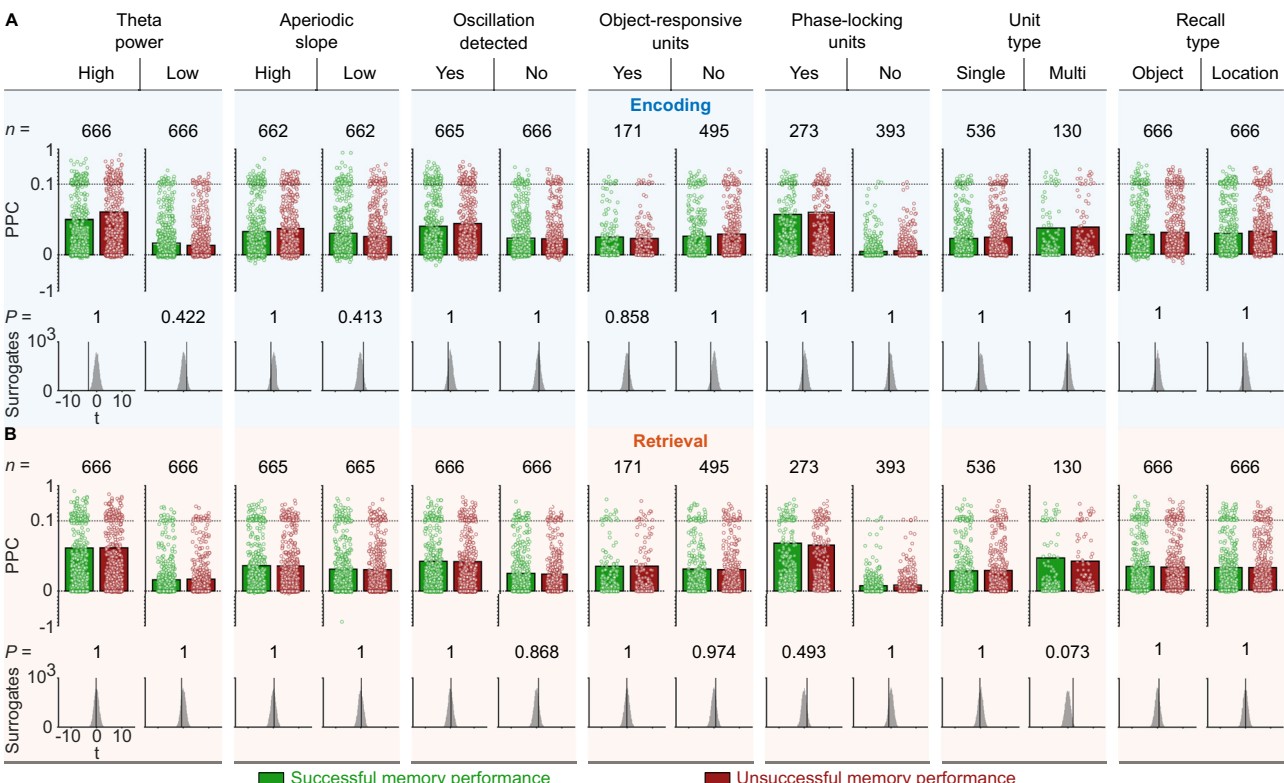

**Fig. 8 | Theta-phase locking as a function of memory performance for different data subsets. A** Theta-phase locking during successful (green) versus unsuccessful (red) encoding for different data subsets. No significant differences were observed. Recall type refers to object-cued recall and location-cued recall, where the encoding process is the same, but the recall processes differ. See Fig. 7B for comparison with the full dataset results. Bars show mean PPC values; dots represent PPC values of individual neurons. Y-axis is expanded between 0 and 0.1 to highlight the range containing most data points. The number of neurons per condition is indicated at the top. We compared PPC values between successful and unsuccessful encoding using a *t*-test and ranked the empirical *t*-value within surrogate *t*-values (bottom; one-sided surrogate analysis, *P*-values are Bonferroni

corrected for performing this analysis for encoding and retrieval and for two data conditions in each case. **B** Theta-phase locking during successful (green) versus unsuccessful (red) retrieval for different data subsets. No significant differences were observed. Bars show mean PPC values; dots represent PPC values of individual neurons. Y-axis is expanded between 0 and 0.1 to highlight the range containing most data points. See Fig. 7E for comparison with the full dataset results. The number of neurons per condition is indicated at the top. We compared PPC values between successful and unsuccessful retrieval using a *t*-test and ranked the empirical *t*-value within surrogate *t*-values (bottom; one-sided surrogate analysis, *P*-values are Bonferroni corrected for performing this analysis for four tests). Source data are provided as a Source Data file.

## Determinants of theta-phase locking

When we estimated theta-phase locking across the entire task, we found a high number of neurons that exhibited theta-phase locking (about 86%; Fig. 4A). This aligns with studies in rodents reporting similarly high numbers of hippocampal neurons phase-locking to the local theta rhythm[36,78]. Confining the analysis to the shorter periods of encoding and retrieval clearly reduced the number of theta-phase locking neurons. This indicates that the amount of data (and, thus, statistical power) is a major factor in determining the number of phase-locking units.

To compute theta phases, we used a generalized phase approach[59] including broad-band filtering between 1 and 10 Hz. This is different from previous phase-locking studies that estimated theta phases at various individual theta frequencies and computed phase locking for each of these frequencies separately[38,39,54]. We opted for the generalized phase approach as visual inspection of our local field potentials showed that—in contrast to rodent theta oscillations—they were not stable and sustained narrowband oscillations, but often dynamic in frequency and amplitude over time. We observed that the generalized phase estimation fit the raw data best and led to fewer phase distortions. A prior study in monkeys showed that neuronal spiking is coupled more strongly to generalized phase than to narrowband theta or alpha phases[59]. The use of generalized phase may thus constitute another reason for the high number of neurons with theta-phase locking in our study.

Single-neuron spiking occurred most often at the trough of the theta rhythm, similar to previous observations in humans[38]. This effect was present both at the level of single neurons and when we pooled the spike phases of all neurons. This finding is in line with the view that theta peaks are associated with increased inhibition that prevents single neurons from spiking[1,7,74], imposing a major restriction on when action potentials can occur during the theta cycle. In this way, theta phases may gate and coordinate the output of single neurons[19,77,79–82].

We found that different signal properties of the local field potential were associated with theta-phase locking. First, we observed that theta-phase locking was more strongly expressed during periods with high theta power, again in line with prior findings[38]. Higher power may thus reflect an additional layer of inhibition during theta peaks, further restricting the occurrence of single-neuron activity to parts of the theta cycle close to its trough. As it is a prominent observation in rodents that theta power increases with running speed[69,83] at which the animal samples sensory information at a higher rate, we speculate that tighter theta-phase locking during periods of increased theta power may be useful for processing rapidly incoming information in a precise and efficient manner.

Furthermore, using spectral parameterization in a time-resolved manner[60,61] and a cycle-by-cycle algorithm for detecting oscillations[62], we found stronger phase locking when aperiodic activity showed steeper (i.e., more negative) slopes and when clear theta oscillations were detected in the local field

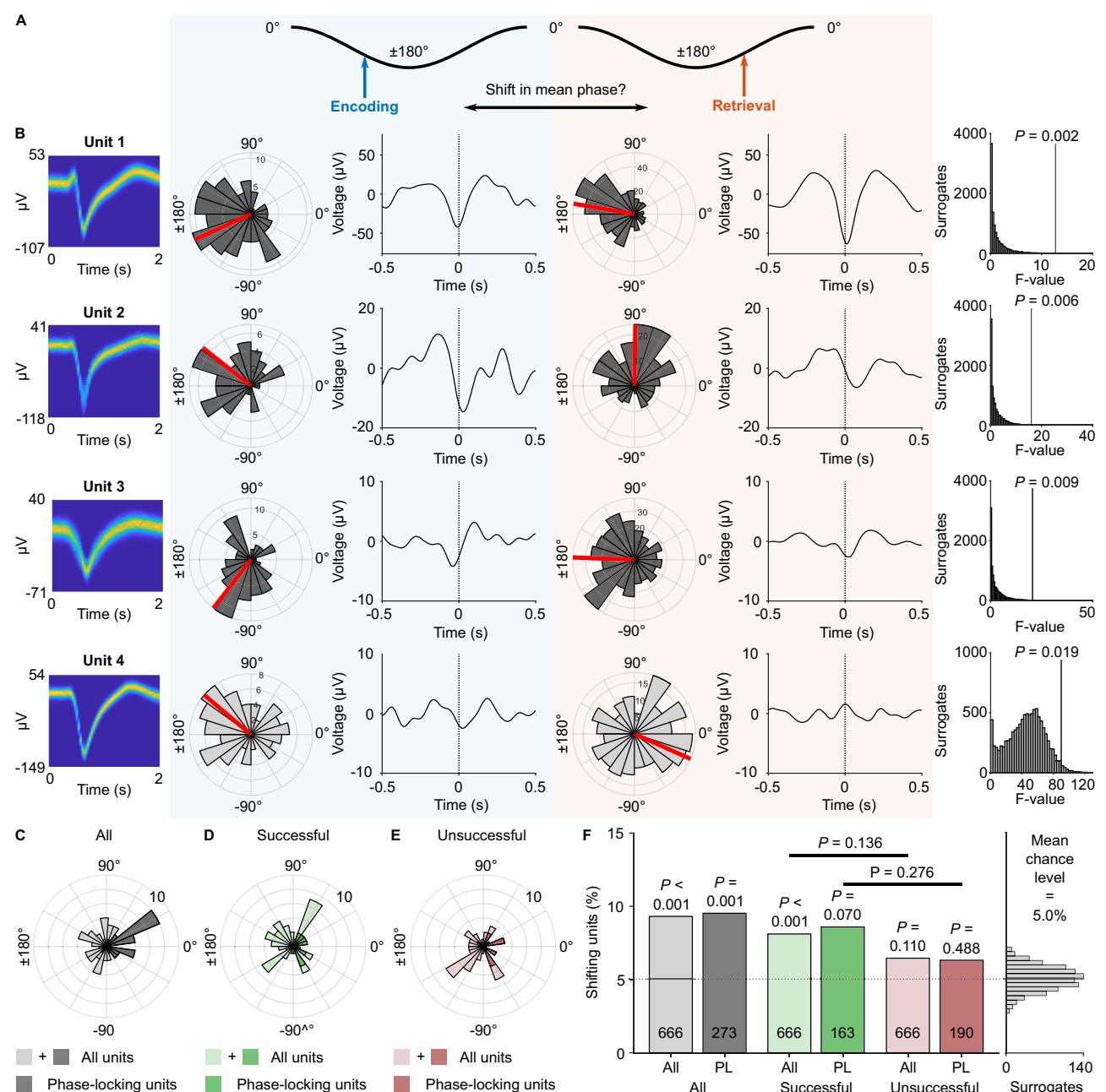

**Fig. 9 | Shifts of preferred theta phases between encoding and retrieval.**
**A** Hypothesis that neurons shift their preferred theta phases between encoding and retrieval[52]. The illustration shows a hypothetical shift from an earlier phase during encoding toward a later phase during retrieval. **B** Example units with significantly shifted theta phases between encoding and retrieval. Polar histograms show theta-phase distributions of all spikes during encoding (blue) and retrieval (orange). Theta-filtered spike-triggered averages are shown next to histograms. Left, spike-waveforms as density plots. Right, theta-phase shifts were considered significant if the empirical *F*-value (vertical lines) of a Watson–Williams test exceeded the 95th percentile of surrogate *F*-values (histograms). Units 1–3 significantly phase-locked during encoding and retrieval, and unit 4 did not. All examples show a shift from a later phase during encoding toward an earlier phase during retrieval. **C**–**E** Polar histograms show angular differences between theta phases during encoding versus retrieval for neurons with significant theta-phase shifts between encoding and retrieval. Results are shown for all

units and for units with significant theta-phase locking. A positive angular difference corresponds to a shift from an earlier phase during encoding to a later phase during retrieval. Shifts occurred in both directions. See Fig. S21 for plots with the exact encoding and retrieval phases. **C** Results for all segments (*n* = 62 significant units from the pool of all units; *n* = 26 significant units from the pool of significantly phase-locking units). **D** Results for successful segments (*n* = 54 and *n* = 14). **E** Results for unsuccessful segments (*n* = 43 and *n* = 12). An angular difference of 0° corresponds to identical theta phases during encoding and retrieval. **F** Percentages of units with significant theta-phase shifts for all (transparent color) and significantly phase locking units (opaque color). Percentages are compared to 5% chance level using a one-sided binomial test. *P*-values are Bonferroni corrected for testing successful and unsuccessful memory performance. Gray histogram on the right, empirically estimated chance level using surrogate data, confirming the a priori chosen 5% chance level. Source data are provided as a Source Data file.

potential by the algorithm (Figs. 5 and 6). This again fits with the idea that higher levels of inhibition lead to more restricted single-neuron firing—and thus stronger phase-locking—as steeper slopes of aperiodic activity are considered markers of increased

inhibition[74]. Because recent studies have shown that aging and mental disorders affect theta oscillations and characteristics of aperiodic activity[84–86], we propose that neuronal theta-phase locking may change under these conditions as well.

Together, these results provide insights into the electrophysiological and behavioral determinants of theta-phase locking in the human medial temporal lobe (Fig. S22). They suggest that studies investigating theta-phase locking as a function of cognitive processes should consider changes in the basic properties of the local field potential as well. This may help delineate whether changes in theta-phase locking are a more or less independent substrate of cognitive functioning.

## Theta-phase locking during successful versus unsuccessful memory

Previous studies observed stronger theta-phase locking related to successful memory performance in human single-neuron recordings[39,54]. These studies showed stronger theta-phase locking at frequencies of 1–9 Hz during successful memory formation in a recognition memory task with static images[39]. They also showed stronger theta-phase locking in subsequent-memory effect neurons (those neurons with increased firing rates during successful encoding) at frequencies of 2–5 Hz during successful encoding in a verbal free recall task with words[54].

Here, we did not find stronger theta-phase locking during successful memory performance, neither during encoding nor during retrieval. We tested for these memory effects in several subgroups of the data, for example only including spikes in the presence of clear oscillations or only including object-responsive cells, and we did not find differences in theta-phase locking as a function of memory performance in any of these groups (Fig. 8). These findings differ from the above-mentioned studies that described a positive relationship between the strength of theta-phase locking and the successful encoding of images and words[39,54]. This may offer a starting point for reconsidering the relationship between theta-phase locking and human memory, triggering further investigations into the conditions under which theta-phase locking is related to successful memory performance.

We speculate that the dynamic nature of our spatial memory task is relevant to explaining our null effect. Specifically, in contrast to studies with static images or words that have a clear-cut onset and are shown after a blank screen—which may lead to a phase reset of theta oscillations time-locked to stimulus onset[20]—our spatial memory task featured interactive encoding and retrieval periods that were smoothly embedded into the other trial periods. As phase resets have been shown to be associated with successful encoding of static stimuli[20,87], we propose that stronger phase resets of theta oscillations—in combination with a consistent delay of single-neuron activity relative to stimulus onset—may underlie stronger theta-phase locking during successful memory encoding. Future studies may investigate this idea by testing whether theta-phase locking exhibits a memory effect for static but not for dynamic stimuli that can be explained through phase resets and stimulus-locked single-neuron activity.

Beyond theta-phase locking, various other neural processes may be associated with successful memory encoding. For example, theta power in the hippocampus increases during the presentation of items that are later remembered[31,57,88]. Pre-stimulus increases in rhinal and hippocampal theta and alpha power are also linked to successful memory encoding[89]. Coherence between regions may be important, too, as whole-brain connectivity in the theta- and gamma-frequency ranges and, more specifically, rhinal–hippocampal gamma synchronization accompany successful memory formation[90–93]. Future research may investigate the relationship of these diverse electrophysiological phenomena to theta-phase locking and identify their relative importance for successful memory encoding.

## Theta-phase shifts between encoding and retrieval

Single-neuron activity occurring at different theta phases during encoding versus retrieval may be beneficial for memory by preventing interference between the learning of new information and the retrieval of old information[52,53]. This idea has been articulated by the Separate Phases of Encoding And Retrieval (SPEAR) model[52]. Specifically, this model suggests that encoding happens at the theta peak of local field potentials in the hippocampal CA1 region, when input from the entorhinal cortex is strong and input from CA3 is weak. Conversely, the model proposes that retrieval occurs at the theta trough of local field potentials in CA1, when input from the entorhinal cortex is weak and input from CA3 is strong[53].

Guided by this model, we investigated whether neurons in various regions of the human medial temporal lobe showed theta-phase locking at different theta phases during the encoding and retrieval periods of our spatial memory task. While we found that the majority of neurons locked to similar theta phases during encoding and retrieval, we also identified a small but significant number of neurons that shifted their preferred theta phases between encoding and retrieval (Fig. 9). Interestingly, these shifts were numerically slightly more pronounced during successful as compared to unsuccessful memory. Our results thus provide limited additional evidence for the SPEAR model, in line with earlier investigations of theta-phase shifts in rodents and humans[54,76,94,95].

We found that theta-phase shifts varied in magnitude and direction between neurons. The absolute extent of these significant shifts between encoding and retrieval was approximately 90° when considering all phase-shifting neurons. A previous human single-neuron study found theta-phase shifts of comparable magnitude (about 100°) during a verbal free-recall task when investigating neurons whose firing rates showed a subsequent memory effect[54]. Similarly, in rodents, hippocampal CA1 neurons showed theta-phase shifts of less than 90° when rats navigated through novel versus familiar environments that were used to trigger encoding and retrieval states, respectively[76]. Furthermore, rodent CA1 pyramidal cells showed phase shifts of less than 90° when processing novel objects and odors compared to familiar ones[94], and vector trace cells in the subiculum spiked at slightly earlier theta phases (difference of ~40°) when presented with a novel cue[95]. Although these empirically observed shifts are smaller in size than predicted by the SPEAR model, they may still help separate encoding and retrieval processes. We believe that the general phase locking of our neurons to theta troughs is a major reason for why we did not observe strong 180° phase shifts between encoding and retrieval. Accordingly, when only considering phase-shifting neurons that also exhibited significant phase locking during encoding and retrieval, the average phase shift was only about 40° on average. We furthermore observed that neurons shifted their preferred phase in different directions between encoding and retrieval, suggesting that there is no stereotypical assignment of neuronal spikes to earlier or later theta phases during encoding versus retrieval. Future studies are thus required to further clarify the extent and direction of theta-phase shifts, their distribution across different brain regions, and their prevalence during different types of memory.

## Limitations of this study

To examine theta-phase locking as broadly as possible, we analyzed theta-phase locking across the entire task and tested subsequently whether theta-phase locking was more strongly pronounced during periods with clear theta oscillations detected by an oscillation-detection algorithm (Bycycle). We opted for this two-pronged approach as it currently remains an open question whether it is better to focus on periods with clear theta oscillations to increase specificity, or to use all data including subthreshold theta oscillations to increase statistical power. This open question results from the fact that human theta oscillations are variable in strength and frequency, because of which it is not easily possible to obtain a binary classification into periods with theta oscillations versus those without theta oscillations. This situation is complicated by the fact that current

oscillation-detection algorithms depend on a variety of parameters, for which the best thresholds are unknown. Here, we used a cycle-by-cycle approach (Bycycle) with default settings[62] to identify periods with theta oscillations. Visual inspection suggested that Bycycle performed well, but it may have missed periods with subthreshold theta oscillations. We thus believe it is a useful approach to investigate and compare theta-phase locking both in the presence of clear oscillations and across the remaining signal.

In addition to identifying periods with theta oscillations using Bycycle, time-resolved spectral parameterization allowed us to estimate the slope of aperiodic activity over short time windows. This showed that theta-phase locking dynamically changed as a function of aperiodic activity, with stronger theta-phase locking during periods with steeper aperiodic slopes. We note, however, that time-resolved spectral parameterization involves a number of analysis settings due to the complexity of the analysis, and we observed some variability in how well power spectra were fitted depending on particular settings. Specifically, spectral analysis at low frequencies is difficult as these frequencies are close to the lower bound of the power spectrum and the frequency of low-theta oscillations can thus be misidentified. Visual inspection of our data nevertheless showed that our final choice of settings led to a good fit of the original power spectra overall. Computational studies may help to pinpoint the optimal settings for time-resolved spectral parameterization of local field potentials in humans.

In this study, we chose a broad 1–10 Hz frequency range as theta to include both low-theta (1–5 Hz) and high-theta (6–10 Hz) components of the signal. We acknowledge that this may limit the direct comparability of our findings to studies using other definitions of the theta-frequency range such as 3–10 Hz, 4–8 Hz, or 2–10 Hz. As described above, visual inspection of our data suggested that the 1–10 Hz frequency range captured the raw signal well, which is why we opted for this frequency range. We also note that this band is different from the original generalized phase study[59], where a frequency range of 5–40 Hz was chosen to minimize effects related to arousal and to limit the influence of potential low-frequency intrusions that can cause mis-centering in the complex plane, leading to inaccurate phase estimates. Our control analyses did not indicate that 1–3 Hz activity was related to arousal in our awake human participants (in line with previous work[16,31,32,57]), and our analyses and simulations suggested no strong effects of potential low-frequency intrusions. We acknowledge, however, that we cannot fully exclude that some of our phase estimates may have been biased by low-frequency intrusions below the 1–10 Hz frequency-band-of-interest. We also acknowledge that it might be better to use the terms non-oscillatory phase locking or non-rhythmic phase locking to refer to our phase-locking effects relative to the broad 1–10 Hz frequency range, following previous work in bats[68].

Another limitation of this study is the finding that the strength of theta-phase locking was not associated with memory performance. As discussed above, we currently believe that this null effect is due to the dynamic nature of our spatial memory task, which may have prevented theta-phase resets at the onset of the encoding and retrieval periods. Hence, our task design may not have been optimal for detecting differences in theta-phase locking as a function of memory performance, and future studies may utilize static instead of dynamic stimuli to continue investigating relationships between theta-phase locking and memory. Furthermore, our task may not have been difficult enough and the number of encoding and retrieval periods may have been too small (100 each in a full session), because of which the analysis of theta-phase locking as a function of memory performance may have been underpowered. Relatedly, we did not find strong differences in theta-phase shifts between successful versus unsuccessful memory performance, also perhaps because of statistical power. Additional studies are thus needed to better understand the role of stable and shifting theta phases for memory success in humans[6,7,53].

## Methods

### Human participants
We tested $N = 18$ human participants (10 female; age range, 19–61 years; mean age ± SEM, 38 ± 3 years), who were epilepsy patients undergoing treatment for pharmacologically intractable epilepsy at the Freiburg Epilepsy Center, Freiburg im Breisgau, Germany (Table S1). Written informed consent was obtained from all patients. The study conformed to the guidelines of the ethics committee of the University Hospital Freiburg, Freiburg im Breisgau, Germany.

### Spatial memory task
During experimental sessions, patients performed a computerized spatial memory task ("Treasure Hunt"), while seated in a hospital bed. The task was developed with Unity3D (Unity Technologies, San Francisco, CA, USA) and has been used in previous studies[55–57]. The virtual environment of the task comprised a plain beach surrounded by a circular wooden fence. Outside the environment, the background scenery comprised multiple mountains, palm trees, barrels, the sea, and the vast sky. Participants performed up to 40 experimental trials. At the beginning of each trial, participants were placed at a random location on the virtual beach. Participants then actively navigated to successively appearing treasure chests (Fig. 1A), using a game controller. Upon reaching a chest, the chest opened to reveal an object and its name (Fig. 1A). After 1.5 s, the chest and object disappeared. During each navigation–encoding period, participants traveled to 2 or 3 chests. In a complete session, participants traveled to a total of 100 chests. After encountering the last chest of a trial, participants were moved to an overhead perspective of the environment. Participants then played a distractor game (Fig. 1A), in which they had to track the position of a coin hidden under one of three moving boxes. The distractor game had a mean duration of 6.6 s.

After the distractor game, the retrieval period began. In each trial, participants completed either location-cued object recall or object-cued location recall. During object recall, $n + 1$ locations were successively shown on the beach (in random order), where $n$ corresponds to the number of treasure chests encountered during the preceding navigation–encoding period. Participants had four seconds to speak out loud the name of the object that was contained in the treasure chest at the highlighted location or "Nichts" (German for "nothing") for the one location not associated with a treasure chest (Fig. 1B, left). Correctness of the response was evaluated using Cortana (Microsoft, Redmond, WA, USA) and manually checked (and corrected, if necessary) during data analysis. During object-cued location recall, the names of the objects contained in the $n$ treasure chests from the preceding navigation–encoding period were shown to the participants in random order. Participants were asked to move a small target circle across the beach to the remembered location of the associated treasure chest using a game controller (self-paced duration; Fig. 1D, left).

After being probed for all the locations/objects from a given trial, participants completed a recency-judgment task in which they were asked to judge which of two objects they had encountered later during the preceding navigation–encoding period (not analyzed in this study). Finally, participants received feedback on whether they had correctly recalled the object names (Fig. 1B, right) or locations (Fig. 1D, right) and how they performed on the recency-judgment task. The next trial started by transporting the participant back onto the beach.

To quantify the participants' memory performance, we calculated two different metrics: object recall was evaluated based on whether a location-cued object was correctly recalled or not. Location-recall performance was quantified by calculating the Euclidean distance between the remembered location and the correct location of the cued object (drop error). Following previous studies[55,57,96], drop errors were ranked within one million potential drop errors to obtain normalized performance values. A value of 1 indicated the best possible response,

while a value of 0 indicated the worst. We defined a location recall as successful, if its normalized performance value was higher than the session's median normalized performance value.

We defined the electrophysiological data during an encoding or retrieval event as an encoding segment and as a retrieval segment, respectively. An encoding–retrieval pair was classified as successful if the participant correctly recalled the encoded item or its location during the retrieval phase. Any data from the paradigm that was not part of the encoding or retrieval segments were classified as baseline segments.

## Neurophysiological recordings

Patients underwent surgical implantation of intracranial depth electrodes in the medial temporal lobe to identify the epileptic seizure focus for potential subsequent surgical resection. Electrode numbers and locations varied across patients and were solely determined by clinical needs. Neuronal signals were recorded using Behnke–Fried depth electrodes (Ad-Tech Medical Instrument Corp., Oak Creek, WI, USA). Each depth electrode contained a bundle of nine platinum–iridium microelectrodes with a diameter of 40 μm that protruded from the tip of the electrode[63]. We recorded spikes and local field potentials from eight microelectrodes, with the ninth microelectrode serving as the reference. Microelectrode coverage included amygdala, entorhinal cortex, fusiform gyrus, hippocampus, insula, parahippocampal cortex, temporal pole, and visual cortex. We recorded the microelectrode data at 30 kHz using NeuroPort (Blackrock Microsystems, Salt Lake City, UT, USA). We aligned neurophysiological recordings and behavioral data using triggers that were sent from the paradigm to the recording system.

## Spike detection and sorting

Neuronal spikes were detected and sorted using Wave_Clus 3[97]. We used default settings with the following exceptions: "template_sdnum" was set to 1.5 to assign unsorted spikes to clusters in a more conservative manner; "min_clus" was set to 60 and "max_clus" was set to 10 to avoid over-clustering; and "mintemp" was set to 0.05 to avoid under-clustering. We evaluated all clusters visually based on the spike shape and its variance, inter-spike interval (ISI) distribution, and the presence of a plausible refractory period. If necessary, we manually adjusted or excluded clusters. Spike waveforms are shown as density plots in all figures (except for Fig. 2A).

In total, we identified $N = 1025$ clusters on 649 wires across 27 experimental sessions from all 18 participants. Clusters from different sessions were treated as statistically independent units. An experienced rater (L.K.) assigned the tips of depth electrodes to brain regions based on post-implantation MRI scans in native space so that units recorded from the corresponding microelectrodes could be assigned to these regions. To assess recording quality (Fig. S1), we calculated the number of units recorded on each wire; the ISI refractoriness for each unit; the mean firing rate for each unit; and the waveform peak signal-to-noise ratio (SNR) for each unit. The ISI refractoriness was assessed as the percentage of ISIs with a duration of <3 ms. The waveform peak SNR was determined as: $SNR = A_{peak}/SD_{noise}$, where $A_{peak}$ is the absolute amplitude of the peak of the mean waveform, and $SD_{noise}$ is the standard deviation of the raw data trace (filtered between 300 and 3000 Hz).

We classified a unit as a single unit based on the following criteria[98,99]: (1) Visual inspection of spike shape and its variance; (2) ratio between peak of the mean waveform (i.e., spike amplitude) and standard deviation at the first sampling point (i.e., noise) greater than 3; and (3) < 1% of the spikes with an inter-spike interval <3 ms, which takes into account that neurons have a refractory period. Units that did not fulfill the criteria of a single unit were considered multi-units.

To ensure sufficient statistical power, we excluded units with a low number of spikes. Specifically, we excluded units with fewer than 25 spikes during any of the following conditions: successful encoding, unsuccessful encoding, successful retrieval, or unsuccessful retrieval. To mitigate the likelihood that effects were driven by a few segments only, units with no spikes in more than 80% of the encoding or retrieval segments of any of these conditions were also excluded from all analyses. Of all 1,025 units, a total of 666 units on 502 wires met the inclusion criteria and were thus included in our analyses. We recorded $n = 210$ units from the amygdala, $n = 130$ units from the entorhinal cortex, $n = 24$ units from the fusiform gyrus, $n = 125$ units from the hippocampus, $n = 2$ units from the insula, $n = 76$ units from the parahippocampal cortex, $n = 92$ units from the temporal pole, and $n = 7$ from the visual cortex.

## Preprocessing of local field potentials

We preprocessed the local field potentials in several ways to prepare them for analyses of spike-field relationships. To minimize the contribution of spike shapes toward phase and power estimations, we computed the mean waveform of each unit in the 100–3000 Hz bandpass-filtered signal and subtracted it from the original local field potential whenever a spike occurred. The start and end samples of the mean spike were tapered to zero to prevent edge artifacts. We then resampled the local field potential to 2 kHz and applied a 4th order finite impulse response band-stop filter to remove electrical line noise at 48–52 Hz and its harmonics at 98–102 and 148–152 Hz. Finally, we demeaned the local field potential to center it around zero.

## Estimation of theta phase and theta power in the local field potential

To estimate theta phase and theta power, we filtered the preprocessed local field potentials between 1 and 10 Hz (comprising both the low- and high-theta frequency range) using an FIR filter (order = 6000). This filtering centers the LFP around a mean of zero and removes DC shifts and other low-frequency intrusions, which might otherwise shift the analytic signal in the complex plane and distort phase estimates obtained from it. To extract instantaneous power and phase angles of our filtered signal, we used the previously established generalized phase approach[59,100]. The generalized phase approach enhances the accuracy of phase estimations in local field potentials filtered across a broad frequency range and ensures that the phase information reflects the broadband fluctuations of the local field potential. We explain the generalized phase approach in detail in Fig. S3. To examine the generalized phase approach's suitability for different forms of LFP fluctuations, we applied it to different simulated signals and observed good phase estimates (Fig. S5). With our chosen settings, the generalized phase approach identified signal periods in which estimated phases progressed with a frequency below 1 Hz (including negative frequencies during which phase progression reverses direction) and replaced the phase values during these periods with phase values from shape-preserving piecewise cubic interpolation. For each spike, we then identified the temporally closest phase and instantaneous power (Fig. 2B). Since we filtered in the 1–10 Hz frequency range we refer to the assigned values as theta phase and theta power.

## Detection of interictal epileptiform discharges

We automatically detected interictal epileptiform discharges (IEDs) in the preprocessed local field potentials following the same methods and detection criteria as described before[96]. Timepoints with high amplitudes, sharp amplitude changes, and power increases across a broad range of frequencies were considered to belong to an IED and excluded from further analyses. Spikes occurring during these timepoints were excluded from all analyses. We do not refer to IEDs as interictal spikes, because we reserved the term spike for extracellularly recorded action potentials throughout the paper[1,39].

## Theta-phase locking

To investigate if there was consistent phase-locking towards a specific phase of the local field potential we visualized the distribution of phases assigned to the spikes of all units in a polar histogram (Fig. 2C). For power-specific analyses, we used a median split of the spikes of each unit, categorizing them as either high or low based on their theta power (Fig. 3A). We computed the pairwise phase consistency (PPC) to quantify the phase synchronization of the resulting theta-phase distributions[71]. The PPC is calculated by averaging the dot product of all possible pairs of unit vectors representing the phases and can take on values between −1 and 1. A value of 1 indicates perfect phase locking and values toward −1 suggest a lack of phase locking toward a specific phase. Aside from exhibiting greater variance with smaller sample sizes, the PPC is unaffected by the number of spikes.

To assess the strength of theta-phase locking and the preferred theta phase for each unit, we calculated the PPC and the circular mean of the theta phases at which the spikes occurred. For each unit, we ranked the empirical PPC within 1001 surrogate PPC values, which we obtained by circularly shifting the theta-phase data relative to the spike time series and recomputing the PPC (Fig. S10). A unit was considered to exhibit significant theta-phase locking, if the empirical PPC exceeded the 95th percentile of all surrogate PPC values.

When calculating the PPC was computationally impractical due to very large sample sizes (e.g., Fig. 3A), we used the Rayleigh test for non-uniformity as an alternative. In contrast to PPC values, Rayleigh $z$-values are strongly biased by the number of spikes (Fig. S11). The Rayleigh $z$-statistic was derived using the equation $z = n \times r^2$, where $n$ is the number of spikes, and $r$ is the mean resultant vector length (MRVL). The MRVL quantifies the consistency of phase angles. A value close to 0 indicates low phase locking and a value close to 1 indicates high phase locking.

## Theta-phase locking as a function of theta power

To compare phase locking between spikes with high and low power (Fig. 3A), we calculated the empirical difference of the two Rayleigh $z$-values and ranked it within a distribution of 10,001 $z$-value differences obtained by randomly shuffling the spike labels "high" and "low" (Fig. S10). As the Rayleigh $z$-value increases with the number of spikes (Fig. S11), we ensured that the exact number of spikes in both conditions was maintained in all surrogate rounds.

We examined the modulation of theta-phase locking by theta power across different experimental periods: encoding, recall, and baseline (where the baseline period comprised the entire session except for encoding and recall periods). For each period, we separated neuronal spikes into high and low theta power groups based on a median split. We then computed the PPC for both high and low power spikes of each unit, followed by a paired $t$-test to compare the PPC between conditions (Fig. 3B). We ranked the empirical $t$-value among 10,001 surrogate $t$-values (Fig. S10). The surrogates were generated by randomly swapping the labels of the high and low power condition for each unit.

## Theta-phase locking as a function of brain region

We investigated whether phase locking varied across different brain regions during encoding, retrieval, and baseline. We included only brain regions with data from at least five sessions: amygdala, entorhinal cortex, hippocampus, parahippocampal cortex, and temporal pole. We first compared the number of significantly phase-locking units with the number of units with no significant phase locking across brain regions using Chi-squared tests (Fig. 4A). Resulting $P$-values were Bonferroni corrected for applying these tests on the three experimental periods baseline, encoding, and retrieval. We next compared the PPC across units using a linear mixed-effects model (dependent variable: PPC; fixed effect: brain region; random effects: experimental session and period; Fig. 4B).

We also examined regional differences in spike counts and spike-associated theta power, given their potential effects on phase locking. Due to the heterogeneous distribution of spike counts, we decided to use non-parametric Kruskal–Wallis tests to compare spike counts across brain regions (Fig. 4C). We applied three tests, each corresponding to one experimental period, and Bonferroni corrected the resulting $P$-values. To compare spike-associated theta power (Fig. 4D), we computed the average log-transformed theta power assigned to the spikes of each unit and compared the results using a linear mixed-effects model (dependent variable: log-transformed theta power; fixed effect: brain region; random effects: experimental session and period).

## Spectral parameterization resolved in time

In addition to periodic, narrowband oscillations, local field potentials typically display a characteristic power distribution known as 1/f-like aperiodic activity, which appears as a straight line when plotted in log-log space. The slope of this aperiodic activity may reflect the balance of neuronal excitation and inhibition with steeper slopes reflecting stronger inhibition[74].

We utilized the previously established Spectral Parameterization Resolved in Time (SPRiNT) approach to disentangle aperiodic and periodic activity in a temporally resolved manner[61]. SPRiNT leverages spectral parameterization[60] on short time windows. Our analyses showed that the outcome of the SPRiNT algorithm exhibited variable results based on different configuration settings. We thus adhered to default settings unless our visual inspection identified specific configurations that enhanced the algorithm's compatibility with our dataset.

We applied SPRiNT to the 2 kHz resampled local field potentials of each session. The algorithm first computed short-time Fourier transforms from 1-s time windows with 50% overlap. Every 500 ms, it then averaged the power spectra from five consecutive time windows. Power spectra containing automatically detected interictal epileptiform discharges or artifacts were excluded from further analyses. For spectral parameterization, the algorithm fit an aperiodic component to each average power spectrum within the 1–40 Hz frequency range. To distinguish periodic activity from the aperiodic component, up to three Gaussian peaks were identified. Peaks required a signal-to-noise ratio >2 and a height >0.5. We set the maximum peak width to 6 Hz (default = 12 Hz) to extract only narrow oscillatory peaks. We set the minimum peak width to 2 Hz (default = 0.5 Hz) after observing artificial peaks at stereotypical frequencies that lacked clear correlates in the power spectra. This adjustment effectively reduced the presence of these artifacts. We applied a proximity threshold to remove a peak if its center frequency fell within one standard deviation (default, 2 SD) of a relatively larger peak's center frequency or within one standard deviation (default, 2 SD) from the edge of the frequency window. We adjusted this threshold to facilitate fitting peaks closer to the spectrum's edge.

To examine how steeper aperiodic slopes correlated with a stronger influence of lower frequencies within the 1–10 Hz range, we estimated how slope steepness correlated with the cycle-by-cycle frequencies computed from the generalized phase estimates (described in the paragraph below). We then computed a mean Spearman's *rho* for each session and performed a two-sided $t$-test across sessions.

## Oscillation detection

We used a cycle-by-cycle approach to identify 1–10 Hz oscillations in the time domain[62]. The first step was to segment the entire 1–10 Hz filtered signal into putative oscillatory cycles lasting from one peak to the next. To this end, peaks and troughs were identified throughout the recording. In contrast to the original Bycycle approach, we did not determine peaks and troughs in the bandpass filtered signal of interest. Instead, we identified peaks and troughs based on the phase estimates provided by the generalized phase approach. In the rare case where

peaks and troughs did not alternate (due to incomplete phase reversals not corrected by the generalized phase approach), repeating peaks or troughs were identified. For repeating peaks, the peak with the maximum value was retained; for troughs, the trough with the minimum value was kept. Although the peaks and troughs identified using this adjusted method did not always match the local maxima and minima in the raw LFP signal perfectly, it ensured that the analyzed cycles corresponded to the 1–10 Hz fluctuations captured by the generalized phase approach for phase estimation. After the signal was segmented into putative cycles, we applied the Bycycle algorithm to categorize consecutive cycles as being part of an oscillation or not. Oscillations were identified as time periods in which consecutive cycles had similar cycle amplitudes (amplitude consistency threshold = 0.3), similar periods (period consistency threshold = 0.5), and predominantly monotonic decay and rise flanks (monotonicity threshold = 0.6). These settings are identical to those used in the original Bycycle publication for the detection of beta oscillations in motor cortical recordings[62]. Oscillations had to consist of at least two cycles and cycle amplitudes had to be above the twentieth percentile to avoid low-power periods of the signal being considered part of an oscillation. The frequency of an oscillation was computed as the ratio of the number of cycles divided by how long in seconds the oscillation lasted. We confirmed accurate peak, trough, and oscillation detection by visually inspecting the results.

### Firing rate as a function of aperiodic slope and theta oscillations

As a control analysis, we assessed the influence of aperiodic slope and theta oscillations on neuronal firing rates. To this end, we computed the aperiodic slope for each 500-ms time window using SPRiNT. We then split the time windows into high slope time windows (indicating a steeper negative slope) and low slope time windows (indicating a flatter negative slope) using a median split. We calculated the overall firing rate during high and low slope periods for each neuron. We used a paired *t*-test to compare firing rates between high and low slope conditions. To assess significance, we ranked the empirical *t*-value within a distribution of *t*-values derived from 10,001 surrogate datasets. These surrogate datasets were generated by randomly swapping the labels "high slope" and "low slope" for each neuron's firing rates with a 50% probability. We performed this analysis for both putative interneurons and putative pyramidal cells. To investigate the influence of clear theta oscillations on neuronal firing rates, we performed the same analysis with the only difference being that we split the data into two groups based on whether Bycycle detected theta oscillations.

### Phase locking as a function of aperiodic slope

We aimed at understanding the impact of aperiodic slope on neuronal theta-phase locking. To this end, we assigned each spike the aperiodic slope value derived from its corresponding 500 ms SPRiNT time window. We then did a median split of the spikes of each unit, categorizing them as either high or low based on their slopes. Similar to the power-specific analyses, we computed the PPC for both high- and low-slope spikes of each unit, followed by a paired *t*-test to compare the PPC between high and low slopes across units. To assess significance, we ranked the empirical *t*-value within 10,001 surrogates (Fig. S10). These surrogates were generated by randomly swapping the labels of the high- and low-slope condition for each unit. We performed this analysis separately for spikes during baseline, encoding, and retrieval.

### Analysis of theta oscillations

We analyzed the occurrence of theta oscillations in the local field potentials by computing how much time of the signal was occupied by 1–10 Hz oscillations divided by the total signal duration for all 502 wires (Fig. 6D). We rounded the frequencies of all detected oscillations to the nearest integer and computed the mode frequency between 1 and 10 Hz for each wire (Fig. 6E). We also calculated the probability distribution of frequencies for all oscillations on each wire, using a 0.5 Hz frequency resolution (Fig. 6F; Fig. S16).

### Phase locking as a function of theta oscillations

We analyzed the impact of the presence of theta oscillations on theta-phase locking for spikes during baseline, encoding, and retrieval. To this end, we determined whether a spike occurred within a time window where a theta oscillation was detected by Bycycle or not. We computed the PPC for spike phases in the presence versus absence of clear theta oscillations, followed by a paired *t*-test to compare the resulting PPC across units. We then ranked the resulting *t*-value of the paired *t*-test among 10,001 surrogates. These surrogates were generated by randomly swapping the labels of the theta and no theta condition for each unit (Fig. S10).

### Theta-phase locking during successful and unsuccessful encoding and retrieval

To assess and compare theta-phase locking strength and theta-phase angles at the unit level between segments associated with successful versus unsuccessful memory performance, we performed the following procedure separately for encoding and retrieval. We extracted all spike phases during successful and unsuccessful segments and computed the PPC and the mean phases of the two groups. We then performed a paired *t*-test to compare the PPC of the units' spike phases. Next, we ranked the empirical *t*-value of the *t*-test in a distribution of 10,001 surrogate *t*-values. These surrogates were generated by randomly swapping the labels of the successful and unsuccessful condition for each unit (Fig. S10). The units' theta-phase locking strength was considered significantly higher for successful versus unsuccessful memory performance if the empirical *t*-value was above 95% of the surrogate *t*-values. The resulting *P*-values were Bonferroni corrected to account for the fact that we performed the test for both encoding and recall segments.

We also performed a Watson–Williams test to compare the preferred theta phases during successful versus unsuccessful segments. We ranked the empirical *F*-value of the Watson–Williams test in a distribution of 10,001 surrogate *F*-values. Surrogates were obtained by randomly reassigning each spike to either the successful or unsuccessful condition, while maintaining equal spike counts for both conditions in each surrogate dataset (Fig. S10). This approach ensured that any potential bias introduced by varying spike numbers was preserved in the surrogates. We considered the units' preferred theta phases to differ between successful and unsuccessful memory performance, if the empirical *F*-value of the Watson–Williams test was above 95% of the surrogate *F*-values. The resulting *P*-values were Bonferroni corrected to account for performing the test for encoding and recall.

### Spike-field coherence

We used spike-field coherence (SFC) to quantify phase locking during successful and unsuccessful memory performance for both encoding and retrieval segments[39,101] in a frequency-resolved way (Fig. 7D). SFC measures the percentage of the power spectrum of the spike-triggered average (STA) over the average power spectrum of the local field potentials around the spikes, which were used to compute the STA. For each frequency, the SFC takes values between 0 and 100%. High SFC values indicate strong phase locking at a specific frequency[39].

To compute the SFC, we first extracted 500 ms of the 2 kHz resampled and mean-spike-removed local field potentials before and after each spike. Next, we resampled the local field potentials to a sampling rate of 250 Hz. The spike-triggered average was obtained by averaging the resulting traces. The power spectrum of the STA and the power spectra of the spike-triggered local field potential traces were calculated at linearly spaced frequencies from 1 to 100 Hz with intervals of 1 Hz using the FieldTrip implementation for multitaper analysis[102,103]. We used Slepian tapers and a 4-Hz width of frequency

smoothing. When comparing the SFC between the two conditions, we used the same number of spikes for each condition. We randomly selected the number of spikes of the condition with fewer spikes for the condition with more spikes. This subsampling procedure was repeated 100 times and the resulting 100 SFC values were averaged for each frequency. To test for significant differences between successful and unsuccessful segments, we performed a paired *t*-test between conditions across units at each frequency bin and Bonferroni corrected the *P*-values for 100 comparisons[39] and for performing the analysis separately for encoding and retrieval.

### Memory effects on phase locking for different subgroups
To investigate memory-related effects possibly restricted to specific parts of our dataset, we divided it into subgroups to compare phase-locking strength between successful versus unsuccessful memory performance. In each subgroup analysis, we applied Bonferroni correction for four comparisons given that we tested two data groups (e.g., high and low theta power) and two trial periods (encoding and retrieval).

We grouped the data into the following subgroups: (1) separation of spikes based on a median split of their associated theta-power values; (2) separation of spikes based on a median split of their aperiodic slopes; (3) separation of spikes based on the presence versus absence of clear theta oscillations; (4) separation of units based on whether they were object-responsive or object-non-responsive units; (5) separation of units based on whether they exhibited theta-phase locking; (6) separation of units based on whether they were single units or multi-units; and (7) separation of recall periods based on whether they were object-recall or location-recall periods.

We identified object-responsive units as those units that exhibited a significant increase in firing during the encoding period of the task. To this end, we calculated firing rates for each encoding segment and its corresponding baseline segment, the latter defined as the 1.5 s preceding the encoding period. We then compared the firing rates between encoding and baseline segments using a paired *t*-test. We ranked the resulting *t*-value within a distribution of 10,001 surrogate *t*-values that we obtained by randomly swapping/keeping the labels "encoding" and "baseline" for each encoding–baseline pair (Fig. S10). Units with a *t*-value greater than 95% of the surrogate *t*-values were considered object-responsive units.

### Theta-phase shifts between encoding and retrieval
We also performed analyses to investigate how many units exhibited a significant phase shift between encoding and retrieval. For each unit, we used a Watson–Williams test to compare the phase distribution of all its spikes during encoding segments with the phase distribution of all its spikes during retrieval segments. We then created 10,001 surrogate datasets by randomly shuffling the segment labels "encoding" and "retrieval" (Fig. S10) and performed a Watson–Williams test for every surrogate dataset. We ranked the empirical *F*-value of the unit's Watson–Williams test in the distribution of the 10,001 surrogate *F*-values and considered a unit to exhibit a significant theta-phase shift between encoding and retrieval if the empirical *F*-value exceeded the 95th percentile of all surrogate *F*-values.

To confirm that the 95th percentile was chosen correctly when evaluating the statistical significance of units, we performed the following control analysis: We concatenated all encoding and retrieval phases in the order of their occurrence and circularly shifted the resulting list of phases by a random lag. We then reassigned the same number of spikes and their theta phases from the shifted list to the corresponding encoding and retrieval segments. This way, we created 1001 surrogate datasets on which we performed the same analysis as the one we had performed on the empirical data. By calculating the average percentage of shifting units in each of the surrogate datasets, we obtained a distribution whose mean confirmed our a priori chosen alpha level of 5% (Fig. 9F).

We performed the analysis of phase-shifting neurons separately for all units and phase-locking units, and we separately analyzed all, successful, and unsuccessful segments (Fig. 9C–F). We evaluated whether the number of shifting units observed in the empirical data was higher than the chance level of 5%. We also tested for significant phase shifts as a function of high versus low theta power, high versus low aperiodic slope, and the presence versus absence of clear theta oscillations.

### Reporting summary
Further information on research design is available in the Nature Portfolio Reporting Summary linked to this article.

## Data availability
Data to produce all figures are provided at https://github.com/BonnSpatialMemoryLab/GuthPhaseLocking2025[103–105] within the directory "SourceData_20250129". Source data are provided with this paper.

## Code availability
Data analyses were performed with custom Matlab code and using the Matlab toolboxes CircStat[104] and FieldTrip version 20210912[103]. All custom Matlab code generated during this study for data analysis is available at https://github.com/BonnSpatialMemoryLab/GuthPhaseLocking2025[105].

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

## Acknowledgements

We are very grateful to all patients who participated in this study. We thank the clinical team of the Freiburg Epilepsy Center, Freiburg im Breisgau, Germany, for their continuous support. We thank the members of the Jacobs Lab at Columbia University and the members of the Mormann Lab at the University Hospital Bonn for feedback. T.A.G., A.B., A.S.-B., and L.K. were supported by the Federal Ministry of Education and Research (BMBF; 01GQ1705A and 01GQ2402A) and by NIH/NINDS grant U01 NS113198-01. T.A.G. was supported by a stipend of the German National Academic Foundation (Studienstiftung des deutschen Volkes). P.C.R. received research support from the Fraunhofer Society (Munich, Germany) and personal fees, travel support, and honoraria from Boston Scientific, Brainlab, and Inomed. J.J. was supported by NIH grant MH104606 and the National Science Foundation (NSF). L.K. received funding from the German Research Foundation (DFG; Projektnummer 447634521) and was supported by the return program of the Ministry of Culture and Science of the State of North Rhine-Westphalia.

## Author contributions

T.A.G. and L.K. developed hypotheses and designed the study; L.K., A.S.-B., and J.J. acquired funding; L.K., A.S.-B., P.C.R., and A.B recruited study participants; P.C.R. implanted electrodes; L.K., A.B., and T.A.G. collected the data; T.A.G. analyzed the data; T.A.G., J.J., and L.K. discussed the results; T.A.G. and L.K. wrote the paper; all authors reviewed and revised the final manuscript.

## Funding

## Competing interests

The authors declare no competing interests.
