## [Transparent Peer Review file · Nature Communications]

Theta-phase locking of single neurons during human spatial memory

Corresponding Author: Dr Tim Guth

Version 0:

Reviewer comments:

Reviewer #1

(Remarks to the Author)

The significance of this study is that a large amount of extracellular recordings from epileptic human subjects were obtained during a location and object recall behavioral task in a virtual environment. Similar work by these authors focused on ripples rather than theta has been published in Nature Neuroscience. This type of study is required to determine whether theories and results from rodent and monkey studies apply to humans. They attempted to determine whether theta phase locking of individual neurons was stronger during successful memory recall and whether the theta phase shifted between encoding and retrieval phases.

This study did not find stronger theta phase locking during successful memory performance during encoding or retrieval, in contrast to previous results in humans by Rutishauser et al 2010 and Yoo et al 2021. Their explanation was that their spatial memory task featured encoding and retrieval periods smoothly embedded into the other trial periods. I question whether the design of the spatial memory task was optimal to find differences in theta phase locking during successful versus unsuccessful trials. Also previous studies on humans found that rhinal and hippocampal theta and successive alpha power enhancement before word presentation predicted successful memory encoding.¹ Another study by the same group found that successful as opposed to unsuccessful memory formation was associated with a general rhinal-hippocampal coherence enhancement, but without alterations in spectral power.² Maybe coherence between regions would be a better readout of successful recall.

This study did find some evidence for a phase shift between encoding and retrieval, but the average value was 90 degrees, not the 180 degrees predicted by the Specific Phases for Encoding and Retrieval hypothesis. However, this is consistent with the smaller phase shifts observed in rodents 3–5, which should be noted in this manuscript. The result that the aperiodic slope of the power spectrum is a major determinant of the strength of theta phase locking is novel. The result that firing rates did not differ between periods with and without theta oscillations is somewhat surprising based on⁶: "the phase-locking phenomenon is primarily related to increased firing during high-power oscillations".

Statistics were rigorous and generally performed using surrogate data. The generalized phase approach is well justified for nonstationary oscillations in the theta band. The description of the methodology is comprehensive and sound with one exception. I disagree with the statement in lines 588 to 589 that "It currently remains an open question whether it is useful to investigate theta-phase-locking during periods without clear theta oscillations." The concept of phase only applies when the clear presence of an oscillation has been established. If there is locking to the theta frequency component of an aperiodic (or at least theta aperiodic signal) that implies theta resonance in individual neurons. If that is what the authors intend to imply, it should be clearly stated, otherwise references to and figures regarding theta phase in the absence of theta oscillations should be removed.

1. Fell, J. et al. Medial temporal theta/alpha power enhancement precedes successful memory encoding: evidence based on intracranial EEG. *J. Neurosci. Off. J. Soc. Neurosci.* 31, 5392–5397 (2011).
2. Fell, J. et al. Human memory formation is accompanied by rhinal-hippocampal coupling and decoupling. *Nat. Neurosci.* 4, 1259–1264 (2001).
3. Manns, J. R., Zilli, E. A., Ong, K. C., Hasselmo, M. E. & Eichenbaum, H. Hippocampal CA1 spiking during encoding and retrieval: Relation to theta phase. *Neurobiol. Learn. Mem.* 87, 9–20 (2007).
4. Douchamps, V., Jeewajee, A., Blundell, P., Burgess, N. & Lever, C. Evidence for Encoding versus Retrieval Scheduling in

the Hippocampus by Theta Phase and Acetylcholine. *J. Neurosci.* 33, 8689–8704 (2013).

5. Poulter, S., Lee, S. A., Dachtler, J., Wills, T. J. & Lever, C. Vector trace cells in the subiculum of the hippocampal formation. *Nat. Neurosci.* 24, 266–275 (2021).

6. Jacobs, J., Kahana, M. J., Ekstrom, A. D. & Fried, I. Brain Oscillations Control Timing of Single-Neuron Activity in Humans. *J. Neurosci.* 27, 3839–3844 (2007).

(Remarks on code availability)

Reviewer #2

(Remarks to the Author)

In the manuscript, authors provide a thorough analysis of theta phase-locking in human via deep intra-cranial recordings during memory formation and retrieval. This aims to provide a comprehensive account of the relationship between theta and neuronal spikes through latest conceptual and methodological approach to describe oscillations. It is meticulous, but while the analysis appears well conducted, I find the results difficult to interpret for non-specialists, especially the part on phase locking during aperiodic epochs. While it is partly based on recent methodology showing promising framework for interpreting oscillations, it needs more contextualization, as it may at first read appear paradoxical, and thus it is difficult to grasp what the finding is. Next, the phase locking analysis reveals no strong and important relationships between theta and spikes depending on learning, and the negative result is important. It contrasts with recent findings suggesting theta modulation with learning. Thus, perhaps the strongest message of the paper, is that despite strong and robust attempt to reveal differences between successful and unsuccessful trials during encoding or recall (fig 4), there were no differences in any of the measures tested. The paper may gain by stressing out this even more clearly, although authors put forward as one possible explanation, the difference of task structure compared to other tasks. I provide some comments below to help with readability and interpretations.

1. Many neurons have theta phase locking, and this is stronger during high theta. This is observed for all periods (baseline, encoding, recall). How can Rayleigh z-values vary so much across periods from 10 to 200 to 1000 between Fig2E and G? I understand that the duration varies but why didn't authors use random time duration samples for baseline that match duration of encoding or recall? Is there a way to identify periods of theta oscillation, or a periodicity in an agnostic way to the epoch of the task and then quantify the probability of these oscillations or aperiodicity to be associated to specific task segments given overall duration and relative duration of each period? Or alternatively, to construct a distribution randomly sampling equal population size during baseline as for the other epochs?

2. Authors describe a variation of theta phase locking across regions. There is a higher percentage of cells with phase locking in the parahippocampal cortex compared to hippocampus.

Here authors suggest that 1) because the lower phase locking in other regions stems from more variability in theta phase locking (i.e. Rayleigh tests are low because coupling of spikes with phases were more dissimilar), 2) this means an increased potential for phase coding (note here the use of coding instead of locking).

I presume here that authors suggest that phase coding lies within the variability in the relationship between the theta phase and the spikes emitted, but proven that within each cell, this relationship is preserved across the coded events, and that different cells have different phase codes.

If the individual cell does not preserve its phase lock within a session for an event, then it is hard to qualify as "codes" these random relationships between the phase and the spikes. So, I am not sure I understand the author's rationale, given that a code should be reproduced to be decoded. This becomes even more difficult to understand if the relationship changes between learning and retrieval. Is that to relate to the last paragraph for which the majority of cells did not alter code between learning and retrieval?

3. If I understand correctly, the aperiodic component (rather than activity) is the overall slope of the fit (in parallel to oscillations, but once the oscillatory peak has been removed), whose steepness depends on the relative ratio of low frequency over high frequency. A high slope means more low frequency compared to high frequency, while no slope would mean all frequencies at same power. Changes in slope steepness from one moment to another, could mean that theta oscillations got lower (shifted from high to low theta), or even that higher oscillations got suppressed (the latter fitting with lower firing rate is one assumes high oscillations are linked to firing rate).

The section aiming to describe the results linked to aperiodic epochs is really hard to follow for non-specialists. It is difficult to understand that it is not exactly oscillatory and non-oscillatory activity that authors compare in the section, but whether the phase locking is different depending on the steepness of the slope of the aperiodic components.

It might be necessary to explain more clearly whether there is a difference between periods without theta oscillations and "aperiodic" epochs, because otherwise, it is difficult to understand how one can obtain phase locking on an arrhythmic period given that the phase is the relationship between the oscillation and spike calculated based on the generalized phase as shown in Figure 2B.

Likewise, It is difficult to put together the 2 sentences in the same framework:

Line 265: "We found that theta-phase locking was stronger for neuronal spikes associated with high aperiodic slopes as compared to those associated with low aperiodic slopes"

And

Line 295 "During baseline and recall (but not during encoding) we found that phase locking was stronger for neuronal spikes occurring during theta oscillation"

4. I was wondering if steepness of the slope and theta phase locking could be related to putative excitatory principles cells or inhibitory interneurons. Could be interesting to test the phase locking strength depending on firing rate?

5. The thorough analysis of phase during encoding and retrieval suggests that there isn't any difference between correct and error trials. The analysis of theta phase locking shifts revealed a very modest shift between encoding and retrieval both in proportion and shift amplitude. There is some modulation considering comparison between epochs through other measures (aperiodic, high theta), but all results are very modest. In sum, the paper, actually provides some grounds to reconsider the robustness of phase-locking codes as a meaningful physiological signal linked to memory or information processing. Many results are modest or negative, and but provided that the methods are very robust, the message of the paper may gain in clarity if the lack of conclusiveness is better stated.

6. What are the measures shown in Fig1C and E? distribution of the average percent correct for each session? If a recall is a single shot, how is the distribution in figure 1E obtained (3 out of 3)?

(Remarks on code availability)

Reviewer #3

(Remarks to the Author)

In the submission, Guth et al. present interesting and detailed results on the phase-locking of single neurons in the human MTL to local field potentials. The general research question, approach, and potential implications are timely and relevant for a better understanding of MTL electrophysiology and could lead to a more profound knowledge of human spatial memory functioning.

The experimental setup and data are rare, and the methods and analyses sound. A key novelty of the study is an innovative generalized-phase approach to investigate phase-locking in a frequency-adaptive manner. As discussed in the manuscript, this is a major difference from previous studies on single-neuron phase-locking.

From my understanding, the generalized-phase algorithm ensures the centering of complex plane representations by excluding lower frequencies. In the original manuscript introducing the generalized-phase approach, an important initial step is to exclude this low-frequency content through filtering in the 5-40 Hz frequency range. In this previous work, low frequencies were excluded for these technical reasons but also because they are thought to reflect slow global changes such as arousal¹.

Theta frequencies might vary in humans, and MTL oscillations are not continuous. Previous literature investigated theta-phase locking of single neurons in the 3-8 Hz² frequency range, distinguished between "slower" (2-5 Hz) and "faster" (5-9 Hz) theta³, and included slower frequencies in the 1-4 Hz range, which were discussed as delta oscillations. Given the 1-10 Hz frequency range in the present submission, I would consider discussing the findings as phase-locking to non-oscillatory LFP equally justified. However, in the manuscript, the authors present and discuss the results as theta-phase locking (Title, Abstract, Discussion).

One could argue that theta-phase locking would require the presence of an oscillation in the ~3-10 Hz range to draw comparisons to previous human and animal work. The absence of slower fluctuations is needed to estimate the phase of these oscillations correctly, given the chosen generalized phase approach at 1-10 Hz in the present submission. In other words, theta oscillations need to be present and become the most prominent fluctuation to estimate theta-phase locking appropriately.

Otherwise, phase locking to non-oscillatory LFP fluctuations is investigated, which is still interesting and novel, especially following literature in bats⁴, but the present submission does not distinguish phase-locking to oscillations or non-rhythmic fluctuations. The manuscript includes analyses on phase-locking, dependent on the presence of oscillations, but that does not guarantee that these oscillations are the most prominent fluctuations within these time windows. Therefore, the influence of slower fluctuations potentially biases the phase estimates, and related results cannot be ruled out.

These limitations temper my general enthusiasm for the manuscript, which I consider to be excellent apart from these comments. Yet, the points I raised above might considerably impact the results and conclusions drawn in the submission, which need to be clarified before publication. Please find my specific comments and suggestions below:

Major:

- 1.) Please clarify how the generalized-phase approach can prevent bias from low frequencies within the chosen 1-10 Hz range (as opposed to the 5-40Hz filter of the original work), affecting the phase estimates in the present submission.
- 2.) Given the inclusion of broad frequencies, the results in the current form might be rather discussed as phase-locking of single neurons to non-oscillatory LFP fluctuations. As mentioned above, the presence of oscillations does not necessarily guarantee correct phase estimation here. To ensure the absence of slow frequencies and appropriate oscillatory phase estimates, filtering in the ~3-10 Hz might be required to claim phase locking to theta oscillations. It would be especially innovative and novel to distinguish between non-oscillatory phase locking and phase-locking to theta oscillations.

Minor:

- i) Please mention that participants actively navigated through the virtual environment using the keyboard, etc.

- ii) Figure 1C: One would expect a chance level such as in E here; I guess the chance might be 1 divided by the number of objects. If so, some mention of the chance level would be helpful here.
- iii) Line 148: A brief description of the generalized-phase approach would be beneficial here. At least mention that the instantaneous phase is not simply estimated from the analytical signal⁵. From my understanding, generalized phase estimation overcomes the bias from low frequencies through filtering and suppresses confounds of higher frequencies by correcting for negative frequency components.
- iv) Line 151. Some reference might be helpful here to support the observation that theta oscillations are non-continuous in humans^{6–8} in bats⁴.
- v) Phase-locking related to baseline periods. Theta power includes all frequencies between 1-10 Hz here. How could arousal effects in the 1-3 Hz range affect these results related to disengagement from any task?
- vi) Figure 2: Mention the frequency range for the LFP illustrated in panel B.
- vii) Line 195f: Briefly mention how power was computed here.
- viii) Line 255: I'm unsure if "neuronal spikes" is the best term here, given that spikes commonly refer to epileptic spikes; maybe "neuronal fluctuations" might be more appropriate.
- ix) Line 272: Steeper slopes could also increase the influence of lower frequencies biasing phase estimation, as discussed above. Could you rule such influence out or estimate it?
- x) Line 275ff: There might be a discrepancy in what "theta" means in the manuscript. Here, the presence of theta oscillations within the ~3-10 Hz range was tested (Fig. 3G), which resonates with previous literature, whereas phase-locking of single neurons to LFP and power in the theta range refers to phenomena at frequencies between 1-10 Hz.
- xi) Figure 4E: It would be interesting to see the phase differences between encoding and retrieval as a function of frequency.
- xii) Figure 5: The sign of the phase differences is not clear here (encoding-retrieval). From the examples in the polar plots, it looks like neurons lock to earlier phases during retrieval than encoding, which is the opposite of the hypothetical example as illustrated on top. Please clarify.

References

1. Davis, Z. W., Muller, L., Martinez-Trujillo, J., Sejnowski, T. & Reynolds, J. H. Spontaneous travelling cortical waves gate perception in behaving primates. *Nature* 587, 432–436 (2020).
2. Rutishauser, U., Ross, I. B., Mamelak, A. N. & Schuman, E. M. Human memory strength is predicted by theta-frequency phase-locking of single neurons. *Nature* 464, 903–907 (2010).
3. Yoo, H. B., Umbach, G. & Lega, B. Neurons in the human medial temporal lobe track multiple temporal contexts during episodic memory processing. *NeuroImage* 245, 118689 (2021).
4. Eliav, T. et al. Nonoscillatory Phase Coding and Synchronization in the Bat Hippocampal Formation. *Cell* 175, 1119–1130.e15 (2018).
5. Gabor, D. Theory of communication. Part 1: The analysis of information. *Journal of the Institution of Electrical Engineers - Part III: Radio and Communication Engineering* 93, 429–441 (1946).
6. Watrous, A. J. et al. A comparative study of human and rat hippocampal low-frequency oscillations during spatial navigation. *Hippocampus* 23, 656–661 (2013).
7. Aghajan, Z. M. et al. Theta Oscillations in the Human Medial Temporal Lobe during Real-World Ambulatory Movement. *Curr Biol* 27, 3743–3751.e3 (2017).
8. Bohbot, V. D., Copara, M. S., Gotman, J. & Ekstrom, A. D. Low-frequency theta oscillations in the human hippocampus during real-world and virtual navigation. *Nat Commun* 8, 14415 (2017).

(Remarks on code availability)

Version 1:

Reviewer comments:

Reviewer #2

(Remarks to the Author)

I thank the authors for revising their manuscript accounting for my comments. I really appreciate the whole new analysis of the data, and additions for relating aperiodic slopes and theta oscillations. The pedagogical efforts in figure S12 and Fig3C are also a greatly appreciated. I think the manuscript is much stronger and the nature of the results it provides are clearer now and deserves publication.

(Remarks on code availability)

Reviewer #3

(Remarks to the Author)

The authors addressed all of my comments, and I recommend publication. The rebuttal is rigorous, comprehensive, and genuinely insightful. The additional data and analyses supported the author's main findings and clarified additional questions.

I only have a few suggestions that may be included during the production process and do not require another revision from

my side.

+ I do not insist on the terminology “non-oscillatory phase locking”; the acknowledgment that non-stationary theta oscillations were observed in this study would suffice. I leave the exact terminology to the authors.

+ It would be interesting to test whether there are systematic phase shifts for the data presented in Fig. S21G, as the upper frequency shows a cluster of phase shifts between 30-60 degrees. One way to assess this would be to test whether the population's mean vector has a preferred phase by using phase circular statistics, e.g., comparing to surrogate data with shuffled phase-to-spike relations.

+ In Fig. S21D-G, what "10" means in the polar plot could be more apparent. I assume the number of neurons, so this label could be plotted in grey or briefly mentioned in the caption (to distinguish the phases in the polar plot).

(Remarks on code availability)

Theta-phase locking of single neurons during human spatial memory

Response to Referees

We thank the Reviewers and the Editor for the careful assessment of our manuscript. We are happy to report that the Reviewers' comments and suggestions have helped to further improve the manuscript. Here is a summary of our main revisions:

- We now use “pairwise phase consistency” (Vinck et al., NeuroImage, 2010) in most analyses instead of the Rayleigh test to estimate theta-phase locking. All findings are qualitatively identical compared to our previous manuscript version, demonstrating their robustness. The advantage of using pairwise phase consistency is that this metric does not depend on the sample size, ensuring better comparability across experimental conditions and simplifying statistics. (Reviewer 2, comment 1.)
- We now use “Bycycle” (Cole & Voytek, J Neurophysiol, 2019) to identify theta oscillations because of its better temporal resolution and better detection of low-frequency oscillations as compared to SPRiNT. (Reviewer 1, comment 5; Reviewer 3, comment 2.)
- We provide new analyses and simulations that demonstrate the robustness of the generalized phase approach in handling lower frequencies. (Reviewer 3, comments 1, iii, and ix.)
- We provide new information on finding residual theta-phase locking in the absence of clear theta oscillations, explaining that it is presumably due to subthreshold oscillations not detected by the oscillation-detection algorithm. (Reviewer 1, comment 6; Reviewer 2, comment 3.)
- We now better explain our rationale for choosing the 1–10 Hz frequency band as “theta” and we have extended our discussion of the literature linking low theta to human spatial memory. (Reviewer 3, comments 1, 2, and x.)
- We show that our phase-locking results are similar across the frequency ranges of 1–10 Hz and 3–10 Hz. (Reviewer 3, comment 2.)
- We have added various new results, including theta-phase shifts as a function of higher versus lower theta frequencies (Reviewer 3, comment xi); aperiodic slopes and theta oscillations as a function of task period (Reviewer 2, comment 1); theta-phase locking and firing rates as a function of aperiodic slope for putative interneurons versus pyramidal cells (Reviewer 2, comment 4); and 1–3 Hz oscillations as a function of arousal (Reviewer 3, comment v).
- We have revised our text and figures to enhance clarity and contextualization, and we have added new references as suggested by the Reviewers. (Reviewer 1, comments 2, 3, 4, 5, and 6; Reviewer 2, comments 2, 3, 5 and 6; Reviewer 3, comments i, ii, iii, iv, vi, vii, viii, x, and xii.)

Below, we provide a point-by-point response to each of the Reviewers' comments. Reviewer comments are shown in *italic* font, responses are shown in normal font, and text copied from the manuscript is shown in blue font. Text changed in the manuscript is highlighted in yellow.

We would be happy to make any further changes based on additional reviewer comments or editorial needs.

Reviewer #1, comment #1

The significance of this study is that a large amount of extracellular recordings from epileptic human subjects were obtained during a location and object recall behavioral task in a virtual environment. Similar work by these authors focused on ripples rather than theta has been published in Nature Neuroscience. This type of study is required to determine whether theories and results from rodent and monkey studies apply to humans. They attempted to determine whether theta phase locking of individual neurons was stronger during successful memory recall and whether the theta phase shifted between encoding and retrieval phases.

Response: Thank you very much for your detailed review. Please find our point-by-point responses to your comments below.

Reviewer #1, comment #2

This study did not find stronger theta phase locking during successful memory performance during encoding or retrieval, in contrast to previous results in humans by Rutishauser et al 2010 and Yoo et al 2021. Their explanation was that their spatial memory task featured encoding and retrieval periods smoothly embedded into the other trial periods. I question whether the design of the spatial memory task was optimal to find differences in theta phase locking during successful versus unsuccessful trials.

Response: Thank you for raising this question. We have updated the limitations section of the Discussion to mention potential disadvantages of our spatial memory task. Following your question, we have added that our task may not have been optimal for finding differences in theta-phase locking during successful versus unsuccessful trials (lines 720–731).

Another limitation of this study is the finding that the strength of theta-phase locking was not associated with memory performance. As discussed above, we currently believe that this null effect is due to the dynamic nature of our spatial memory task, which may have prevented theta-phase resets at the onset of the encoding and retrieval periods. Hence, our task design may not have been optimal for detecting differences in theta-phase locking as a function of memory performance, and future studies may utilize static instead of dynamic stimuli to continue investigating relationships between theta-phase locking and memory. Furthermore, our task may not have been difficult enough and the number of encoding and retrieval periods may have been too small (100 each in a full session), because of which the analysis of theta-phase locking as a function of memory performance may have been underpowered. Relatedly, we did not find strong differences in theta-phase shifts between successful versus unsuccessful memory performance, also perhaps because of statistical power. Additional studies are thus needed to better understand the role of stable and shifting theta phases for memory success in humans (6, 7, 50).

Reviewer #1, comment #3

Also previous studies on humans found that rhinal and hippocampal theta and successive alpha power enhancement before word presentation predicted successful memory encoding.1 Another study by the same group found that successful as opposed to unsuccessful memory formation was associated with a general rhinal-hippocampal coherence enhancement, but without alterations in spectral power2. Maybe coherence between regions would be a better readout of successful recall.

Response: We agree that other neural processes beyond theta-phase locking may support successful memory formation and retrieval. Following your comment, we have thus extended our Discussion to describe these findings, including the two studies that you mentioned in your comment (lines 631–639).

Beyond theta-phase locking, various other neural processes may be associated with successful memory encoding. For example, theta power in the hippocampus increases during the presentation of items that are later remembered (30, 54, 85). Pre-stimulus increases in rhinal and hippocampal theta and alpha power are also linked to successful memory encoding (86). Coherence between regions may be important, too, as whole-brain connectivity in the theta- and gamma-frequency ranges and, more specifically, rhinal–hippocampal gamma synchronization accompany successful memory formation (87–90). Future research may investigate the relationship of these diverse electrophysiological phenomena to theta-phase locking and identify their relative importance for successful memory encoding.

Reviewer #1, comment #4

This study did find some evidence for a phase shift between encoding and retrieval, but the average value was 90 degrees, not the 180 degrees predicted by the Specific Phases for Encoding and Retrieval hypothesis. However, this is consistent with the smaller phase shifts observed in rodents 3–5, which should be noted in this manuscript.

Response: Thank you for pointing out these references. Following your suggestion, we have added them to the manuscript (lines 658–669).

We found that theta-phase shifts varied in magnitude and direction between neurons. The absolute extent of these significant shifts between encoding and retrieval was approximately 90° when considering all phase-shifting neurons. A previous human single-neuron study found theta-phase shifts of comparable magnitude (about 100°) during a verbal free-recall task when investigating neurons whose firing rates showed a subsequent memory effect (51). Similarly, in rodents, hippocampal CA1 neurons showed theta-phase shifts of less than 90° when rats navigated through novel versus familiar environments that were used to trigger encoding and retrieval states, respectively (73). Furthermore, rodent CA1 pyramidal cells showed phase shifts of less than 90° when processing novel objects and odors compared to familiar ones (91), and vector trace cells in the subiculum spiked at slightly earlier theta phases (difference of ~40°) when presented with a novel cue (92). Although these empirically observed shifts are smaller in size than predicted by the SPEAR model, they may still help separate encoding and retrieval processes.

Reviewer #1, comment #5

The result that the aperiodic slope of the power spectrum is a major determinant of the strength of theta phase locking is novel. The result that firing rates did not differ between periods with and without theta oscillations is somewhat surprising based on6: " the phase-locking phenomenon is primarily related to increased firing during high-power oscillations".

Response: Thank you very much for pointing this out. Based on this and other Reviewers' comments, we changed our algorithm for detecting theta oscillations to "Bycycle" (Cole & Voytek, J Neurophysiol, 2019). In comparison to our previous method using spectral

parameterization resolved in time (SPRiNT), Bycycle has better temporal resolution by identifying oscillations in the time domain. Using Bycycle, we indeed find increased firing rates during theta oscillations detected by Bycycle. We have updated our manuscript accordingly (lines 347–351).

We furthermore observed that firing rates during clear theta oscillations were higher than during time periods without them (two-sided surrogate analysis with condition-label swapping, $t(665) = 2.598$, $P = 0.007$; Fig. 3D; Fig. S10), which resembles previous observations in humans (36) and may relate to the finding of increased firing rates during periods of increased theta power in the membrane potential of CA1 pyramidal neurons in mice (72).

Reviewer #1, comment #6

Statistics were rigorous and generally performed using surrogate data. The generalized phase approach is well justified for nonstationary oscillations in the theta band. The description of the methodology is comprehensive and sound with one exception. I disagree with the statement in lines 588 to 589 that "It currently remains an open question whether it is useful to investigate theta-phase-locking during periods without clear theta oscillations." The concept of phase only applies when the clear presence of an oscillation has been established. If there is locking to the theta frequency component of an aperiodic (or at least theta aperiodic signal) that implies theta resonance in individual neurons. If that is what the authors intend to imply, it should be clearly stated, otherwise references to and figures regarding theta phase in the absence of theta oscillations should be removed.

Response: Thank you for this important comment. We agree that phase is only defined in the presence of an oscillation. We have thus removed the statement in lines 588–589 as it may be misleading.

The problem that we wanted to address with the sentence in original lines 588–589 is that our empirical data does not allow us to conclude that there are no subthreshold theta oscillations if an oscillation-detection algorithm does not detect them. This is because such algorithms naturally depend on a variety of parameters and the ground truth of whether an oscillation is present is unknown. We thus generally use the term “clear” or “detected” theta oscillations in our manuscript to indicate that our oscillation-detection method identified them, while acknowledging that weaker theta oscillations in other parts of the local field potentials may remain undetected. To explain this point more clearly, we have revised our Discussion regarding our approach of analyzing theta-phase locking across the entire recording and during periods where an oscillation-detection algorithm identified clear theta oscillations (lines 680–693).

To examine theta-phase locking as broadly as possible, we analyzed theta-phase locking across the entire task and tested subsequently whether theta-phase locking was more strongly pronounced during periods with clear theta oscillations detected by an oscillation-detection algorithm (Bycycle). We opted for this two-pronged approach as it currently remains an open question whether it is better to focus on periods with clear theta oscillations to increase specificity, or to use all data including subthreshold theta oscillations to increase statistical power. This open question results from the fact that human theta oscillations are variable in strength and frequency, because of which it is not easily possible to obtain a binary classification into periods with theta oscillations versus those without theta oscillations. This situation is complicated by the fact that current oscillation-detection algorithms depend on a variety of parameters, for which the best thresholds are unknown. Here, we used a cycle-by-cycle

approach (Bycycle) with default settings (59) to identify periods with theta oscillations. Visual inspection suggested that Bycycle performed well, but it may have missed periods with subthreshold theta oscillations. We thus believe it is a useful approach to investigate and compare theta-phase locking both in the presence of clear oscillations and across the remaining signal.

Similarly, in the Results section, we have added language to explain that our use of an oscillation-detection algorithm does not enable a binary separation of the local field potential into periods with versus without theta oscillations. Instead, it distinguishes between periods where theta oscillations are above the detection threshold versus periods where they are below the threshold (lines 331–338).

We note that this practical separation of the local field potential into periods with and without clear theta oscillations is non-trivial and not binary, as it reflects a trade-off between specificity and sensitivity determined by the parameters of the chosen oscillation detection algorithm. Here, we implemented this distinction to enable a relative comparison of theta-phase locking between periods with stronger (suprathreshold) versus weaker (subthreshold) theta oscillations. This does not exclude the presence of weaker theta oscillations during “periods without clear theta oscillations,” which are presumably responsible for residual theta-phase locking in those time periods.

Furthermore, in response to your comment, we have added results from simulations to show that theta resonance alone (in combination with a purely aperiodic local-field potential) does not produce theta-phase locking. This provides further support for the idea that theta-phase locking during periods without clear theta oscillations results from subthreshold oscillatory activity that is missed by the oscillation-detection algorithm (lines 358–362).

Residual theta-phase locking during periods without clear oscillations suggests that some oscillatory activity may have been missed by the oscillation-detection algorithm. In support of this interpretation, control simulations showed that intrinsic theta resonance of individual neurons does not lead to theta-phase locking if the local field potential only consists of pink noise and does not contain any oscillatory activity (Fig. S17).

We show the simulation results in a supplementary figure (Fig. S17), which is copied below.

Figure S17. Theta resonance of individual neurons in combination with a purely aperiodic local field potential does not explain theta-phase locking. We simulated one hundred aperiodic signals between 1–10 Hz with a duration of 3,600 seconds. For each of these signals, we simulated 91 spike trains with theta resonance at linearly increasing frequencies from 1 to 10 Hz. (A) Ten seconds of an example aperiodic signal and a spike train with resonance at 2 Hz. (B) The power spectrum of the example signal in log-log space confirms that it contains 1/f-like aperiodic activity between 1–10 Hz but no oscillatory peaks. We simulated the slope of the aperiodic activity with a steepness of 2. (C) Polar histogram showing the distribution of spike-associated phases of the example spike train and signal. We computed the pairwise phase consistency (PPC = $-8.412 \cdot 10^{-5}$) and compared it to 101 surrogate PPC-values obtained from random phase distributions. The rank of the empirical PPC in the surrogate PPC distribution was subtracted from one to compute a *P*-value of 0.634, indicating a uniform distribution of phases. Accordingly, there is no theta-phase locking in this simulation despite the spike train’s perfect theta resonance at 2 Hz. (D) Histogram showing the distribution of *P*-values (obtained using the same procedure as in C) for all simulated aperiodic signals and associated spike trains. Only 0.9% of the spike trains had a *P*-value < 0.05, indicating a negligible influence of theta resonance of individual neurons on theta-phase locking. Hence, our findings on residual theta-phase locking in the absence of clear oscillations is most likely explained by “subthreshold” oscillatory activity that was missed by our method of oscillation detection.

Overall, we would thus suggest keeping our results on theta-phase locking during periods without clear oscillations detected by our oscillation-detection algorithm, as we cannot exclude that there is residual theta-phase locking in the periods where the algorithm does not detect theta oscillations. We believe that showing these results and discussing their limitations maximizes transparency, which may be useful for future studies on human theta-phase locking.

Reviewer #1, references

1. Fell, J. et al. *Medial temporal theta/alpha power enhancement precedes successful memory encoding: evidence based on intracranial EEG.* *J. Neurosci. Off. J. Soc. Neurosci.* 31, 5392–5397 (2011).
2. Fell, J. et al. *Human memory formation is accompanied by rhinal-hippocampal coupling and decoupling.* *Nat. Neurosci.* 4, 1259–1264 (2001).
3. Manns, J. R., Zilli, E. A., Ong, K. C., Hasselmo, M. E. & Eichenbaum, H. *Hippocampal CA1 spiking during encoding and retrieval: Relation to theta phase.* *Neurobiol. Learn. Mem.* 87, 9–20 (2007).
4. Douchamps, V., Jeewajee, A., Blundell, P., Burgess, N. & Lever, C. *Evidence for Encoding versus Retrieval Scheduling in the Hippocampus by Theta Phase and Acetylcholine.* *J. Neurosci.* 33, 8689–8704 (2013).
5. Poulter, S., Lee, S. A., Dachtler, J., Wills, T. J. & Lever, C. *Vector trace cells in the subiculum of the hippocampal formation.* *Nat. Neurosci.* 24, 266–275 (2021).
6. Jacobs, J., Kahana, M. J., Ekstrom, A. D. & Fried, I. *Brain Oscillations Control Timing of Single-Neuron Activity in Humans.* *J. Neurosci.* 27, 3839–3844 (2007).

Reviewer #2, general comment

In the manuscript, authors provide a thorough analysis of theta phase-locking in human via deep intracranial recordings during memory formation and retrieval. This aims to provide a comprehensive account of the relationship between theta and neuronal spikes through latest conceptual and methodological approach to describe oscillations. It is meticulous, but while the analysis appears well conducted, I find the results difficult to interpret for non-specialists, especially the part on phase locking during aperiodic epochs. While it is partly based on recent methodology showing promising framework for interpreting oscillations, it needs more contextualization, as it may at first read appear paradoxical, and thus it is difficult to grasp what the finding is.

Next, the phase locking analysis reveals no strong and important relationships between theta and spikes depending on learning, and the negative result is important. It contrasts with recent findings suggesting theta modulation with leaning. Thus, perhaps the strongest message of the paper, is that despite strong and robust attempt to reveal differences between successful and unsuccessful trials during encoding or recall (fig 4), there were no differences in any of the measures tested. The paper may gain by stressing out this even more clearly, although authors put forward as one possible explanation, the difference of task structure compared to other tasks.

Response: Thank you very much for your detailed review. We have revised our manuscript to improve clarity and better contextualize our findings. Please find our point-by-point responses below, including responses to your comments on theta-phase locking as a function of aperiodic slope. Following your suggestions, we now also highlight the null findings more strongly.

Reviewer #2, comment #1

1. Many neurons have theta phase locking, and this is stronger during high theta. This is observed for all periods (baseline, encoding, recall). How can Rayleigh z-values vary so much across periods from 10 to 200 to 1000 between Fig2E and G? I understand that the duration varies but why didn't authors use random time duration samples for baseline that match duration of encoding or recall?

Response: Thanks a lot for raising this question. The Rayleigh z -value depends heavily on the number of spikes. Long baseline periods (up to ~1 hour in our case) can thus lead to very high Rayleigh z -values.

Based on your comment, we now use pairwise phase consistency (Vinck et al., Neuroimage, 2010) to quantify phase locking in most of our analyses, and we have updated our results and figures accordingly. The advantage of pairwise phase consistency is that it does not depend on the number of spikes, which we illustrate using simulations in a revised supplementary figure (Fig. S11). Pairwise phase consistency values are comparable across the different conditions and do not require subsampling. The only disadvantage of pairwise phase consistency is the time that it needs to be computed, which is why we keep the Rayleigh z -value for a few analyses with a very high number of spikes (e.g., Fig. 2D).

Using pairwise phase consistency (with values ranging between -1 and 1), theta-phase locking is now easier to compare between different conditions. For example, in the revised Fig. 2E (copied below), pairwise phase consistency is consistently around 0.05 for the high-power

condition and around 0.01 for the low-power condition, irrespective of trial period. These values for pairwise phase consistency are in a range that is typical for neuronal recordings (e.g., Vinck et al., NeuroImage, 2010; Roux et al., eLife, 2022).

Figure 2. Neural recordings and general theta-phase locking. ... (E) Average theta-phase locking during high and low theta power across units, quantified with pairwise phase consistency (PPC). Results are shown for baseline (left; entire session except for encoding and recall periods), encoding (middle), and recall (right). We compared pairwise phase consistency between high and low power using *t*-tests and ranked the empirical *t*-values within surrogate *t*-values (plots to the right). *P*-values are Bonferroni corrected for three tests.

Reviewer #2, comment #1, continued

Is there a way to identify periods of theta oscillation, or a periodicity in an agnostic way to the epoch of the task and then quantify the probability of these oscillations or aperiodicity to be associated to specific task segments given overall duration and relative duration of each period? Or alternatively, to construct a distribution randomly sampling equal population size during baseline as for the other epochs?

Response: Thanks for these suggestions. In the revised manuscript, we have included new results on the modulation of aperiodic activity and theta oscillations by different task periods. As suggested, we subsampled all periods to the same duration for this analysis. We did not find any differences in aperiodic activity and theta oscillations during different task periods. We show these results in Fig. S15, copied below.

Figure S15. Aperiodic slope and oscillation presence during encoding, recall, and baseline. Analysis of the steepness of aperiodic slopes and the percentage of theta oscillations during different task periods. Recall and baseline periods are subsampled to match the durations of the encoding periods. (A) Steepness of the aperiodic slope during encoding, recall, and baseline (left). There were no significant differences in slope steepness between

the three conditions (right; two-sided surrogate analysis with session-label swapping: encoding versus recall, $t(26) = -0.419$, $P_{\text{corr.}} = 1$; encoding versus baseline, $t(26) = -1.014$, $P_{\text{corr.}} = 0.946$; recall versus baseline, $t(26) = -0.752$, $P_{\text{corr.}} = 1$; Bonferroni corrected for three tests). (B) Percentage of detected oscillations during encoding, recall, and baseline (left). There were no significant differences between the three conditions (right; two-sided surrogate analysis with session-label swapping: encoding versus recall, $t(26) = -1.422$, $P_{\text{corr.}} = 0.487$; encoding versus baseline, $t(26) = -0.121$, $P_{\text{corr.}} = 1$; recall versus baseline, $t(26) = 2.248$, $P_{\text{corr.}} = 0.087$; Bonferroni corrected for three tests).

Reviewer #2, comment #2

2. Authors describe a variation of theta phase locking across regions. There is a higher percentage of cells with phase locking in the parahippocampal cortex compared to hippocampus. Here authors suggest that 1) because the lower phase locking in other regions stems from more variability in theta phase locking (i.e. Rayleigh tests are low because coupling of spikes with phases were more dissimilar), 2) this means an increased potential for phase coding (note here the use of coding instead of locking). I presume here that authors suggest that phase coding lies within the variability in the relationship between the theta phase and the spikes emitted, but proven that within each cell, this relationship is preserved across the coded events, and that different cells have different phase codes. If the individual cell does not preserve its phase lock within a session for an event, then it is hard to qualify as "codes" these random relationships between the phase and the spikes. So, I am not sure I understand the author's rationale, given that a code should be reproduced to be decoded.

Response: Thank you for this important comment. We made this statement because weaker theta-phase locking enables, broadly speaking, a greater potential for theta-phase coding. We agree with you that theta-phase coding requires stimulus-specific theta-phase locking (for example, spiking at 0° whenever stimulus A is presented, and spiking at 90° whenever stimulus B is presented), because otherwise the code would not be temporally stable. However, when analyzing theta-phase locking over long time periods and across stimuli (as we do it in our study), then stronger theta-phase locking reduces the possibility of theta-phase coding that can distinguish between different stimuli.

We have created a supplementary figure (Fig. S12, copied below) to better explain the possible relationships between theta-phase locking and theta-phase coding. In this supplementary figure, we illustrate three different scenarios of how theta-phase locking can relate to theta-phase coding. In the first scenario ("pure theta-phase locking"), there is theta-phase locking but no theta-phase coding. This is because the neuron spikes at a similar theta phase over time, irrespective of the stimulus that is presented. Hence, the theta phases at which the neuron spikes do not contain information about the stimuli. The stronger a neuron's theta-phase locking to a specific theta phase, the lower the probability that it can represent different stimuli through these very similar theta phases. This is what we meant with our suggestion in the manuscript that stronger theta-phase locking in the parahippocampal cortex gives less room for potential theta-phase coding.

In the second scenario of our new supplementary figure, there is theta-phase coding but no theta-phase locking ("pure theta-phase coding"). This is because the neuron spikes at many different theta phases that are not biased toward a particular theta phase over time. However,

the neuron spikes consistently at the same theta phase when different stimuli are considered separately from each other. Hence, the neuron shows theta-phase locking regarding individual stimuli. Accordingly, we agree with your comment that theta-phase coding requires theta-phase locking per stimulus. However, when analyzed across stimuli, the neuron does not show theta-phase locking.

In the third scenario, theta-phase locking and theta-phase coding occur simultaneously (“combined theta-phase locking and coding”). This is because the theta phases are overall biased toward a particular theta phase, even when analyzed across different stimuli, while also being systematically different for different stimuli. We have added this supplementary figure to the revised manuscript, and we hope that it clarifies the possible relationships between theta-phase locking and theta-phase coding.

Figure S12. Illustration of the concepts of theta-phase locking, theta-phase coding, and theta-phase shifts. (A) Pure theta-phase locking. A neuron activates at similar phases over time, for example at the oscillatory troughs. The theta phases at which the neuron spikes do not differ between different stimuli, and they do not shift between encoding and retrieval. Each vertical line represents an action potential. Schematic polar histogram on the right shows the phase distribution of the action potentials, illustrating the presence of theta-phase locking. **(B) Pure theta-phase coding** in the absence of theta-phase locking. For specific stimuli, the neuron spikes at similar theta phases over time. When pooling across the different stimuli, the neuron spikes at different theta phases because of which there is no overall theta-phase locking. Each vertical line represents an action potential, and different colors indicate different stimuli during which the action potentials occur. Schematic polar histogram on the right shows that different stimuli occupy different angles in phase space. **(C) Combined theta-phase locking and coding.** A neuron generally spikes around the oscillatory trough, leading to theta-phase locking, while also spiking at different theta phases for different stimuli, thus supporting a stimulus-specific phase code.

Reviewer #2, comment #2, continued

This becomes even more difficult to understand if the relationship changes between learning and retrieval. Is that to relate to the last paragraph for which the majority of cells did not alter code between learning and retrieval?

Response: Thank you for this comment. We agree with you that strong phase shifts between encoding and retrieval impede stimulus-specific phase coding, as the stimulus-specific phase code is not stable over time. We have added this information to our new supplementary figure illustrating the concepts of phase locking, phase coding, and phase shifts (Fig. S12D).

Strong theta-phase shifts between encoding and retrieval impede stimulus-specific phase coding as the stimulus-specific phase codes would not be stable over time.

We also agree that strong theta-phase locking, as observed in our study, reduces the likelihood of large theta-phase shifts between encoding and retrieval. Accordingly, we believe that the high prevalence of theta-phase locking in our data is a possible reason for why we found only a small number of neurons with weak theta-phase shifts between encoding and retrieval.

To better explain the possible relationships between theta-phase locking and theta-phase shifts, we have added them to our supplementary figure where we illustrate relationships between phase locking, phase coding, and phase shifts (Fig. S12). As for the relationship between theta-phase locking and theta-phase coding (see above), we consider three possible scenarios. In the first scenario (“pure theta-phase locking”), there is only theta-phase locking without additional shifts in theta phases between encoding and retrieval. In the second scenario (“pure theta-phase shifts”), the theta-phase shifts are very strong (180°), preventing general theta-phase locking to occur when pooling across all encoding and retrieval periods. In the third scenario (“combined theta-phase locking and shifts”), the theta-phase shifts are weaker so that they can co-occur with theta-phase locking. Our empirical findings are mostly in line with the third scenario of combined theta-phase locking and shifts.

Figure S12. Illustration of the concepts of theta-phase locking, theta-phase coding, and theta-phase shifts. (A) Pure theta-phase locking. A neuron activates at similar phases over time, for example at the oscillatory troughs. The theta phases at which the neuron spikes do not differ between different stimuli, and they do not shift between encoding and retrieval. Each vertical line represents an action potential. Schematic polar histogram on the right shows the phase distribution of the action potentials, illustrating the presence of theta-phase locking. ... (D) Pure theta-phase shifts in the absence of theta-phase locking. The theta phases during encoding are very different from the theta phases during recall. Each vertical line represents an action potential, and the different shades of gray represent encoding and retrieval processes during which the action potentials occur. All theta phases combined, there is no unimodal theta-phase locking, as illustrated in the polar histogram on the right. Strong theta-phase shifts between encoding and retrieval impede stimulus-specific phase coding as the stimulus-specific phase codes would not be stable over time. (E) Combined theta-phase locking and shifts. A neuron generally spikes around the oscillatory trough, leading to theta-phase locking, while also shifting its theta phase between encoding and retrieval, thus leading to a separation of these two conditions.

Reviewer #2, comment #3

3. If I understand correctly, the aperiodic component (rather than activity) is the overall slope of the fit (in parallel to oscillations, but once the oscillatory peak has been removed), whose steepness depends on the relative ratio of low frequency over high frequency. A high slope means more low frequency compared to high frequency, while no slope would mean all frequencies at same power. Changes in slope steepness from one moment to another, could mean that theta oscillations got lower (shifted from high to low theta), or even that higher oscillations got suppressed (the latter fitting with lower firing rate is one assumes high oscillations are linked to firing rate).

The section aiming to describe the results linked to aperiodic epochs is really hard to follow for non-specialists. It is difficult to understand that it is not exactly oscillatory and non-oscillatory activity that authors compare in the section, but whether the phase locking is different depending on the steepness of the slope of the aperiodic components. It might be necessary to explain more clearly whether there is a difference between periods without theta oscillations and “aperiodic” epochs, because otherwise, it is difficult to understand how one can obtain phase locking on an arrhythmic period given that the phase is the relationship between the oscillation and spike calculated based on the generalized phase as shown in Figure 2B.

Response: We apologize for the lack of clarity here. We have revised the manuscript to enhance clarity and emphasize that analyzing the effects of aperiodic activity on theta-phase locking is independent of analyzing the effects of clear theta oscillations on theta-phase locking.

Following previous work (Donoghue et al., Nature Neuroscience, 2020), local field potentials consist of two distinct components, periodic activity and aperiodic activity. Aperiodic activity is present at any point in time and results from non-rhythmic neural activity. On top of aperiodic activity, local field potentials can also contain periodic activity (i.e., oscillations).

Properties of the aperiodic activity (such as its slope and offset when analyzing the power spectrum as described by Donoghue et al., Nature Neuroscience, 2020) can change irrespective of whether periodic activity is present or not. Hence, aperiodic activity with high and low slopes (as analyzed in Figure 3E) can occur both during periods with clear theta oscillations and during periods without clear theta oscillations.

We have added these explanations to the revised manuscript to better guide readers through the analysis of theta-phase locking as a function of aperiodic slopes (lines 280–299).

Local field potentials are composed of two components, aperiodic (non-oscillatory) and periodic (oscillatory) activity (57), both of which may reflect and influence different types of neural processing (71). Both components can also lead to changes in absolute theta power (6, 57) (Fig. S13) and may thus underlie our observation that higher theta-phase locking emerges during periods of elevated theta power (Fig. 2E). We thus aimed at understanding how theta-phase locking was influenced by aperiodic, non-oscillatory and periodic, oscillatory activity. In brief, we used Spectral Parameterization Resolved in Time (SPRiNT) to characterize aperiodic activity (58), and we used Bycycle to identify periodic theta oscillations (59). These methods allowed us to perform two independent analyses where we grouped neuronal spikes (I) as a function of varying properties of aperiodic activity and (2) according to whether they occurred in the presence or absence of clear theta oscillations detected by Bycycle (Fig. 3A–C).

We first asked whether theta-phase locking varied in strength as a function of non-oscillatory, aperiodic activity, which summarizes neural activity that does not need to arise from any regular, rhythmic process (57). One can fit and parameterize aperiodic activity by means of two major parameters: the aperiodic slope and aperiodic offset of the power spectrum (57). Here, we focused on aperiodic slopes, which are thought to reflect the balance between neural excitation and inhibition (71). Specifically, we tested whether theta-phase locking was stronger during periods with higher (i.e., more negative) slopes as compared to those with lower slopes. This analysis was, by design, independent of the question whether theta-phase locking was stronger during periods of clear theta oscillations (see the next section). Accordingly, periods with high versus low aperiodic slopes occurred both during periods with clear theta oscillations and during periods without them (Fig. S14).

To empirically show that the steepness of aperiodic slopes is independent of whether theta oscillations are present, we quantified the prevalence of high and low slopes in the presence versus absence of theta oscillations in our data. We found that high and low aperiodic slopes occurred at similar percentages in the presence and absence of detected theta oscillations (Fig. S14, copied below).

Figure S14. Prevalence of high and low aperiodic slopes in the presence and absence of clear theta oscillations. Bar graph shows the prevalence of high and low aperiodic slopes (median split within each session) during periods with and without clear theta oscillations (mean \pm standard error of the mean). Theta oscillations were detected in approximately 38% of the signal, with similar proportions (~19%) occurring during high versus low aperiodic slopes (two-sided paired *t*-test, $t(26) = 0.646$, $P = 0.524$). Accordingly, the prevalence of high and low aperiodic slopes was also similar during periods without theta oscillations (two-sided paired *t*-test, $t(26) = -0.693$, $P = 0.494$).

In the revised manuscript, we now also describe more clearly that periods without clear theta oscillations are not “aperiodic epochs,” as we cannot exclude that they contain subthreshold theta oscillations. This is because oscillation-detection algorithms depend on many parameters and can miss theta oscillations that are less pronounced. We thus do not use the term “aperiodic epochs.” Instead, we use terms like “periods without clear theta oscillations” to refer to time periods in which our oscillation-detection algorithm did not identify theta oscillations. We have added language to the manuscript to better explain the analysis of theta-phase locking in the presence versus absence of clear theta oscillations (lines 330–338).

To identify theta oscillations, we isolated individual cycles in the 1–10 Hz filtered signal and used Bicycle to categorize each as being part of an oscillation or not (Fig. 3A). We note that this practical separation of the local field potential into periods with and without clear theta oscillations is non-trivial and not binary, as it reflects a trade-off between specificity and sensitivity determined by the parameters of the chosen oscillation detection algorithm. Here, we implemented this distinction to enable a relative comparison of theta-phase locking between periods with stronger (suprathreshold) versus weaker (subthreshold) theta oscillations. This does not exclude the presence of weaker theta oscillations during “periods without clear theta oscillations,” which are presumably responsible for residual theta-phase locking in those time periods.

We hope these changes will help readers better understand that the analyses of aperiodic activity and theta oscillations on theta-phase locking are independent of each other, and that periods without clear theta oscillations are not equivalent with “aperiodic epochs.” We would be happy to make any additional changes if needed.

Reviewer #2, comment #3, continued

Likewise, It is difficult to put together the 2 sentences in the same framework: Line 265: “We found that theta-phase locking was stronger for neuronal spikes associated with high aperiodic slopes as compared to those associated with low aperiodic slopes’ And Line 295 “During baseline and recall (but not during encoding) we found that phase locking was stronger for neuronal spikes occurring during theta oscillation”

Response: We apologize for the lack of clarity here. These analyses simply show that theta-phase locking is stronger (1) in the presence of higher aperiodic slopes; and (2) in the presence of clear theta oscillations. As explained above, these two analyses are independent of each other. We now state this explicitly in the revised manuscript (lines 284–289).

We thus aimed at understanding how theta-phase locking was influenced by aperiodic, non-oscillatory and periodic, oscillatory activity. In brief, we used Spectral Parameterization Resolved in Time (SPRiNT) to characterize aperiodic activity (58), and we used Bycycle to identify periodic theta oscillations (59). These methods allowed us to perform two independent analyses where we grouped neuronal spikes (1) as a function of varying properties of aperiodic activity and (2) according to whether they occurred in the presence or absence of clear theta oscillations detected by Bycycle (Fig. 3A–C).

We added a new panel to Fig. 3 to illustrate the two analyses (Fig. 3C, copied below).

(C) Schematic illustrating the two independent analyses of theta-phase locking as a function of aperiodic slope and as a function of clear theta oscillations detected by Bycycle. Top: high aperiodic slopes result from relatively increased activity in lower frequencies and relatively decreased activity in higher frequencies. These increases and decreases in activity need not result from any rhythmic process. Bottom: oscillations are more likely to be detected if consecutive cycles are similar in amplitude and frequency.

Reviewer #2, comment #4

4. I was wondering if steepness of the slope and theta phase locking could be related to putative excitatory principles cells or inhibitory interneurons. Could be interesting to test the phase locking strength depending on firing rate?

Response: Thank you for this suggestion, based on which our revised manuscript contains new results on theta-phase locking in putative interneurons and pyramidal cells (lines 198–201).

When differentiating single units into putative interneurons and pyramidal cells (69, 70), we found a higher percentage of theta-phase locking in putative interneurons compared to pyramidal cells (94.1% and 83.8%, respectively; two-sided surrogate analysis with unit-label shuffling, $\Delta = 10\%$, $P = 0.029$; Fig. S9).

We have added a supplementary figure to describe this finding in greater detail. Following previous work (Gast et al., Clinical Neurophysiology, 2016; Ison et al., Journal of Neurophysiology, 2011), we differentiated between putative interneurons and pyramidal cells using the spike width of their average action potential (panel A). We found that putative interneurons had a higher percentage of theta-phase locking neurons as compared to putative pyramidal cells (panel C). In both cell populations, firing rates were higher during periods in which the local field potential exhibited lower aperiodic slopes (panel D).

Figure S9. Theta-phase locking of putative interneurons and putative pyramidal cells. (A) Distribution of spike width across single units ($n = 536$). Spike width was measured from the minimum to subsequent maximum of the average waveform. We classified single units with a spike width ≤ 0.65 ms as putative interneurons and > 0.65 ms as putative pyramidal cells (2, 3). (B) Spike width was negatively correlated with pairwise phase consistency (PPC; Spearman's $\rho = -0.126$, $P = 0.003$, $n = 536$ single units). (C) The percentage of theta-phase locking units was higher for putative interneurons than putative pyramidal cells (two-sided surrogate analysis with unit-label shuffling, $\Delta = 10\%$, $P = 0.029$). We note that this difference may be (partially) explained by higher baseline firing rates in putative interneurons compared to pyramidal cells. (D) Firing rate as a function of aperiodic slope (as in Fig. 3D) for putative interneurons (left) and pyramidal cells (right). Each line, one unit. Black lines connect median firing rates. In both cases, firing rates were higher during periods of lower aperiodic slopes. Bottom: Empirical t -values of paired t -tests comparing firing rates between conditions (vertical lines), along with distributions of surrogate t -values (gray histograms). Surrogates were obtained by randomly shuffling the condition labels. PPC, pairwise phase consistency.

Regarding your question on firing rates, we found that neurons with higher firing rates showed weaker theta-phase locking (Spearman's $\rho = -0.244$, $P < 0.001$). We believe that firing rate is not the best parameter for differentiating between putative interneurons and putative pyramidal cells in human single-neuron recordings, which is why we opted for the spike-width criterion instead (see Fig. S9 above), following previous human single-neuron studies (Gast et al., Clinical Neurophysiology, 2016).

Reviewer #2, comment #5

5. The thorough analysis of phase during encoding and retrieval suggests that there isn't any difference between correct and error trials. The analysis of theta phase locking shifts revealed a very modest shift between encoding and retrieval both in proportion and shift amplitude. There is some modulation considering comparison between epochs through other measures (aperiodic, high theta), but all results are very modest. In sum, the paper, actually provides some grounds to reconsider the robustness of phase-locking codes as a meaningful physiological signal linked to memory or information processing. Many results are modest or negative, and but provided that the methods are very robust, the message of the paper may gain in clarity if the lack of conclusiveness is better stated.

Response: Following your comment, we now state more clearly that further investigations are needed to better understand how theta-phase locking may support human memory and information processing. For example, at the end of the Introduction (lines 90–93), we now write:

These findings provide new insights into the analysis and properties of human theta-phase locking; may help identify the relevance of theta-phase locking for human memory processes; and may trigger new investigations into the question of how theta-phase locking supports information processing in the human brain.

Furthermore, in the first paragraph of the Discussion (lines 553–557), we now write:

Together, these results help identify the electrophysiological determinants and functional properties of theta-phase locking in humans. This contributes to a better understanding of the relationship between single neurons and neuronal populations in the human brain and may lead to new investigations into the relevance of theta-phase locking in human memory and information processing.

We now also state more clearly that the relationship between theta-phase locking and memory performance should be reconsidered (lines 611–619).

Here, we did not find stronger theta-phase locking during successful memory performance, neither during encoding nor during retrieval. We tested for these memory effects in several subgroups of the data, for example only including spikes in the presence of clear oscillations or only including object-responsive cells, and we did not find differences in theta-phase locking as a function of memory performance in any of these groups (Fig. 4). These findings differ from the above-mentioned studies that described a positive relationship between the strength of theta-phase locking and the successful encoding of images and words (37, 51). This may offer a starting point for reconsidering the relationship between theta-phase locking and human memory, triggering new investigations into the conditions under which theta-phase locking is related to successful memory performance.

Reviewer #2, comment #6

6. What are the measures shown in Fig1C and E? distribution of the average percent correct for each session? If a recall is a single shot, how is the distribution in figure 1E obtained (3 out of 3)?

Response: We apologize for the lack of clarity here. We have improved the caption of Fig. 1 to better explain how the distributions in panels C and E were obtained.

Specifically, Fig. 1C shows the distribution of object-recall performance across sessions. Each session contributes one value to the histogram. We have updated the figure caption as follows:

(C) Histogram of session-wise memory performance during object recall (left) and improvement of object-recall memory performance over time, i.e., during the first half of all trials versus the second half of all trials (right). We computed object-recall performance in each session as the ratio of the number of successfully recalled objects divided by the total number of object recalls. For instance, in session number one, the participant successfully recalled 9 of 33 objects, resulting in an object-recall performance of 0.273.

Furthermore, Fig. 1E shows the distribution of location-recall performance across trials. Each trial contributes one value to the histogram. We have updated the figure caption as follows:

(E) Histogram of trial-wise memory performance during location recall (left) and improvement of location-recall memory performance over time (right). We computed location-recall performance in each trial as the drop error (Euclidean distance) between the participant's response location and the correct location of the object, normalized by the possible distribution of drop errors. The closer to 1, the better memory performance. Red dotted line, chance performance. Gray lines, session-wise data; blue line, average.

Reviewer #3, general comment

In the submission, Guth et al. present interesting and detailed results on the phase-locking of single neurons in the human MTL to local field potentials. The general research question, approach, and potential implications are timely and relevant for a better understanding of MTL electrophysiology and could lead to a more profound knowledge of human spatial memory functioning.

The experimental setup and data are rare, and the methods and analyses sound. A key novelty of the study is an innovative generalized-phase approach to investigate phase-locking in a frequency-adaptive manner. As discussed in the manuscript, this is a major difference from previous studies on single-neuron phase-locking.

From my understanding, the generalized-phase algorithm ensures the centering of complex plane representations by excluding lower frequencies. In the original manuscript introducing the generalized-phase approach, an important initial step is to exclude this low-frequency content through filtering in the 5–40 Hz frequency range. In this previous work, low frequencies were excluded for these technical reasons but also because they are thought to reflect slow global changes such as arousal¹.

Theta frequencies might vary in humans, and MTL oscillations are not continuous. Previous literature investigated theta-phase locking of single neurons in the 3–8 Hz² frequency range, distinguished between “slower” (2–5 Hz) and “faster” (5–9 Hz) theta³, and included slower frequencies in the 1–4 Hz range, which were discussed as delta oscillations. Given the 1–10 Hz frequency range in the present submission, I would consider discussing the findings as phase-locking to non-oscillatory LFP equally justified. However, in the manuscript, the authors present and discuss the results as theta-phase locking (Title, Abstract, Discussion).

One could argue that theta-phase locking would require the presence of an oscillation in the ~3–10 Hz range to draw comparisons to previous human and animal work. The absence of slower fluctuations is needed to estimate the phase of these oscillations correctly, given the chosen generalized phase approach at 1–10 Hz in the present submission. In other words, theta oscillations need to be present and become the most prominent fluctuation to estimate theta-phase locking appropriately.

Otherwise, phase locking to non-oscillatory LFP fluctuations is investigated, which is still interesting and novel, especially following literature in bats⁴, but the present submission does not distinguish phase-locking to oscillations or non-rhythmic fluctuations. The manuscript includes analyses on phase-locking, dependent on the presence of oscillations, but that does not guarantee that these oscillations are the most prominent fluctuations within these time windows. Therefore, the influence of slower fluctuations potentially biases the phase estimates, and related results cannot be ruled out.

These limitations temper my general enthusiasm for the manuscript, which I consider to be excellent apart from these comments. Yet, the points I raised above might considerably impact the results and conclusions drawn in the submission, which need to be clarified before publication. Please find my specific comments and suggestions below:

Response: Thank you very much for your helpful evaluation of our manuscript. Please find our detailed point-by-point responses below, including responses to your comments on the comparison of 1–10 Hz phase locking versus 3–10 Hz phase locking, the prevention of low frequency biases, and the use of the term “non-oscillatory phase locking.”

To foreshadow our detailed responses, we have taken the following approaches to address your main points:

- We have re-performed some of our 1–10 Hz analyses for the 3–10 Hz frequency range as well as for 2–5 Hz and 6–9 Hz. This includes a direct comparison of theta-phase locking strength in the 1–10 Hz frequency range versus the 3–10 Hz frequency range. We observe similar effects across the different bands. This suggests that including the 1–3 Hz range does not change our theta-phase locking results in a qualitative way, which speaks in favor of its inclusion to better reflect the raw signal and avoid discarding parts of low theta.
- We agree that lower-frequency activity can shift the complex plane representations. To the best of our knowledge, this is common to any frequency range. In the revised manuscript, we thus (1) explain that estimating generalized phase relies on a balance between choosing a particular frequency range and accepting that lower frequencies in that band influence the phase estimates more strongly than higher frequencies; (2) provide new analyses and simulations to show that generalized phase estimates in the 1–10 Hz frequency range are valid in our data; (3) added some discussion of this topic to the limitations section of our manuscript; and (4) performed new analyses indicating that 1–3 Hz activity is not clearly related to arousal in our awake human participants.
- We suggest keeping the term “theta-phase locking” despite our broad 1–10 Hz frequency range. This choice is based on previous intracranial studies in humans that indicate the existence of both high theta (6–10 Hz) and low theta (1–5 Hz). Defining theta as 3–10 Hz would exclude parts of the low-theta range. Furthermore, although we agree with you that human theta oscillations are dynamic in frequency and unstable over time, it remains unclear to us whether “non-oscillatory phase locking” is a good term given that the absence of oscillations is difficult to prove. Nevertheless, to acknowledge other views on this topic (e.g., Eliav et al., Cell, 2018), we have added language to the Discussion stating that our 1–10 Hz results are also in line with the idea of non-oscillatory phase locking.

We would be happy to make any further changes if the Reviewer or Editor feels strongly about these points.

Reviewer #3, comment #1

Major:

1.) Please clarify how the generalized-phase approach can prevent bias from low frequencies within the chosen 1–10 Hz range (as opposed to the 5–40Hz filter of the original work), affecting the phase estimates in the present submission.

Response: Thank you for raising this important point. In the revised manuscript, we have taken three major actions to address this point.

Action 1. We emphasize that we chose the theta-frequency range as 1–10 Hz because this band includes both high and low theta-frequency ranges (6–10 and 1–5 Hz, respectively), both of which have been demonstrated to be important for human memory and navigation in previous studies (e.g., Lega et al., Hippocampus, 2012; Bush et al., PNAS, 2017; Miller et al., Nature Communications, 2018; Goyal et al., Nature Communications, 2020). Hence, including low

frequencies (1–3 Hz) is a feature of our approach, not a bug. These frequencies would only represent a bias, if they were not part of the frequency-band-of-interest.

Accordingly, the original generalized phase approach (Davis et al., Nature, 2020) excluded lower frequencies (1–5 Hz) because they were not of interest and would, thus, have biased the phase estimates in the 5–40 Hz range. In contrast, as we intentionally chose the 1–10 Hz frequency band, fluctuations in the 1–5 Hz range do not represent a mis-centering in the complex plane. They contribute to accurate phase estimates within this frequency range by validly influencing the signal within the complex plane. Instead, we removed DC offsets and slow drifts below 1 Hz to ensure centered complex signals in the 1–10 Hz frequency range.

In the revised manuscript, we provide a better explanation of choosing the 1–10 Hz frequency range for investigating theta-phase locking and describe that the generalized phase approach leads to valid phase estimates within this frequency range (for analyses and simulations, see Action 2). We also explicitly mention that our frequency range is different from the 5–40 frequency range in Davis et al., Nature, 2020 (lines 145–182).

We also extracted the low-frequency component of the microelectrode data in a broad theta-frequency range (1–10 Hz) using a generalized phase approach (56) to characterize the timing relationship between the neurons' spiking activity and the theta rhythm. We opted for this approach with a broad filter band of 1–10 Hz as our visual inspection of the data identified many periods in which the local field potential showed fluctuations and oscillatory activity with rapidly changing frequencies, similar to previous observations of non-stationary theta oscillations in bats and humans (16, 26, 27, 65). Our preliminary analyses with the filter-Hilbert method including narrow-band filtering often failed to fit these fluctuations neatly, and we thus decided to treat them as a coherent entity and opted for the generalized phase approach instead. This led to a good fit between the original and the filtered signal in the theta-frequency range (e.g., Fig. 2B; Fig. S2).

In addition to this better fit of the raw signal, our choice of using a broad 1–10 Hz frequency range was motivated by several previous studies using intracranial neural recordings in humans. These studies showed that behavior-associated theta oscillations in humans occur across higher and lower theta frequencies, which extend beyond the traditional 4–8 Hz theta band and are often referred to as “high theta” (6–10 Hz) and “low theta” (1–5 Hz) [e.g., (16, 28, 30, 31, 66)]. For example, previous work found that increased low-theta power at 1–3 Hz is related to successful memory encoding (30, 54); that 3-Hz oscillations are often present during virtual spatial navigation (16, 31); and that high- and low-theta power increases before the onset of virtual movement (66). Hence, to not exclude any components of high theta and low theta, we opted for the 1–10 Hz frequency range and refer to fluctuations within this range as “theta” throughout the manuscript. We acknowledge though that human “theta” has been defined in various ways [e.g., as 4–7.5 Hz (67) or as 4–8 Hz (6)] and that the 1–10 Hz band includes parts of the traditional delta and alpha bands (67).

To obtain phase estimates within this broad frequency band, we used the recently developed generalized phase approach (56). In brief, the generalized phase approach involves filtering in a broad band (here, 1–10 Hz), computing the analytic signal using the Hilbert transform, estimating the phase from the analytic signal, and interpolating periods with high-frequency intrusions where phase progression reverses direction or phase progresses with a frequency <1 Hz (Fig. S3). Due to the 1/f-like shape of the power spectrum with stronger power in lower as compared to higher frequencies, generalized phase estimates are, by design, more strongly influenced by lower than higher frequencies. In comparison to the 5–40 Hz band of the original study using generalized phase (56), our study also included frequencies between 1–5 Hz to not miss low-theta effects. We thus performed control analyses to examine whether our frequency band might add effects related to arousal and whether it might be more prone to low-frequency intrusions. Analyzing the frequency distributions during the distractor period, we did not find evidence for the idea that lower arousal was related to a higher prevalence of 1–5 Hz activity (Fig. S4A).

Furthermore, control analyses and simulations did not indicate that potential low-frequency intrusions biased the 1–10 Hz phase estimates in a relevant way (Figs. S2, S3, and S5). Taken together, computing generalized phase within the 1–10 Hz frequency range appeared as a valid approach to investigate theta-phase locking in our data.

Action 2. We performed new analyses and computer simulations to demonstrate the robustness of the generalized phase approach in handling the 1–10 Hz frequency range.

We first estimated the power spectra of our local field potentials, showing that signals below 1 Hz were of minimal amplitude. This demonstrates that our band-pass filter for the 1–10 Hz range worked well, minimizing the potential for low-frequency intrusions from signals below this band. We added this result as Fig. S3G, copied below.

(G) To examine whether our band-pass filter effectively removed activity below 1 Hz, we computed the grand-average power spectrum across all wires, showing negligible power beyond the 1–10 Hz frequency range. Black line, mean across microwires; gray shaded area, standard error of the mean across microwires.

Next, we visually inspected random selections of the analytic signals and observed no relevant phase biases. To provide the Reviewer and readers with some intuition, we included the complex plane representations of our examples in Fig. S2, copied below.

Figure S2. Examples of neuronal theta-phase locking in the human medial temporal lobe. **Left:** Examples of raw (black) and filtered (1–10 Hz; colored) local field potentials from different experimental sessions. Spike trains are shown as vertical lines below the local field potentials. Color indicates the instantaneous theta-phase angle estimated using a generalized phase approach (as in Fig. 2B), and gray areas indicate the 1.5-s encoding period. **Right:** Complex plane representation of the analytic signal, suggesting no relevant phase biases due to potential

low-frequency intrusions. AMY, amygdala; EC, entorhinal cortex; HC, hippocampus; PHC, parahippocampal cortex.

Next, we analyzed whether generalized phases had a preference toward a particular phase and added this result to the revised manuscript (Fig. S3, H and I). To this end, we pooled the instantaneous phase estimates of all samples of all wires and calculated the mean resultant vector length. The mean resultant vector length was 0.004, indicating a quasi-uniform distribution of phases and suggesting that our phase estimates were not biased (panel H). Similarly, when we plotted all analytic signals in the complex plane (pooled across all wires), we did not find a relevant offset from the center (panel I).

(H) To rule out a general bias in our phase estimates, we examined whether the instantaneous phases of all samples, pooled across all 502 wires, were distributed uniformly. A mean resultant vector length close to zero confirmed a practically uniform distribution of phase estimates. (I) 2D histogram of all analytic signals in the complex plane, showing a good centering of our signals in the complex plane. Accordingly, the means of the imaginary and real parts of the signals were close to zero (mean \pm SEM: -0.034 ± 0.001 and $-0.035i \pm 0.001i$, respectively).

Moreover, we performed an analysis where we locally re-centered the complex plane representations within 10-second sliding-time windows. We compared these re-centered phases with the original phases and found that they were highly similar. We have added this result as Fig. S3J, copied below, which provides further evidence that estimating generalized phase within the 1–10 Hz frequency range led to phase estimates that were not mis-centered in the complex plane.

(J) Sample-wise phase difference between original phase estimates and phase estimates after local recentering in the complex plane using 10-second sliding-time windows. We performed this recentering to attenuate the effect

of potential shifts in the complex plane due to potential low-frequency intrusions. The clustering of the phase differences around 0° indicates a negligible effect of such potential shifts on phase estimates.

Overall, these empirical analyses indicate that generalized phase estimates within the 1–10 Hz range were not relevantly biased by low-frequency intrusions below 1 Hz.

In addition to these empirical analyses of our data, we aimed at validating the suitability of the 1–10 Hz generalized phase approach using simulated data, where we modeled various phenomena of local field potentials (Fig. S5, copied below). These conditions included changing amplitudes, changing frequencies, asymmetric oscillations, summation of strong 2-Hz oscillations and weaker 9-Hz oscillations, summation of potential low-frequency intrusions at 0.5-Hz oscillations and weaker 2-Hz oscillations, and signals with artifacts. In all cases except for artifacts, the generalized phase approach effectively captured the phase within the 1–10 Hz range. In particular, potential low-frequency intrusions at 0.5-Hz below the frequency-band-of-interest did not prevent accurate phase estimates of weaker 2-Hz oscillations. The only exception were artifacts, which is why we excluded periods with artifacts from our analyses (already in the original manuscript).

Figure S5. Generalized phase approach applied to simulated signals. (A–H) We tested the generalized phase approach using various simulated local field potentials (left). We plotted a power spectrum for each signal to inspect its different frequency components (middle left). Using the generalized phase approach including 1–10 Hz band-pass filtering, we obtained the analytic signal (middle) and instantaneous phases in the complex plane

representation (middle right). We then inspected the accuracy of the phase estimates (right). (A) Sine wave at 2 Hz. As expected, the generalized phase approach produced accurate phase estimates of this sinusoidal oscillation. (B) The phase estimates remained accurate despite varying amplitudes of the 2 Hz sine wave. (C) Oscillation with varying frequency. The accurate phase estimates show that the generalized phase approach has a particular advantage over methods with narrow-band filtering for oscillations with variable frequencies. (D–E) It also provides accurate phase estimates for oscillations with their peak tilted to one side and for oscillations with a triangle-like shape. (F) Summation of a dominant low-frequency component within the frequency-band-of-interest (2 Hz) and a weaker high-frequency component (9 Hz). The generalized phase approach provides valid phase estimates for the low-frequency component at 2 Hz. This is because the generalized phase approach has a general preference for lower frequencies, caused by the $1/f$ -like shape of the power spectrum. (G) Summation of dominant low-frequency intrusions below the frequency-band-of-interest (0.5 Hz) and a weaker low-frequency component within the frequency-band-of-interest (2 Hz). The 1–10 Hz band-pass filter effectively removes the contribution of low-frequency intrusions, leading to valid phase estimates of the 2-Hz oscillation. (H) Artifacts can strongly distort phase estimates. For this reason, we excluded time periods during artifacts from further analysis.

Action 3. We have added language to the revised Discussion stating that we included lower frequencies than the original generalized phase study, noting that this might be a limitation of our study (lines 705–719).

In this study, we chose a broad 1–10 Hz frequency range as theta to include both low-theta (1–5 Hz) and high-theta (6–10 Hz) components of the signal. We acknowledge that this may limit the direct comparability of our findings to studies using other definitions of the theta-frequency range such as 3–10 Hz, 4–8 Hz, or 2–10 Hz. As described above, visual inspection of our data suggested that the 1–10 Hz frequency range captured the raw signal well, which is why we opted for this frequency range. We also note that this band is different from the original generalized phase study (56), where a frequency range of 5–40 Hz was chosen to minimize effects related to arousal and to limit the influence of potential low-frequency intrusions that can cause mis-centering in the complex plane, leading to inaccurate phase estimates. Our control analyses did not indicate that 1–3 Hz activity was related to arousal in our awake human participants [in line with previous work, e.g., (16, 30, 31, 54)], and our analyses and simulations suggested no strong effects of potential low-frequency intrusions. We acknowledge, however, that we cannot fully exclude that some of our phase estimates may have been biased by low-frequency intrusions below the 1–10 Hz frequency-band-of-interest. We also acknowledge that it might be better to use the terms “non-oscillatory phase locking” or “non-rhythmic phase locking” to refer to our phase-locking effects relative to the broad 1–10 Hz frequency range, following previous work in bats (65).

Reviewer #3, comment #2

2.) Given the inclusion of broad frequencies, the results in the current form might be rather discussed as phase-locking of single neurons to non-oscillatory LFP fluctuations. As mentioned above, the presence of oscillations does not necessarily guarantee correct phase estimation here. To ensure the absence of slow frequencies and appropriate oscillatory phase estimates, filtering in the ~3–10 Hz might be required to claim phase locking to theta oscillations. It would be especially innovative and novel to distinguish between non-oscillatory phase locking and phase-locking to theta oscillations.

Response: Thanks a lot for this helpful comment. We respond by better explaining our decision of referring to the 1–10 Hz frequency as “theta.” We also show new results on theta-phase locking in the 3–10 Hz frequency range (Fig. S6C), which are qualitatively identical to the phase-locking results in the 1–10 Hz frequency range (Fig. 3I).

Based on previous work suggesting that human theta comprises both 1–5 Hz “low theta” and 6–10 Hz “high theta,” we decided to use a broad low-frequency band that included both bands. We do not believe that there are strong reasons to restrict the theta-frequency range to 3–10 Hz, as there are many different definitions of human theta, and it remains unclear to us which of these definitions is best. For example, a highly cited article by Klimesch, *Brain Research Reviews*, 1999 defines human theta as 4–7.5 Hz, which would miss important parts of both the low- and high-theta range. Here, we took the approach of treating activity within 1–10 Hz as a coherent entity whose frequency can dynamically change over time, and we refer to this band as “theta” as it includes both high and low theta. This decision was driven by visually inspecting the raw data, observing that narrow-band filtering often failed to fit these fluctuations neatly, and seeing that the generalized phase approach worked well in estimating the phase of these fluctuations. In the revised manuscript, we have added new language to better explain our rationale of using 1–10 Hz as theta-frequency range (lines 155–166).

In addition to this better fit of the raw signal, our choice of using a broad 1–10 Hz frequency range was motivated by several previous studies using intracranial neural recordings in humans. These studies showed that behavior-associated theta oscillations in humans occur across higher and lower theta frequencies, which extend beyond the traditional 4–8 Hz theta band and are often referred to as “high theta” (6–10 Hz) and “low theta” (1–5 Hz) [e.g., (16, 28, 30, 31, 66)]. For example, previous work found that increased low-theta power at 1–3 Hz is related to successful memory encoding (30, 54); that 3-Hz oscillations are often present during virtual spatial navigation (16, 31); and that high- and low-theta power increases before the onset of virtual movement (66). Hence, to not exclude any components of high theta and low theta, we opted for the 1–10 Hz frequency range and refer to fluctuations within this range as “theta” throughout the manuscript. We acknowledge though that human “theta” has been defined in various ways [e.g., as 4–7.5 Hz (67) or as 4–8 Hz (6)] and that the 1–10 Hz band includes parts of the traditional delta and alpha bands (67).

We are currently not sure whether the term “non-oscillatory phase locking” is a good choice because we cannot exclude the presence of oscillations in the signal. Nevertheless, we have added this idea to the Discussion of the manuscript, stating that our phase-locking results within the 1–10 Hz range may also be interpreted in the context of non-oscillatory phase locking (lines 717–718). In general, we are happy to further adjust our terminology if the Reviewer or Editor feel strongly about this.

We also acknowledge that it might be better to use the terms “non-oscillatory phase locking” or “non-rhythmic phase locking” to refer to our phase-locking effects relative to the broad 1–10 Hz frequency range, following previous work in bats (65).

In addition to these changes in language, based on your comments, we performed new analyses on theta-phase locking when restricting the frequency range to 3–10 Hz. Specifically, we examined the overall tendency of neurons to spike at the theta trough; the percentage of theta-phase locking neurons during baseline, encoding, and retrieval; and the strength of theta-phase locking in the presence versus absence of clear theta oscillations. We obtained similar results as compared to the 1–10 Hz frequency range, suggesting no particular advantage of the 3–10 Hz frequency range over the 1–10 Hz frequency range. We have added these results as Fig. S6.

Figure S6. Theta-phase locking in the 3–10 Hz frequency range. We repeated some of our main analyses on theta-phase locking with the only difference of using a bandpass filter of 3–10 Hz before estimating generalized phase to enhance comparability with other studies on theta oscillation in humans that did not include low frequencies between 1–3 Hz. **(A)** General theta-phase locking across the entire experiment. Spikes pooled across all neurons preferentially occurred at the theta trough. This pattern was consistent with the analysis of theta-phase locking relative to the 1–10 Hz frequency range (Fig. 2C). **(B)** Percentages of units with significant phase locking. Percentages were similar but lower compared to the 1–10 Hz frequency range (Fig. 2F). **(C)** Theta-phase locking during the presence versus absence of clear theta oscillations analyzed within the 1–10 Hz and 3–10 Hz frequency bands. Bars show mean \pm SEM across sessions. We performed an ANOVA for a linear mixed-effects model with “frequency band” and “oscillation detected” as fixed effects and “session” as a random effect [PPC \sim 1 + frequency band * oscillation detected + (1 | session)]. There was a significant main effect of “frequency band” ($F(2660) = 15.942$, $P < 0.001$) and a significant main effect of “oscillation detected” ($F(2660) = 32.198$, $P < 0.001$). The interaction between “frequency band” and “oscillation detected” was not significant ($F(2660) = 0.036$, $P = 0.850$). PPC-values in the condition “1–10 Hz, no oscillation detected” were significantly lower than in the condition “3–10 Hz, oscillation detected” (two-sided t -test: $t(665) = -2.389$, $P = 0.017$). These findings may help guide researchers in selecting a particular frequency-band-of-interest and in deciding whether to focus on signal segments with clear oscillations when investigating theta-phase locking. **(D)** 3–10 Hz theta-phase locking during the presence versus absence of clear theta oscillations detected by Bycycle, analyzed separately for baseline (left), encoding (middle), and recall (right). Empirical t -values of paired t -tests were ranked in surrogate distributions of t -values to assess significance (plots to the right). P -values were Bonferroni corrected for three tests. Effects were similar to those observed in the 1–10 Hz frequency range (Fig. 3I), with higher pairwise phase consistency values for the 1–10 Hz frequency range. PPC, pairwise phase consistency.

In a new analysis, we directly contrasted phase-locking strength between the 1–10 Hz and 3–10 Hz frequency ranges and observed that phase-locking strength was generally higher for the 1–10 Hz range (Fig. S6C, copied below). This speaks in favor of using 1–10 Hz, as it captures bigger effects in the data. Additionally, the figure directly addresses your comment that it would be innovative to distinguish between “non-oscillatory phase locking” and “phase-locking to theta oscillations” when following your nomenclature. Specifically, we found stronger theta-phase locking when comparing periods with clear theta oscillations in the 3–10 Hz frequency range as compared to periods without clear theta oscillations in the 1–10 Hz frequency range (Fig. S6C, copied below). As described above, however, we believe it is advantageous to use 1–10 Hz as “theta” (as it includes low theta) and we would thus prefer presenting our theta-phase locking results based on the 1–10 frequency range.

Figure S6. Theta-phase locking in the 3–10 Hz frequency range. ... (C) Theta-phase locking during the presence versus absence of clear theta oscillations analyzed within the 1–10 Hz and 3–10 Hz frequency bands. Bars show mean \pm SEM across sessions. We performed an ANOVA for a linear mixed-effects model with “frequency band” and “oscillation detected” as fixed effects and “session” as a random effect [PPC \sim 1 + frequency band * oscillation detected + (1 | session)]. There was a significant main effect of “frequency band” ($F(2660) = 15.942, P < 0.001$) and a significant main effect of “oscillation detected” ($F(2660) = 32.198, P < 0.001$). The interaction between “frequency band” and “oscillation detected” was not significant ($F(2660) = 0.036, P = 0.850$). PPC-values in the condition “1–10 Hz, no oscillation detected” were significantly lower than in the condition “3–10 Hz, oscillation detected” (two-sided t -test: $t(665) = -2.389, P = 0.017$). These findings may help guide researchers in selecting a particular frequency-band-of-interest and in deciding whether to focus on signal segments with clear oscillations when investigating theta-phase locking.

Furthermore, based on your other comments and to improve comparability with prior studies, we also provide theta-phase locking results for low theta (2–5 Hz) and high theta (6–9 Hz) using the filter-Hilbert method. We obtained similar (though weaker) results as compared to the 1–10 Hz frequency band. We have added these results as Fig. S7 for low theta and Fig. S8 for high theta, copied below.

Figure S7. Theta-phase locking to low theta (2–5 Hz). We repeated our main analyses on theta-phase locking in low theta (2–5 Hz), using the filter-Hilbert method for phase estimation. (A) Sample-wise phase difference between low-theta phases and generalized phases of the 1–10 Hz filtered signal during cycles with a cycle

frequency in the low-theta range. The moderate deviations might be caused by the narrower filtering for low theta compared to the 1–10 Hz signal. **(B)** General phase locking to low theta across the entire experiment. Spikes pooled across all neurons preferentially occurred at the trough. See Fig. 2C for comparison with the 1–10 Hz results. **(C)** Percentages of units with significant phase locking to low theta. See Fig. 2F for comparison with the 1–10 Hz results. **(D–F)** Phase locking to low theta as a function of power, aperiodic slopes, and oscillations. See Fig. 2E, 3E, and 3I for comparison with the 1–10 Hz results. **(G)** Phase locking to low theta as a function of memory performance during encoding. See Fig. 4, B and C, for comparison with the 1–10 Hz results. **(H)** Phase locking to low theta as a function of memory performance during retrieval. See Fig. 4, F and G, for comparison with the 1–10 Hz results. **(I–K)** Analysis of theta-phase shifts relative to low theta. Polar histograms show the angular differences between low theta phases during encoding versus retrieval for neurons with significant theta-phase shifts between encoding and retrieval. Results are shown for all units and for units with significant theta-phase locking. See Fig. 5C–E for comparison with the 1–10 Hz results. **(I)** Results for all segments. **(J)** Results for successful segments. **(K)** Results for unsuccessful segments. **(L)** Percentages of units with significant phase shifts for all (transparent color) and significantly phase locking units (opaque color). Percentages are compared to 5% chance level. See Fig. 5F for comparison with the 1–10 Hz results. $***P < 0.001$; $**P < 0.01$; $*P < 0.05$; n.s., not significant. All results are Bonferroni corrected for performing each test on low and high theta. PPC, pairwise phase consistency.

High theta (6-9 Hz)

Figure S8. Theta-phase locking to high theta (6-9 Hz). We repeated our main analyses on theta-phase locking for high theta (6-9 Hz), using the filter-Hilbert method for phase estimation. (A) Sample-wise phase difference between high-theta phases and generalized phases of the 1-10 Hz filtered signal during cycles with a cycle

frequency in the high-theta range. The moderate deviations might be caused by the narrower filtering for high theta compared to the 1–10 Hz signal. (B) General phase locking to high theta across the entire experiment. Spikes pooled across all neurons preferentially occurred at the trough. See Fig. 2C for comparison with the 1–10 Hz results. (C) Percentages of units with significant phase locking to high theta. See Fig. 2F for comparison with the 1–10 Hz results. (D–F) Phase locking to high theta as a function of power, aperiodic slopes, and oscillations. See Fig. 2E, 3E, and 3I for comparison with the 1–10 Hz results. (G) Phase locking to high theta as a function of memory performance during encoding. See Fig. 4, B and C, for comparison with the 1–10 Hz results. (H) Phase locking to high theta as a function of memory performance during retrieval. See Fig. 4, F and G, for comparison with the 1–10 Hz results. (I–K) Polar histograms show the angular differences between high theta phases during encoding versus retrieval for neurons with significant theta-phase shifts between encoding and retrieval. Results are shown for all units and for units with significant theta-phase locking. See Fig. 5C–E for comparison with the 1–10 Hz results. (I) Results for all segments. (J) Results for successful segments. (K) Results for unsuccessful segments. (L) Percentages of units with significant phase shifts for all (transparent color) and significantly phase locking units (opaque color). Percentages are compared to 5% chance level. See Fig. 5F for comparison with the 1–10 Hz results. $**P < 0.01$; n.s., not significant. All results are Bonferroni corrected for performing each test on low and high theta. PPC, pairwise phase consistency.

Overall, these results suggest that the 1–10 Hz frequency band is a good choice for investigating theta-phase locking in humans as it comprises both high theta and low theta and is associated with strongest theta-phase locking.

Reviewer #3, comment #i

Minor:

i) Please mention that participants actively navigated through the virtual environment using the keyboard, etc.

Response: We have adjusted the Results section accordingly (lines 99–101).

In this “Treasure Hunt” task, participants actively navigated a virtual environment using a game controller and were asked to encode and remember the locations of objects within the environment (Fig. 1; Methods).

We have also adjusted the caption of Fig. 1 (lines 121–123).

Figure 1. Spatial memory task. (A) During each navigation–encoding period of the task, participants freely and actively navigated a virtual beach environment using a game controller (1) and successively encountered two or three unique objects in treasure chests at random locations on the beach (2).

Reviewer #3, comment #ii

ii) Figure 1C: One would expect a chance level such as in E here; I guess the chance might be 1 divided by the number of objects. If so, some mention of the chance level would be helpful here.

Response: Thank you for this comment. During object recall, participants were instructed to freely recall the objects and verbalize them aloud. In principle, participants could have chosen

to give no verbal responses during any of the object-recall periods. To the best of our knowledge, this makes it impossible to define a meaningful chance level.

Reviewer #3, comment #iii

iii) Line 148: A brief description of the generalized-phase approach would be beneficial here. At least mention that the instantaneous phase is not simply estimated from the analytical signal⁵. From my understanding, generalized phase estimation overcomes the bias from low frequencies through filtering and suppresses confounds of higher frequencies by correcting for negative frequency components.

Response: Thank you for this comment, based on which we have added a brief explanation in the main text (lines 168–171).

In brief, the generalized phase approach involves filtering in a broad band (here, 1–10 Hz), computing the analytic signal using the Hilbert transform, estimating the phase from the analytic signal, and interpolating periods with high-frequency intrusions where phase progression reverses direction or phase progresses with a frequency <1 Hz (Fig. S3).

We have also included a supplementary figure in the revised manuscript, copied below, which describes the different steps of the generalized phase approach.

Figure S3. Generalized phase approach including illustration and control analyses. (A–F) Illustration of the generalized phase approach (56). (A) Raw local field potential (LFP). (B) The LFP is filtered between 1–10 Hz. This removes low frequency content <1 Hz, which would shift the analytic signal representation in the complex plane. Low-frequency intrusions, such as DC shifts, can distort phase estimates, if not removed. (C) The filtered signal is Hilbert-transformed to obtain an analytic signal with a real and an imaginary part. (D) Phases are computed in the complex plane representation using the four-quadrant arctangent function. (E) Phase estimates can be distorted by high-frequency intrusions (turquoise arrow). (F) The generalized phase approach corrects for

these distortions by identifying negative frequency epochs, in which estimated phase progression reverses direction or phase progresses with a frequency <1 Hz, and by replacing the phase values during these epochs with values from interpolation.

Reviewer #3, comment #iv

iv) Line 151. Some reference might be helpful here to support the observation that theta oscillations are non-continuous in humans^{6–8} in bats⁴.

Response: Thank you for pointing out these references, which we have added to the revised manuscript (lines 147–150).

We opted for this approach with a broad filter band of 1–10 Hz as our visual inspection of the data identified many periods in which the local field potential showed fluctuations and oscillatory activity with rapidly changing frequencies, similar to previous observations of non-stationary theta oscillations in bats and humans (16, 26, 27, 65).

Reviewer #3, comment #v

v) Phase-locking related to baseline periods. Theta power includes all frequencies between 1–10 Hz here. How could arousal effects in the 1–3 Hz range affect these results related to disengagement from any task?

Response: Thank you for this question. We agree that theta-phase locking during the baseline periods may have been influenced by participants being in a state of lower arousal, potentially leading to increases in 1–3 Hz power.

We note, however, that there are several findings in previous studies and our results suggesting that 1–3 Hz delta oscillations are not related to lower arousal in awake humans. For example, increased hippocampal 1–3 Hz power (“low theta”) during memory encoding has been shown to be related to successful memory recall (Miller et al., Nature Communications, 2018; Lega et al., Hippocampus, 2012); delta oscillations around 3 Hz are often present during virtual spatial navigation (Watrous et al., Hippocampus, 2013); increasing delta and theta oscillations are related to increasing virtual speed (Watrous et al., Journal of Neurophysiology, 2011); and 2–5 Hz power increases before virtual movement onset (Bush et al., PNAS, 2017). Similar to this previous intracranial work in humans, we found that 1–3 Hz power increased during encoding and that it tended to be higher during successful versus unsuccessful memory recall (Fig. S20).

Based on your comment, we performed an additional analysis where we quantified whether theta frequencies changed during the “distractor period.” In this “distractor period,” which is part of the baseline period, participants were asked to track the position of a coin hidden under one of three moving boxes (Fig. 1A). At the end of the distractor period, participants were asked to select the box under which the coin was hidden and received points for selecting the correct box (“successful distractor periods”). We compared successful and unsuccessful distractor periods, reasoning that arousal should be lower during unsuccessful periods, as

patients may have disengaged from the task and, consequently, missed the correct answer. We estimated the median theta frequency during successful and unsuccessful distractor trials (obtained through Bycycle) and compared these frequencies across sessions. The results showed no significant difference in median frequency. We have added these results to the manuscript (lines 177–178).

Analyzing the frequency distributions during the distractor period, we did not find evidence for the idea that lower arousal was related to a higher prevalence of 1–5 Hz activity (Fig. S4A).

Figure S4. Theta frequencies as a function of distractor task performance and aperiodic slope. (A) As we included the 1–3 Hz range in our 1–10 Hz theta-frequency band, we performed a control analysis to test whether 1–3 Hz components of this band might be related to arousal as reflected by task disengagement. Hence, assuming that unsuccessful performance in the distractor task could reflect such a disengagement, we compared the median cycle-by-cycle frequency during successful versus unsuccessful distractor trials. We did not find a difference in median frequency between successful and unsuccessful trials ($t(26) = -0.173, P = 0.864$), suggesting that 1–3 Hz frequencies did not increase due to task disengagement (i.e., lower arousal states).

Together, the above-mentioned previous and our findings do not indicate that 1–3 Hz activity reflects disengagement or low arousal states in awake humans engaged in cognitive tasks.

Reviewer #3, comment #vi

vi) Figure 2: Mention the frequency range for the LFP illustrated in panel B.

Response: Thanks for this suggestion. We have added the frequency range for the LFP in Fig. 2B, copied below.

Reviewer #3, comment #vii

vii) Line 195f: Briefly mention how power was computed here.

Response: Thank you for this suggestion. We have added this information to the revised manuscript (lines 226–228).

We thus asked whether theta-phase locking was stronger during periods of elevated theta power. To this end, we computed the sample-wise instantaneous theta power at each spike by squaring the magnitude of the complex signal obtained through the generalized phase approach.

Reviewer #3, comment #viii

viii) Line 255: I'm unsure if “neuronal spikes” is the best term here, given that spikes commonly refer to epileptic spikes; maybe “neuronal fluctuations” might be more appropriate.

Response: Thank you for this comment. Following previous publications (e.g., Rutishauser et al., Nature, 2010; Buzsáki et al., Nature Reviews Neuroscience, 2012), we now explicitly state that the term “spikes” refers to neuronal action potentials (lines 142–144).

We excluded neurons with fewer than 25 spikes [i.e., action potentials; (1, 37)] in total or no spikes in more than 80% of segments across any trial condition, resulting in a total number of 666 neurons for all analyses.

We also clarify that we use the term “interictal epileptiform discharges” instead of epileptic spikes (lines 886–892).

We automatically detected interictal epileptiform discharges (IEDs) in the preprocessed local field potentials following the same methods and detection criteria as described before (93). ... Spikes occurring during these timepoints were excluded from all analyses. We do not refer to IEDs as interictal spikes, because we reserved the term ‘spike’ for extracellularly recorded single-unit action potentials throughout the paper (1, 37).

Reviewer #3, comment #ix

ix) Line 272: Steeper slopes could also increase the influence of lower frequencies biasing phase estimation, as discussed above. Could you rule such influence out or estimate it?

Response: Thank you for this comment. Our data shows indeed that steeper slopes are associated with a higher prevalence of lower frequencies in the 1–10 Hz frequency band. On average, if the slope gets steeper by 1, then the frequency of the oscillatory cycles gets slower by 1.2 Hz. We have added this observation to the manuscript (lines 306–307).

In general, steeper slopes were associated with lower frequencies within the 1–10 Hz theta frequency range (Fig. S4, B and C).

We also show this relationship between aperiodic slope and frequency in Fig. S4, copied below.

Figure S4. Theta frequencies as a function of distractor task performance and aperiodic slope. ... (B) Histogram showing mean Spearman's ρ -values for the session-wise correlations between aperiodic slopes and cycle-by-cycle theta frequencies. Negative ρ -values indicate that steeper slopes were associated with lower frequencies within the 1–10 Hz theta frequency range (two-sided t -test of correlation values versus 0: $t(26) = -29.200$, $P < 0.001$). **(C).** To quantify aperiodic slope-associated frequency shifts, we calculated the session-wise mean slopes of linear fits between aperiodic slopes and cycle frequencies. A mean of approximately -1 indicates that an increase in aperiodic slope steepness by 1 corresponds to a frequency decrease of approximately 1 Hz.

In our view, this does not mean that steeper slopes result in biased theta-phase estimates. As described in our response to your comments 1 and 2, we suggest referring to the entire 1–10 Hz frequency range as “theta” (and not just to 3–10 Hz). The main reason for this is that we do not want to arbitrarily exclude parts of low theta (1–5 Hz). This broader definition allows for a more inclusive analysis that better captures the dynamics of theta oscillations across the full spectrum.

Reviewer #3, comment #x

x) Line 275ff: There might be a discrepancy in what “theta” means in the manuscript. Here, the presence of theta oscillations within the ~3–10 Hz range was tested (Fig. 3G), which resonates with previous literature, whereas phase-locking of single neurons to LFP and power in the theta range refers to phenomena at frequencies between 1–10 Hz.

Response: Thank you very much for this comment. In our manuscript, we use “theta” to refer to the 1–10 Hz frequency range throughout the text (e.g., lines 22–23). We nevertheless agree that, based on the empirical distribution of theta frequencies in Fig. 3G of the original manuscript, most of the theta oscillations identified through Spectral Parameterization Resolved in Time (SPRiNT) were in the frequency range of 3–10 Hz.

In the revised manuscript, we have changed our oscillation-detection method from SPRiNT to Bycycle (Coyle & Voytek, Journal of Neurophysiology, 2019) based on several reviewer comments. Bycycle identifies oscillations in the time domain and thus has two advantages over SPRiNT. First, its temporal resolution is better because SPRiNT requires relatively long time windows (3 seconds) for computing the power spectra. Second, it enables a better identification of low-theta oscillations close to 1 Hz because SPRiNT often has difficulties in accurately fitting the power spectra at these low frequencies. Using Bycycle, the frequency distribution of

detected theta oscillations contains more oscillations at <3 Hz (cumulative percentage of 30%), which is better in line with our broad definition of theta as spanning the 1–10 Hz frequency range, including both low and high theta. We have copied the revised figure below (new Fig. 3H).

(H) Probability distribution of frequencies of detected theta oscillations (see Fig. S16 for results from different brain regions). Gray shaded area, standard error of the mean across microwires.

In addition, based on your comment, we have investigated theta-phase locking when estimating generalized phase after filtering in the 3–10 Hz frequency range. We obtain qualitatively identical results as compared to the 1–10 Hz frequency range, showing stronger theta-phase locking during versus outside clear theta oscillations (consistently across the three task periods of baseline, encoding, and retrieval). This suggests that our findings are robust against a particular definition of the theta-frequency range.

Figure S6. Theta-phase locking in the 3–10 Hz frequency range. We repeated some of our main analyses on theta-phase locking with the only difference of using a bandpass filter of 3–10 Hz before estimating generalized phase to enhance comparability with other studies on theta oscillation in humans that did not include low frequencies between 1–3 Hz. **(A)** General theta-phase locking across the entire experiment. Spikes pooled across all neurons preferentially occurred at the theta trough. This pattern was consistent with the analysis of theta-phase locking relative to the 1–10 Hz frequency range (Fig. 2C). **(B)** Percentages of units with significant phase locking. Percentages were similar but lower compared to the 1–10 Hz frequency range (Fig. 2F). **(C)** Theta-phase locking

during the presence versus absence of clear theta oscillations analyzed within the 1–10 Hz and 3–10 Hz frequency bands. Bars show mean \pm SEM across sessions. We performed an ANOVA for a linear mixed-effects model with “frequency band” and “oscillation detected” as fixed effects and “session” as a random effect [PPC \sim 1 + frequency band * oscillation detected + (1 | session)]. There was a significant main effect of “frequency band” ($F(2660) = 15.942, P < 0.001$) and a significant main effect of “oscillation detected” ($F(2660) = 32.198, P < 0.001$). The interaction between “frequency band” and “oscillation detected” was not significant ($F(2660) = 0.036, P = 0.850$). PPC-values in the condition “1–10 Hz, no oscillation detected” were significantly lower than in the condition “3–10 Hz, oscillation detected” (two-sided t -test: $t(665) = -2.389, P = 0.017$). These findings may help guide researchers in selecting a particular frequency-band-of-interest and in deciding whether to focus on signal segments with clear oscillations when investigating theta-phase locking. (D) 3–10 Hz theta-phase locking during the presence versus absence of clear theta oscillations detected by Bicycle, analyzed separately for baseline (left), encoding (middle), and recall (right). Empirical t -values of paired t -tests were ranked in surrogate distributions of t -values to assess significance (plots to the right). P -values were Bonferroni corrected for three tests. Effects were similar to those observed in the 1–10 Hz frequency range (Fig. 3I), with higher pairwise phase consistency values for the 1–10 Hz frequency range. PPC, pairwise phase consistency.

Reviewer #3, comment #xi

xi) Figure 4E: It would be interesting to see the phase differences between encoding and retrieval as a function of frequency.

Response: Thank you for this suggestion, based on which we have quantified the number of units with significant theta-phase shifts separately for spikes occurring at lower versus higher theta frequencies (median split of spike-associated theta frequencies per unit). We found a similar number of phase-shifting neurons in both frequency groups. We describe this observation in the main text (lines 510–512).

The number of phase-shifting neurons was significant for spikes occurring at both higher and lower frequencies within the 1–10 Hz band (both 56 of 666; 8.4%; binomial test, $P_{\text{corr.}} < 0.001$, Bonferroni corrected; Fig. S21G).

We present this result as part of a supplementary figure (Fig. S21, copied below).

Figure S21. Encoding–retrieval phase shifts as a function of **memory performance, theta power, aperiodic slope, clear theta oscillations, and theta frequency.** ... **(G)** Phase shifts for upper and lower spike-associated instantaneous frequencies (unit-wise median split).

Similarly, when performing the analysis of theta-phase shifts separately for the low- and high-theta frequency band (2–5 Hz and 6–9 Hz, respectively, following the definitions in Bush et al., PNAS, 2017, which are similar to the definitions in your general comment) using the filter-Hilbert method, we found phase-shifting neurons in both bands, with stronger effects for low theta. We have added these results as supplementary figures (Figs. S7 and S8), copied below.

Figure S7. Theta-phase locking to low theta (2–5 Hz). We repeated our main analyses on theta-phase locking in low theta (2–5 Hz), using the filter-Hilbert method for phase estimation. (A) Sample-wise phase difference between low-theta phases and generalized phases of the 1–10 Hz filtered signal during cycles with a cycle

frequency in the low-theta range. The moderate deviations might be caused by the narrower filtering for low theta compared to the 1–10 Hz signal. **(B)** General phase locking to low theta across the entire experiment. Spikes pooled across all neurons preferentially occurred at the trough. See Fig. 2C for comparison with the 1–10 Hz results. **(C)** Percentages of units with significant phase locking to low theta. See Fig. 2F for comparison with the 1–10 Hz results. **(D–F)** Phase locking to low theta as a function of power, aperiodic slopes, and oscillations. See Fig. 2E, 3E, and 3I for comparison with the 1–10 Hz results. **(G)** Phase locking to low theta as a function of memory performance during encoding. See Fig. 4, B and C, for comparison with the 1–10 Hz results. **(H)** Phase locking to low theta as a function of memory performance during retrieval. See Fig. 4, F and G, for comparison with the 1–10 Hz results. **(I–K)** Analysis of theta-phase shifts relative to low theta. Polar histograms show the angular differences between low theta phases during encoding versus retrieval for neurons with significant theta-phase shifts between encoding and retrieval. Results are shown for all units and for units with significant theta-phase locking. See Fig. 5C–E for comparison with the 1–10 Hz results. **(I)** Results for all segments. **(J)** Results for successful segments. **(K)** Results for unsuccessful segments. **(L)** Percentages of units with significant phase shifts for all (transparent color) and significantly phase locking units (opaque color). Percentages are compared to 5% chance level. See Fig. 5F for comparison with the 1–10 Hz results. $***P < 0.001$; $**P < 0.01$; $*P < 0.05$; n.s., not significant. All results are Bonferroni corrected for performing each test on low and high theta. PPC, pairwise phase consistency.

High theta (6-9 Hz)

Figure S8. Theta-phase locking to high theta (6-9 Hz). We repeated our main analyses on theta-phase locking for high theta (6-9 Hz), using the filter-Hilbert method for phase estimation. (A) Sample-wise phase difference between high-theta phases and generalized phases of the 1-10 Hz filtered signal during cycles with a cycle

frequency in the high-theta range. The moderate deviations might be caused by the narrower filtering for high theta compared to the 1–10 Hz signal. **(B)** General phase locking to high theta across the entire experiment. Spikes pooled across all neurons preferentially occurred at the trough. See Fig. 2C for comparison with the 1–10 Hz results. **(C)** Percentages of units with significant phase locking to high theta. See Fig. 2F for comparison with the 1–10 Hz results. **(D–F)** Phase locking to high theta as a function of power, aperiodic slopes, and oscillations. See Fig. 2E, 3E, and 3I for comparison with the 1–10 Hz results. **(G)** Phase locking to high theta as a function of memory performance during encoding. See Fig. 4, B and C, for comparison with the 1–10 Hz results. **(H)** Phase locking to high theta as a function of memory performance during retrieval. See Fig. 4, F and G, for comparison with the 1–10 Hz results. **(I–K)** Polar histograms show the angular differences between high theta phases during encoding versus retrieval for neurons with significant theta-phase shifts between encoding and retrieval. Results are shown for all units and for units with significant theta-phase locking. See Fig. 5C–E for comparison with the 1–10 Hz results. **(I)** Results for all segments. **(J)** Results for successful segments. **(K)** Results for unsuccessful segments. **(L)** Percentages of units with significant phase shifts for all (transparent color) and significantly phase locking units (opaque color). Percentages are compared to 5% chance level. See Fig. 5F for comparison with the 1–10 Hz results. $**P < 0.01$; n.s., not significant. All results are Bonferroni corrected for performing each test on low and high theta. PPC, pairwise phase consistency.

Reviewer #3, comment #xii

xii) Figure 5: The sign of the phase differences is not clear here (encoding-retrieval). From the examples in the polar plots, it looks like neurons lock to earlier phases during retrieval than encoding, which is the opposite of the hypothetical example as illustrated on top. Please clarify.

Response: We apologize for the lack of clarity here. We have adjusted the legend of Fig. 5 to better explain the phase differences.

In the legend of Fig. 5A, we now state that the illustration depicts a single hypothetical phase shift from an earlier to a later phase (lines 523–525).

Figure 5. Shifts of preferred theta phases between encoding and retrieval. **(A)** Hypothesis that neurons shift their preferred theta phases between encoding and retrieval (49). The illustration shows a hypothetical shift from an earlier phase during encoding toward a later phase during retrieval.

In the legend of Fig. 5B, we have clarified that all four examples show shifts toward earlier phases during retrieval (lines 525–530).

(B) Four example units with significantly shifted theta phases between encoding and retrieval. ... All four examples show a shift from a later phase during encoding toward an earlier phase during retrieval.

Across the population of neurons, we find that theta-phase shifts between encoding and retrieval occur in both directions, with no systematic pattern (Fig. 5C–E). We have added this information to the revised figure legend (lines 530–534).

(C–E) Polar histograms show the angular differences between theta phases during encoding versus retrieval for neurons with significant theta-phase shifts between encoding and retrieval. Results are shown for all units and for units with significant theta-phase locking. A positive angular difference corresponds to a shift from an earlier phase during encoding to a later phase during retrieval. Shifts occurred in both directions.

Additionally, to visualize the phase shifts in greater detail, we have added a Supplementary Figure where we plot the preferred theta-phase during encoding versus the preferred theta-phase during retrieval, separately for each neuron with a significant theta-phase shift (Fig. S21, copied below).

Figure S21. Encoding–retrieval phase shifts as a function of memory performance, theta power, aperiodic slope, clear theta oscillations, and theta frequency. (A–C) Polar plots show the angular shifts between theta phases during encoding (blue dots) versus retrieval (orange dots) for neurons with significant theta-phase shifts between encoding and retrieval. Results are shown for all units (darker and brighter lines) and for units with significant theta-phase locking (darker lines). Units are sorted from center to periphery by decreasing phase difference (larger phase differences toward the center). (A) Results for all segments ($n = 62$ significant units from the pool of all units; $n = 26$ significant units from the pool of significantly phase-locking units). (B) Results for successful segments ($n = 54$ and $n = 14$, respectively). (C) Results for unsuccessful segments ($n = 43$ and $n = 12$, respectively).

Reviewer #3, references

1. Davis, Z. W., Muller, L., Martinez-Trujillo, J., Sejnowski, T. & Reynolds, J. H. Spontaneous travelling cortical waves gate perception in behaving primates. *Nature* 587, 432–436 (2020).
2. Rutishauser, U., Ross, I. B., Mamelak, A. N. & Schuman, E. M. Human memory strength is predicted by theta-frequency phase-locking of single neurons. *Nature* 464, 903–907 (2010).
3. Yoo, H. B., Umbach, G. & Lega, B. Neurons in the human medial temporal lobe track multiple temporal contexts during episodic memory processing. *NeuroImage* 245, 118689 (2021).
4. Eliav, T. et al. Nonoscillatory Phase Coding and Synchronization in the Bat Hippocampal Formation. *Cell* 175, 1119–1130.e15 (2018).
5. Gabor, D. Theory of communication. Part 1: The analysis of information. *Journal of the Institution of Electrical Engineers - Part III: Radio and Communication Engineering* 93, 429–441 (1946).
6. Watrous, A. J. et al. A comparative study of human and rat hippocampal low-frequency oscillations during spatial navigation. *Hippocampus* 23, 656–661 (2013).
7. Aghajani, Z. M. et al. Theta Oscillations in the Human Medial Temporal Lobe during Real-World Ambulatory Movement. *Curr Biol* 27, 3743–3751.e3 (2017).

8. Bohbot, V. D., Copara, M. S., Gotman, J. & Ekstrom, A. D. *Low-frequency theta oscillations in the human hippocampus during real-world and virtual navigation. Nat Commun 8, 14415 (2017).*

Theta-phase locking of single neurons during human spatial memory

Response to Reviewers

We thank the Reviewers and the Editor for the positive reassessment of our manuscript.

Below, we provide a point-by-point response to the Reviewers' comments. Reviewer comments are shown in *italic* font, responses are shown in normal font, and text copied from the manuscript is shown in blue font. Text changed in the manuscript is highlighted in yellow.

We would be happy to make any further changes if needed.

Reviewer #2, general comment

I thank the authors for revising their manuscript accounting for my comments. I really appreciate the whole new analysis of the data, and additions for relating aperiodic slopes and theta oscillations. The pedagogical efforts in figure S12 and Fig3C are also a greatly appreciated. I think the manuscript is much stronger and the nature of the results it provides are clearer now and deserves publication.

Response: Thank you!

Reviewer #3, general comment

The authors addressed all of my comments, and I recommend publication. The rebuttal is rigorous, comprehensive, and genuinely insightful. The additional data and analyses supported the author's main findings and clarified additional questions.

Response: Thank you very much for the positive assessment of our revised manuscript.

Reviewer #3, comment #1

+ I do not insist on the terminology “non-oscillatory phase locking”; the acknowledgment that non-stationary theta oscillations were observed in this study would suffice. I leave the exact terminology to the authors.

Response: We appreciate the Reviewer’s flexibility regarding terminology. In agreement with the Reviewer’s comment, our main text states explicitly that theta oscillations in our study were non-stationary (lines 120–123).

Reviewer #3, comment #2

+ It would be interesting to test whether there are systematic phase shifts for the data presented in Fig. S21G, as the upper frequency shows a cluster of phase shifts between 30-60 degrees. One way to assess this would be to test whether the population's mean vector has a preferred phase by using phase circular statistics, e.g., comparing to surrogate data with shuffled phase-to-spike relations.

Response: Following the Reviewer’s suggestion, we have tested whether there are systematic phase shifts for the data presented in Fig. S21G. Our results show that there is no systematic phase shift:

(G) Phase shifts for upper and lower spike-associated instantaneous frequencies (unit-wise median split). To test for a preferred angle in upper-frequency phase shifts, we computed the pairwise phase consistency (PPC = 0.005) and compared it against 10,001 surrogate PPC values from random distributions. The resulting *P*-value of 0.277 indicated no significant angular preference.

Reviewer #3, comment #3

+ In Fig. S21D-G, what "10" means in the polar plot could be more apparent. I assume the number of neurons, so this label could be plotted in grey or briefly mentioned in the caption (to distinguish the phases in the polar plot).

Response: Thank you. We have adjusted Fig. S21D–G as suggested.